# Proliferating coacervate droplets as the missing link between chemistry and biology in the origins of life

Muneyuki Matsuo [1,2,3] & Kensuke Kurihara [3,4,5,6,7✉]

The hypothesis that prebiotic molecules were transformed into polymers that evolved into proliferating molecular assemblages and eventually a primitive cell was first proposed about 100 years ago. To the best of our knowledge, however, no model of a proliferating prebiotic system has yet been realised because different conditions are required for polymer generation and self-assembly. In this study, we identify conditions suitable for concurrent peptide generation and self-assembly, and we show how a proliferating peptide-based droplet could be created by using synthesised amino acid thioesters as prebiotic monomers. Oligopeptides generated from the monomers spontaneously formed droplets through liquid–liquid phase separation in water. The droplets underwent a steady growth–division cycle by periodic addition of monomers through autocatalytic self-reproduction. Heterogeneous enrichment of RNA and lipids within droplets enabled RNA to protect the droplet from dissolution by lipids. These results provide experimental constructs for origins-of-life research and open up directions in the development of peptide-based materials.

[1] Department of Chemistry, Graduate School of Integrated Sciences for Life, Hiroshima University, Hiroshima, Japan. [2] Department of Basic Science, Graduate School of Arts and Sciences, The University of Tokyo, Komaba, Meguro, Tokyo, Japan. [3] Department of Creative Research, Exploratory Research Center on Life and Living Systems (ExCELLS), National Institutes of Natural Sciences, Myodaiji, Okazaki, Aichi, Japan. [4] Institute of Laser Engineering, Osaka University, Suita, Osaka, Japan. [5] Institute for Extra-cutting-edge Science and Technology Avant-garde Research (X-star), Japan Agency for Marine-Earth Science & Technology (JAMSTEC), Yokosuka, Kanagawa, Japan. [6] Faculty of Education, Utsunomiya University, Utsumomiya, Tochigi, Japan. [7] Department of Life and Coordination-Complex Molecular Science, Biomolecular Functions, Institute for Molecular Science, National Institutes of Natural Sciences, Myodaiji, Okazaki, Aichi, Japan. ✉email: kkurihara@cc.utsunomiya-u.ac.jp

Construction in a laboratory of a model protocell, which is an artificial supramolecular system that expresses the essential properties of life, would be an important step in identifying the pathway by which life originated[1,2]. In the 1920s, Oparin[3] and Haldane[4] independently proposed an origin-of-life scenario, and in the 1930s they claimed that a protocell, i.e., a primitive cell, was a proliferating droplet such as a coacervate droplet (CD). According to the abiogenesis scenario, a CD is formed from organic molecules via liquid–liquid phase separation (LLPS) and results mainly from the spontaneous assembly of oppositely charged molecules or from hydrophobic polymers and, in particular, of prebiotic polymers or oligomers. There has been a report of a CD[5] that expresses a higher-order function, which is a function that is not expressed by a single molecule but instead is expressed through intermolecular interactions. Higher-order functions affect the properties of molecular aggregates by, inter alia, encapsulating the main constituents of living organisms (i.e., RNA, lipids, and peptides). However, no CD has yet been constructed that induces interactions among its constituents that lead to self-reproduction, i.e., reproduction that takes place from within the closed boundary of the structure itself. In none of the hypotheses concerning the origin of life has there been an experimental proof of how the molecular assemblies formed by primitive molecules came to proliferate. Experimental proof of proliferation by simple physical mechanisms, independent of molecular species, would therefore be an important milestone in the maturation of explanations for the origins of life. In previous studies, CDs have been produced experimentally at room temperature and under ambient atmospheric pressure using peptides generated at high temperatures or pressures that mimicked the volcanic conditions prevalent on primordial Earth[6,7]. Recent studies have reported the formation of CDs composed of oligonucleotides[8], phospholipids[9], or oligopeptides[10] in aqueous solutions. Renewed interest in the role of CDs in the origins of life has been generated by the studies of Mann and co-workers, who have demonstrated cellular life-like features such as communication[11] and predator−prey interactions[12] in populations of CDs. As is the case with CD formation, autocatalytic self-reproduction is a crucial property of protocells that proliferate steadily[13] and has been demonstrated for supramolecular structures such as DNA origami rafts[14,15], lipid micelles[16–18], and lipid vesicles[19–21] in an aqueous medium. Previous studies of the fusion and division of molecular assemblies without self-reproduction have reported the importance of non-equilibrium states[22–24]. However, few studies have reported recursive self-reproduction of supramolecular assemblies in response to periodic stimuli[25] because metastable assemblies tend to move toward equilibrium. Under present conditions, the robustness of cellular organisms—the ability to use intrinsic response mechanisms to maintain an almost constant state vis-a-vis external stimuli—requires not only self-reproduction but also recursiveness under conditions of cyclic stimulation[26,27]. For example, the division of cyanobacterial cells is synchronised with the light–dark cycle associated with Earth's rotation[28], and L-form bacteria proliferate by membrane destabilisation caused by excessive membrane production and repeated perturbations from the environment, e.g., water flow[29,30]. Self-reproduction and periodic stimuli may have played crucial roles that enabled recursive proliferation (i.e., growth and division through self-reproduction) of prebiotic supramolecular assemblies on primitive Earth. The proliferation of molecular assemblies through self-reproduction to form sophisticated supramolecular assemblies is a "biological" property specific to organisms and has not been observed in viruses and molecular replicators[31–33]. The formation of polymers from monomers and of molecular assemblies from polymers are common "chemical" properties in nature and are based on interactions such as covalent bonding and intermolecular forces. The creation of proliferating CDs via such mechanisms, however, has not been achieved at all. The problem of mimicking this step of chemical evolution in the origins of life has never been solved experimentally during the roughly hundred years since it was first proposed[1,2]. In principle, a CD can self-reproduce only if the conditions are satisfied for the reproduction of both the CD itself and of the peptides that are the building blocks of the CD. In previous studies, peptides that are constituents of CDs have been produced in an elaborate manner via organic synthesis under volcanic conditions, biosynthesis, or solid-phase synthesis. CDs have then been formed by the produced peptides under mild, aqueous conditions[6–12].

In this work, to generate a proliferating droplet through prebiotic polymerisation, we construct an autocatalytic, self-reproducing droplet via LLPS. The procedure was inspired by de Duve's "thioester world" hypothesis, which argues that prebiotic peptides might have been generated from amino acid thioesters under mild, aqueous conditions[34]. We were able to simultaneously form droplets via LLPS and generate peptides by using a designed and synthesised thioesterified cysteine derivative as a monomer precursor for the spontaneous oligomerisation of an amino acid thioester under mild, aqueous conditions. A continuous supply of a monomer precursor that kept the LLPS-formed droplets in a non-equilibrium state enabled the LLPS-formed droplets to undergo a steady growth-division cycle that maintained droplet size while increasing the number of droplets. We also showed that LLPS-formed droplets were able to resist dissolution by lipids and to maintain themselves when nucleic acids and lipids were both present in them if the concentrated nucleic acids were localised at the inner boundary of the LLPS-formed droplet with the assistance of generated peptides. Overall, we were able to demonstrate how a proliferating droplet protocell could be formed by the oligomerisation of amino acid thioesters and functionalised by oligonucleotides (Fig. 1). Such a protocell could have served as a link between "chemistry" and "biology" during the origins of life. This study may serve to explain the emergence of the first living organisms on primordial Earth.

## Results and discussion

**Spontaneous formation of droplets from amino acid thioesters.** We designed a monomer $M$ that was capable of producing peptides and of facilitating the self-assembly of molecules under aqueous conditions sufficiently mild to allow the formation of droplets and self-reproduction of the building blocks of $M$ (Fig. 2a). A monomer with a thioester and unprotected cysteine group at its C- and N-terminus, respectively, would be expected to polymerise spontaneously in water. A disulfide precursor of $M$ ($M_{pre}$) was synthesised (Supplementary Figs. 1–3) to facilitate the production of $M$ via reduction of $M_{pre}$ with dithiothreitol (DTT). The C-terminus of $M$ was capped with a benzyl mercaptan (BnSH) leaving group.

The reduction of the $M_{pre}$ disulfide by DTT in water generated two $M$ molecules, which then reacted spontaneously to yield peptides (Fig. 2a). To confirm the formation of droplets, we monitored turbidity as a function of time and recorded microscopic images of an aqueous solution containing $M_{pre}$ and DTT (Fig. 2b, c). Five minutes after the addition of $M_{pre}$ (5 mg, 10 mM) and DTT (4 mg, 25 mM) to deionised water, the turbidity of the water began to increase, and it continued to increase for 16 h (Fig. 2b, c [red line]). The increase of the turbidity suggested that molecular self-assemblies had been formed in the solution. We, therefore, examined the solution with a differential interference contrast (DIC) microscope to confirm the formation of molecular self-assemblies induced by a

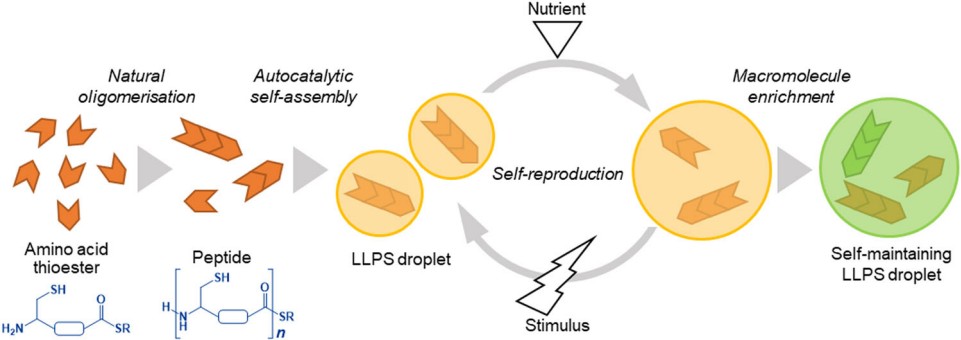

**Fig. 1 Emergence of proliferating and self-maintaining droplet protocells.** In the first stage, the amino acid thioester is oligomerised to produce a peptide. Droplets are formed from the product by liquid−liquid phase separation (LLPS). Continuous addition of the amino acid thioesters as a source of nutrition and physical stimulus to the droplets allows the droplets to divide while they self-reproducing autocatalytically through the incorporation of nutrients. The robustness of the proliferating droplet reflects its ability to concentrate macromolecules such as nucleic acids.

series of reactions after addition of $M_{pre}$ (Fig. 2d and Supplementary Movie 1). No molecular self-assemblies were observed within the first 5 min after mixing the $M_{pre}$ and DTT reagents; however, micrometre-sized molecular self-assemblies appeared after 1 h. Twenty-seven hours after mixing, the spherical molecular self-assemblies had grown larger and were present in significant numbers. Furthermore, DIC microscopy revealed that fusion of the molecular self-assemblies (Supplementary Movie 2) had begun 1 h after the addition of $M_{pre}$. The spherical shapes maintained by the fused assemblies suggested that the formed aggregates were droplets.

To confirm that the reduction of $M_{pre}$ induced the development of turbidity in the solution containing both $M_{pre}$ and DTT, the solution was observed microscopically in real-time, and the turbidity of an aliquot of the solution was measured every hour. When $M_{pre}$ was dissolved in deionised water in the absence of DTT, no molecular assemblies larger than 1 μm were observed via DIC microscopy, although smaller assemblies resulted in the development of slight turbidity (Fig. 2c [green line] and Supplementary Fig. 4a). Moreover, the turbidity was nearly zero when only DTT was dissolved in water because DTT is soluble in water (Fig. 2c [blue line]). In contrast to the turbidity caused by the dispersed reaction that occurred in the solution after the addition of thioesterified cystine $M_{pre}$ and DTT, no turbidity was observed when cystine was mixed with cysteine dihydrochloride, BnSH, and DTT (Supplementary Fig. 4b, c) because the molecular self-assemblies precipitated rather than dispersed, and the aggregates precipitated in a similar way with the addition of only cystine to water (Supplementary Fig. 4d). These results confirmed that droplet formation resulted from thioester-induced reactions. To clarify the contribution of the cysteine moiety in the monomers to droplet formation, we also synthesised thioesterified glycine (Gly-SBn) ("Methods" and Supplementary Fig. 5a, b) by a protocol similar to that for the synthesis of $M_{pre}$ (Supplementary Fig. 1). There was no turbidity in an aqueous solution containing Gly-SBn and DTT at either 5 min or 24 h after its preparation (Supplementary Fig. 6a, b). These results indicated that the reduction of the disulfide precursor with DTT and the subsequent chemical reaction at the thioester site and the cysteine side chain of $M_{pre}$ were essential for spontaneous droplet formation. The pH range for droplet formation was at least 3–11 (Supplementary Fig. 7a–e).

**Oligomerisation-induced self-assembly of LLPS-formed droplets via autocatalysis.** To verify that oligomerisation was induced at the thioester site and the cysteine side chain of $M$, we allowed mixing of $M_{pre}$ (10 mM) with DTT (25 mM) to reduce

$M_{pre}$ and oligomerise the generated $M$ in deionised water. The product was separated from the droplet dispersion. The reaction solution was lyophilised to remove water and BnSH, and the white powder residue was washed with acetonitrile to remove unreacted, oxidised DTT (Fig. 3a and Supplementary Fig. 8). The obtained powder was analysed by proton nuclear magnetic resonance ($^{1}$H NMR) (Supplementary Fig. 9a) and electrospray ionisation mass spectrometry with a time-of-flight mass spectrometer (ESI-TOF-MS). Comparison of oligopeptide spectra with those of $M_{pre}$ (Supplementary Fig. 9b) showed that the peak of the benzene ring (peak a in Supplementary Fig. 9b) and the peaks near the disulfide (peaks b and c in Supplementary Fig. 9b) had almost disappeared, whereas an amide proton (d) was newly detected. The mean degree of polymerisation of the obtained powder was estimated from the ratio of the areas of the peaks at 8.8 and 4.8 ppm in Supplementary Fig. 9a to be 4.1. Each peak was assigned to the proton of terminal amine groups and amide bonds, respectively. In addition, the mass-to-charge ratios observed in the ESI-TOF-mass spectra revealed degrees of polymerisation of 2 to at least 4 in the reaction solution 24 h after mixing $M_{pre}$ and DTT (Supplementary Fig. 10a–c). The intensities of the monomer and dimer peaks decreased with time, whereas the intensities of the trimer and tetramer peaks increased. These results indicated that $M_{pre}$ was reduced by DTT to yield $M$, which then formed at least di-, tri-, and tetra-peptides.

Products, with the exception of biproducts (BnSH and oxidised DTT), were purified from the solution 24 h after mixing $M_{pre}$ and DTT to clarify the contribution of the generated oligopeptide to droplet formation. The ability of the oligopeptide to form a droplet was then investigated (Table in Fig. 3a). The solutions that contained no oligopeptides or BnSH did not become turbid, whereas those containing both oligopeptides and BnSH were dispersed, and the formation of spherical molecular assemblies in them was confirmed via DIC microscopy (Supplementary Fig. 11). To clarify the composition of the droplets, we investigated the ratio of oligopeptides to BnSH in the droplets. Few droplets were formed when the concentration of the oligopeptides was ≤1 mM (amino acid equivalent concentration) or when the solution contained no BnSH (Fig. 3b, Supplementary Fig. 12). The indication that oligomerisation-induced self-assembly had occurred in these mixtures strongly suggested that the association of oligopeptides and BnSH was efficient for droplet formation via LLPS. We concluded that the droplets were formed by associative LLPS[35]. In addition to hydrophobic interactions, π–π interactions and cation–π interactions are plausible mechanisms for droplet formation. The shift of the peak in the NMR spectrum was an indication of the effect of the π–π interactions (Supplementary Fig. 13a). The fact that the terminus of the peptide is an

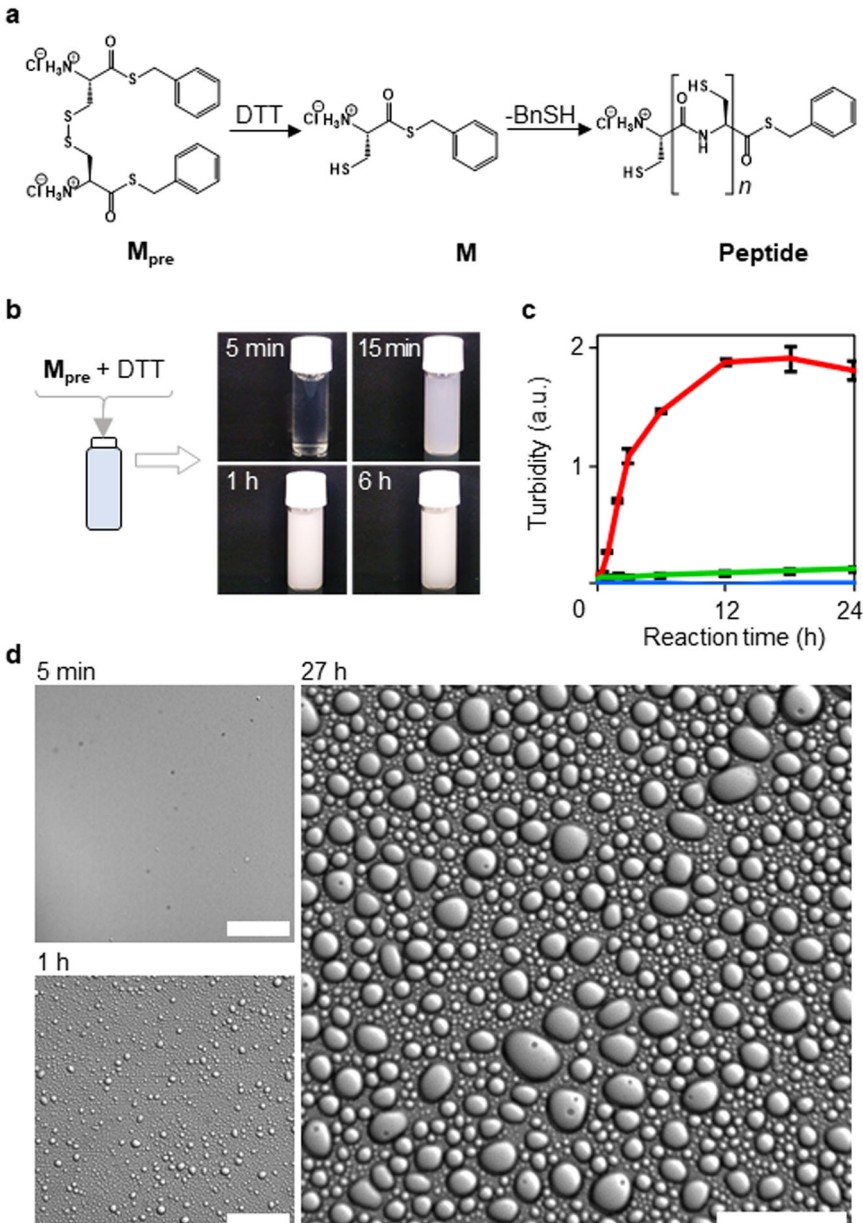

**Fig. 2 Droplet formation from M_pre and DTT. a** Reduction of the monomer precursor **M_pre** and oligomerisation of **M**. **b** Preparation of specimens: time-lapse photographs taken after addition of **M_pre** (10 mM) and DTT (25 mM) to water. **c** Change in turbidity after addition of **M_pre** (10 mM) and DTT (25 mM), either together or individually, to water: red line, **M_pre**, and DTT mixture; green line, aqueous solution of **M_pre**; blue line, aqueous solution of DTT. The number of trials was five. Data are presented as mean values +/− SD. Source Data is provided in the Source Data File. **d** DIC microscopy images of the solution of **M_pre** (50 mM) and DTT (125 mM) after mixing. Scale bars represent 40 μm.

ammonium cation and that BnSH has a benzene ring suggests the possibility that cation–π interactions are involved in droplet formation. Indeed, it has been reported that the formation of droplets via LLPS in vivo is due to cation–π interactions between lysine residues with an ammonium cation and other amino acid residues with an aromatic ring in the protein side chain[36]. The distinctive contribution of the thiol to droplet formation, such as thiol–π interactions or disulfide bonds, should also be considered[37].

The proportion of the **M_pre** residual, i.e., the proportion of the primary amines that were not involved in peptide formation, was determined from the amount of **M_pre** that was consumed (Fig. 3c), which was estimated by the fluorescamine method (Supplementary Fig. 13b). The conversion of **M_pre** to droplet material was calculated from the changes of the areas of

the peaks corresponding to benzene ring protons in the $^{1}$H NMR spectrum of the solution (Supplementary Fig. 13c). The decrease of the **M_pre** residual was consistent with the conversion of **M_pre** to droplets (Fig. 3c). The curve of the conversion of **M_pre** to droplets was sigmoidal and was fit to an autocatalytic reaction equation using the Levenberg–Marquardt method on the assumption that the reaction was autocatalytic (Supplementary Fig. 14). The autocatalytic nature of the peptide synthesis was confirmed by the observation that the shape of the curve of the **M_pre** residual proportion also became sigmoidal (Fig. 3d) when the amount of added DTT was decreased to reduce the reaction rate. These results indicated that LLPS droplets were formed autocatalytically and were consistent with the hypothesis that the droplets themselves served as sites of peptide generation.

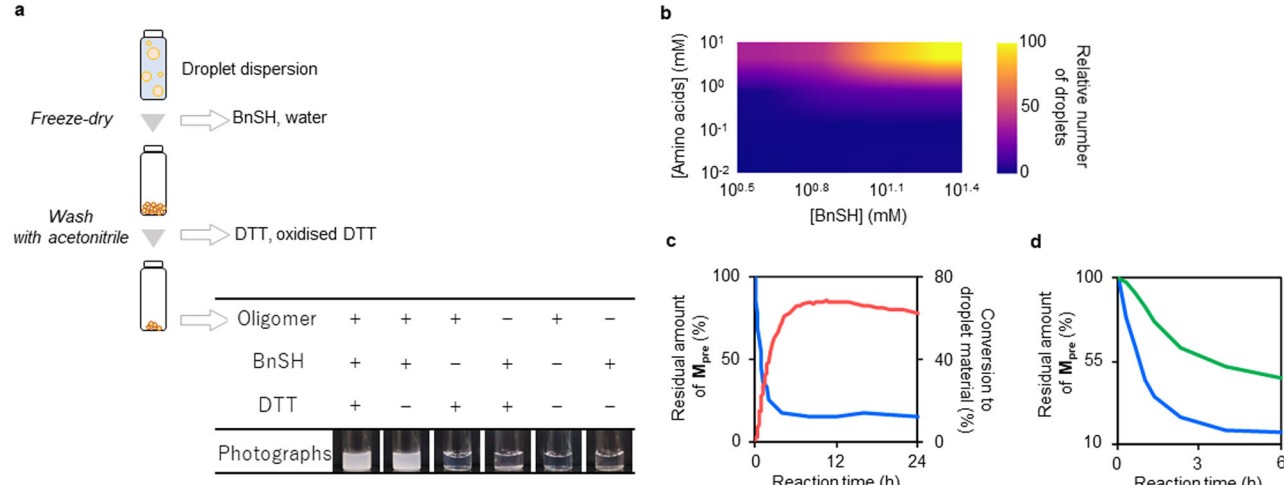

**Fig. 3 Formation of molecular assemblies induced by oligomerisation. a** Schematic illustration of the separation of oligopeptides from the reaction solution. The table shows the droplet-forming abilities of the obtained oligopeptides with and without benzyl mercaptan (BnSH) and DTT; +: addition of corresponding reagent, −: no addition of the corresponding reagent. **b** Heat map of the average number of counted droplets. Only droplets with a diameter larger than 1 μm were counted in the 290 × 217 μm microscopic images (sample number $n = 3$) of each dispersion. The average number of counted droplets was heat-mapped after Gaussian fitting. The average number of droplets was normalised to 100 with [Amino acids] = 10 mM and [BnSH] = 25 mM. Photographs indicating the dependence of droplet formation on the ratio of the concentrations of BnSH and oligopeptides. The concentrations of the oligopeptides were expressed in terms of amino acids. Each specimen was prepared by adding BnSH and deionised water to the peptide. **c** Relative amount of $M_{pre}$ (blue line) in a 24 h interval after mixing $M_{pre}$ (10 mM) and DTT (25 mM). The amount of $M_{pre}$ at 0 h was set to 100. The relative number of droplets (red line) in a 24 h interval after mixing $M_{pre}$ (10 mM) and DTT (25 mM). The number of droplets formed at 0 h was set to zero. The expanded image of the curve for the conversion to droplet material is shown in Supplementary Fig. 14. **d** Relative amounts of $M_{pre}$ versus time for 6 h after mixing $M_{pre}$ (10 mM) with 25 mM (blue line) and 13 mM (green line) DTT. The amount of $M_{pre}$ at 0 h was set to 100. Source Data of Fig. 3b–d are provided in the Source Data File.

**Recursive self-reproduction of LLPS-formed droplets**. To demonstrate the continuous growth of the droplets upon serial additions of $M_{pre}$ and DTT, we measured changes in the size distribution of the droplets that formed after each addition. Figure 4a shows the possible scenarios of size distributions of LLPS-formed droplets after repeated additions of $M_{pre}$ and DTT. After the first addition of $M_{pre}$, the droplet size distribution was expected to shift fully to the right as the droplets grew. This expectation was confirmed by the continuous increase in the size of the LLPS-formed droplets revealed by droplet size analysis (Fig. 4b). This result indicated that nanometre-sized molecular aggregates were formed during the first five minutes after mixing of the $M_{pre}$ and DTT. After 5 min, they grew or fused to become large enough to be observed microscopically.

Upon subsequent addition of $M_{pre}$ and DTT into the dispersion containing the LLPS-formed droplets, the droplet size distribution was expected to change into one of two patterns, depending on the region of oligomerisation in the droplets (Fig. 4a, right). Two possible cases were considered. First, if new droplets formed spontaneously in the solution as oligomerisation proceeded, a new peak at a smaller size would appear in the corresponding distribution (Fig. 4a, upper right). Second, if oligomerisation occurred inside or at the interface of the LLPS-formed droplets, the pre-existing LLPS-formed droplets would enlarge, and no new LLPS-formed droplets would be generated because no oligopeptides would be available in the solution. No separate peak would therefore appear in the size distribution (Fig. 4a, lower right). In the second case, some oligopeptides would also be generated outside and then incorporated into the existing LLPS-formed droplets. To identify the site of oligomerisation, we added equal volumes of $M_{pre}$ and DTT to 1 mL of LLPS-formed droplet dispersion 24 h after the first addition of $M_{pre}$ (10 mM) and DTT (25 mM), and we then monitored the temporal evolution of the size distribution of the LLPS-formed

droplets (Fig. 4c). The sizes of the existing LLPS-formed droplets increased with time, and no additional peak corresponding to new LLPS-formed droplets was detected. Accuracy of the DLS population measurements in the current study was assured by a control experiment with a mixture of extruded dispersions (Supplementary Fig. 15a, b). These results strongly supported the hypothesis that oligomerisation occurred inside or at the interface of the LLPS-formed droplets: that is, these findings pointed to autocatalytic self-reproduction of the LLPS-formed droplets due to the ability of LLPS-formed droplets to serve as active sites for oligopeptide generation.

The LLPS-formed droplets in the current study self-reproduced recursively while they were continuously nourished by consumption of $M_{pre}$ and were extruded as a means of periodic dilution to induce shearing (Fig. 5a). In particular, the LLPS-formed droplets grew and fused by autocatalytic self-reproduction and were then divided by extrusion upon addition of $M_{pre}$ and DTT (Fig. 5b and Supplementary Fig. 16). To quantitatively evaluate the recursive growth and division of the LLPS-formed droplets, we analysed the temporal evolution of the average diameter of LLPS-formed droplets in the dispersion. We monitored the droplet population over six periods: the initial droplet-formation period and five cycles of $M_{pre}$ addition to the existing, uniformly sized droplets (white triangles in Fig. 5b) and extrusion using a syringe (black triangles in Fig. 5b). During each cycle, we observed an increase in the size of the LLPS-formed droplets stimulated by the addition of $M_{pre}$ and DTT that was followed by a decrease in size upon extrusion. From the second to the sixth period, the particle size at the beginning and end of a cycle was almost the same, and the mode of particle size development also remained approximately unchanged. Analysis of the time series of droplet size in Fig. 5b revealed a significant correlation that exceeded the 95% confidence interval (light-blue zone in Supplementary Fig. 17a) at a lag of 33 (the number of measurement points per cycle),

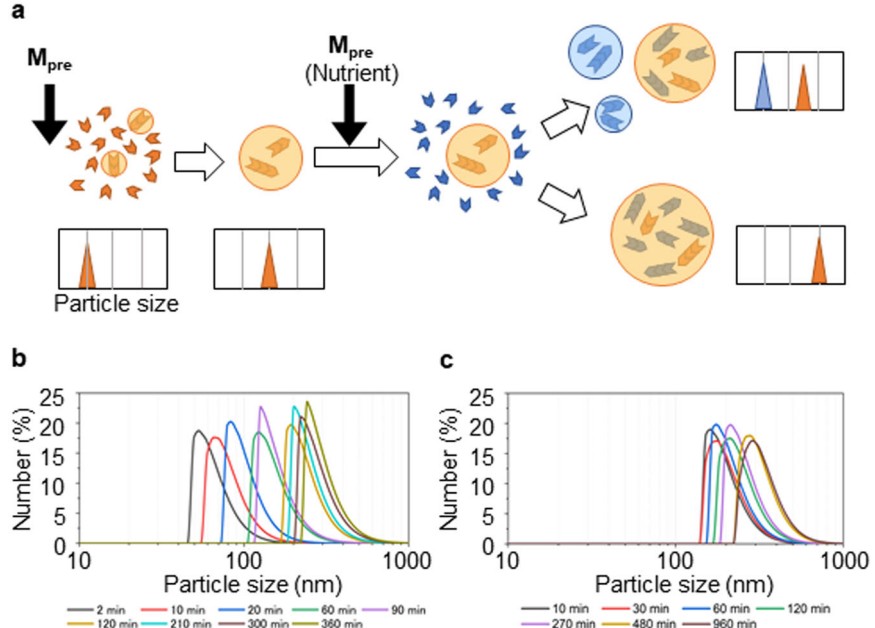

**Fig. 4 Measurement of the sizes of self-reproducing LLPS-formed droplets. a** Schematic particle size distributions after the first and second addition of $M_{pre}$ (nutrient) and DTT. After the first addition, the particle size increased, and the corresponding distribution shifted to the right. After the second addition, if the LLPS-formed droplets did not self-reproduce, a new peak centred at a particle size smaller than that of the original population appeared (upper right). However, if the LLPS-formed droplets self-reproduced, only the original population would be detected, and the corresponding peak would shift to the right (lower right). **b** Particle size distributions measured at different times during the interval between 10 min and 24 h after addition of $M_{pre}$ (10 mM) and DTT (25 mM) to water. **c** Particle size distributions of the solution after the second addition of $M_{pre}$ and DTT 24 h after the first addition (the same amounts of $M_{pre}$ and DTT were added to the solution). Only one population was detected after the second addition. For both **b**, **c**, the x-axis refers to the particle size, and the y-axis to the percentage of droplets of that size to the droplet population. Source Data of Fig. 4b, c are provided in the Source Data File.

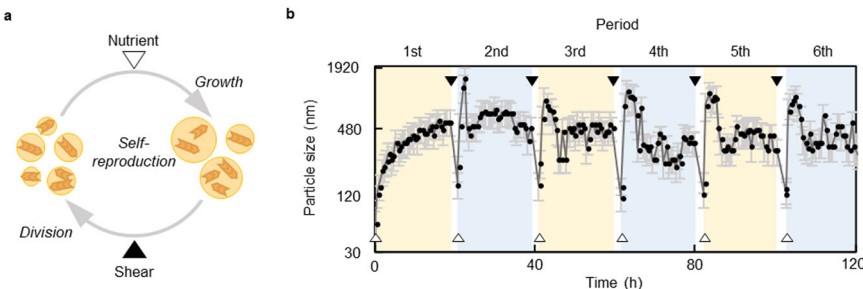

**Fig. 5 Growth-division cycle of self-reproducing LLPS-formed droplets. a** Growth-division cycle of self-reproducing LLPS-formed droplets. LLPS-formed droplets enlarged upon the addition of $M_{pre}$ (white triangles) and subsequently divided in response to an applied stimulus (shear, black triangles). **b** Temporal evolution of average particle size upon repeated nutrient addition and stimulus cycles. Twenty hours after the addition of nutrient, the dispersion was diluted with an equal amount of water and extruded; immediately after extrusion, the same amount of nutrient originally used was added again. These operations were repeated five times. White triangles, the addition of $M_{pre}$ and DTT; black triangles, extrusion. Data are presented as mean values +/− SD. Source Data is provided in the Source Data File.

which corresponded to the time immediately after addition of $M_{pre}$ and DTT. This result clearly indicated that there was a high autocorrelation between particle size changes in every cycle. The use of DIC microscopy revealed similar recursive patterns of LLPS-formed droplet diameters (Supplementary Fig. 17b, c). Droplets with small diameters (< ca. 100 nm) could disperse and self-reproduce in the dispersion. However, the intensity of turbidity, which is quantified on the basis of scattered light, was small because these droplets were much smaller than the wavelength of visible light. Microscopic observations of the dispersions revealed that the number of droplets increased exponentially with each generation (Supplementary Fig. 18). Careful inspection revealed that proliferation of the droplets was

a two-stage process: self-production and subsequent fusion. The consistency between the diameter of droplets formed 3 h after mixing $M_{pre}$ and DTT (Fig. 5b) and the cube root of the conversion to droplet material (Fig. 3c) (correlation coefficient = 0.96) strongly suggested that the sizes of the parent droplets depended mainly on the initial chemical reaction (Supplementary Fig. 19). In addition, the fact that no significant increase in particle size was observed when only water was added to the extruded droplet dispersion with DIC microscope and DLS particle analyser (Supplementary Fig. 20) indicated that the initial increase in particle size was not due to the fusion of droplets after extrusion but instead was the result of the reaction. However, the fact that the correlation coefficient between the two experiments (Supplementary Fig. 14 and

Fig. 5b) more than 3 h after mixing was only 0.068 suggested that the generated droplets grew primarily by fusion after the conversion to droplet material reached plateau. These results imply that more than 3 h after mixing, i.e., at least when the number density of droplets is small, the contribution of fusion to the increase in particle size is negligible. The fact that the sizes of the droplets at a steady stat`e were almost the same during every period, therefore, meant that, despite the effect of dilution, the number density of droplets was kept constant by the addition of precursors. These results demonstrated that LLPS-formed droplets underwent a recursive growth-division process, that is proliferation, in response to the external stimuli of nutrient ($M_{pre}$) addition and extrusion.

**Nucleic acid/lipid concentrations in droplets.** The origins of life may have involved a simple, prebiotic polymer[38,39] that provided a scaffold for the formation of peptide droplets before the synthesis of the common, major components of current organisms, i.e., nucleic acids, lipids, and proteins. However, to evolve into the ancestors of all modern organisms, the proliferation of peptide-CDs would have required cooperation with these major components[33,40,41]. The absence of such a CD up to the present time has been a gap in the logical sequence of events envisioned by all of the three major scenarios—the "RNA world"[42], "lipid world"[43], and "protein world"[44]—each of which envisions that a self-reproducing system of the corresponding molecules evolved into a proliferating protocell via interactions with other molecules. We, therefore, tested the ability of the droplets created in this study to serve as active sites for the incorporation and concentration of fluorescently tagged nucleic acids and lipids into a droplet (Fig. 6a). Twenty-four hours after mixing $M_{pre}$ and DTT, 6-carboxy-tetramethylrhodamine (TAMRA)-tagged RNA (TAMRA-RNA) and boron-dipyrromethene (BODIPY)-tagged hexadecanoyl-*sn*-glycero-3-phosphocholine (BODIPY-HPC) solutions were added to the droplet dispersion. The droplets were then observed with a confocal laser scanning (CLS) microscope. We observed no emission of fluorescence from the droplets for 2 min after the addition of the RNA and lipid solutions. Thirty minutes after addition, however, the molecular assembly gradually began to emit fluorescence derived from TAMRA and BODIPY, and that fluorescence continued for 360 min (Fig. 6b). Line profiles of the fluorescence intensity (Fig. 6c) were generated from the image of each fluorescence channel of the LLPS-formed droplets (Fig. 6b, bottom row). The time-series of the line profiles confirmed the gradual incorporation of RNA oligomers and phospholipids into the LLPS-formed droplets after simultaneous addition of these molecules (Fig. 6c, d). In the case of the RNA oligomers, the maximum fluorescence was detected near the inner boundaries of the droplets. However, the fluorescence peaked at the centre of the droplets in the case of the phospholipids.

A comparison of the CLS microscopy images of the LLPS-formed droplets during independent incorporation of the RNA oligomers or lipids (Fig. 7a, b) with the corresponding line profiles revealed positions of fluorescence intensity peaks similar to the peak positions when TAMRA-RNA (Fig. 7c) or BODIPY-HPC (Fig. 7d) was added separately to the droplet dispersions. To show that the result of this experiment was independent of the particular fluorophore itself, we confirmed that the results were similar for experiments that involved labelling with different fluorophores. Similar distributions of fluorescence intensity were detected for RNA, phospholipids, and DNA labelled with alternative fluorescent probes. Fluorescence derived from fluorescein isothiocyanate-tagged RNA (FITC-RNA) or DNA and TAMRA-tagged DNA (TAMRA-DNA) were detected along the inner boundary of the droplet (Supplementary Fig. 21a–c), and the fluorescence of Texas Red-tagged phospholipids (Texas

Red–DHPE) was detected around the centre of the droplet (Supplementary Fig. 21d). The interactions between nucleic acids, lipids, and peptides generated spatial heterogeneity between the RNA and lipids in the same LLPS-formed droplet. These results implied that the droplets were heterogeneous; their physico-chemical characteristics transitioned from a hydrophilic boundary to a hydrophobic centre that were composed mainly of oligopeptides and BnSH, respectively. Raman microspectrometry revealed that the hydrophobic central part of the droplets was characterised by a relatively large Bn/water ratio and the hydrophilic peripheral part by a relatively small Bn/water ratio (Supplementary Fig. 22a, b).

No significant decrease of droplet size was observed upon addition of RNA and lipid (Fig. 6b−d) or of RNA (Fig. 7c, e); however, when only phospholipids were added, their surfactant nature resulted in a reduction in the size of the droplets (Fig. 7d, f). This result suggested that the incorporation of nucleic acids enabled the proliferating droplet to maintain its size, despite the perturbation associated with lipid addition. To confirm that the ability of the droplets to incorporate nucleic acids was robust and ubiquitous, we mixed the droplet dispersion with a TAMRA-RNA solution and a BODIPY-HPC dispersion or with RNase-free water, followed by 24 h incubation and fluorescence-activated cell sorting (FACS) analysis. Population analysis of the FACS data after addition of either fluorescently labelled RNA or fluorescently labelled lipids indicated that the fluorescence intensity, which corresponded to the amount of RNA or lipids incorporated by the droplet, increased in both cases and that the width of the forward-scattering pulse, which corresponded to the droplet size, decreased only when lipid was added (Supplementary Fig. 23). These results strongly suggested that RNA suppressed any reduction of particle size upon lipid addition. Collectively, these results indicated that the localisation of RNA oligomers near the interface of the droplets contributed to the stability of the droplets. Raman spectroscopy revealed that the hydrophobicity was higher in the centre of the droplet than in the periphery of the droplet that contained the non-fluorescently labelled nucleic acids and lipids (Supplementary Fig. 24a–j). The implication was that, unlike the water inside the droplets without added biomolecules (Supplementary Fig. 22), the water inside the droplets with added biomolecules was replaced by hydrophilic DNA and amphiphilic phospholipids. The heterogeneity of the internal structure of the droplet stabilised the droplet. The nucleic acid localised in the peripheral part of the droplet contributed to the undercoat. The possibility that this localisation was caused by size exclusion as well as by the hydrophilic/hydrophobic balance of molecules within the droplets cannot be dismissed.

To evaluate the effects of oligopeptides on the incorporation and concentration of hydrophilic RNA oligomers and amphiphilic (rather than hydrophobic) phospholipids into BnSH phases that could be among the main components of the droplet centre, we measured the decrease in the fluorescence intensities of aqueous TAMRA-RNA or BODIPY-HPC solutions layered for 24 h over the BnSH solution via high-sensitivity fluorescence spectroscopy with photon-counting detectors (Supplementary Fig. 25a). The fluorescence intensity in the aqueous phase decreased in both the TAMRA-RNA solution and BODIPY-HPC solution, but there was a difference between the two in the magnitude of the decrease of fluorescence in the presence of the oligopeptides (Supplementary Fig. 25b, c). In the case of the TAMRA-RNA solution, the fluorescence intensity in the presence of the oligopeptides was less than one-third of the intensity in the absence of the oligopeptides (Supplementary Fig. 25b), whereas in the case of the BODIPY-HPC solution, there was little difference in the fluorescence intensity in the presence or absence of the oligopeptides (Supplementary Fig. 25c). A similar result was obtained when the fluorescence

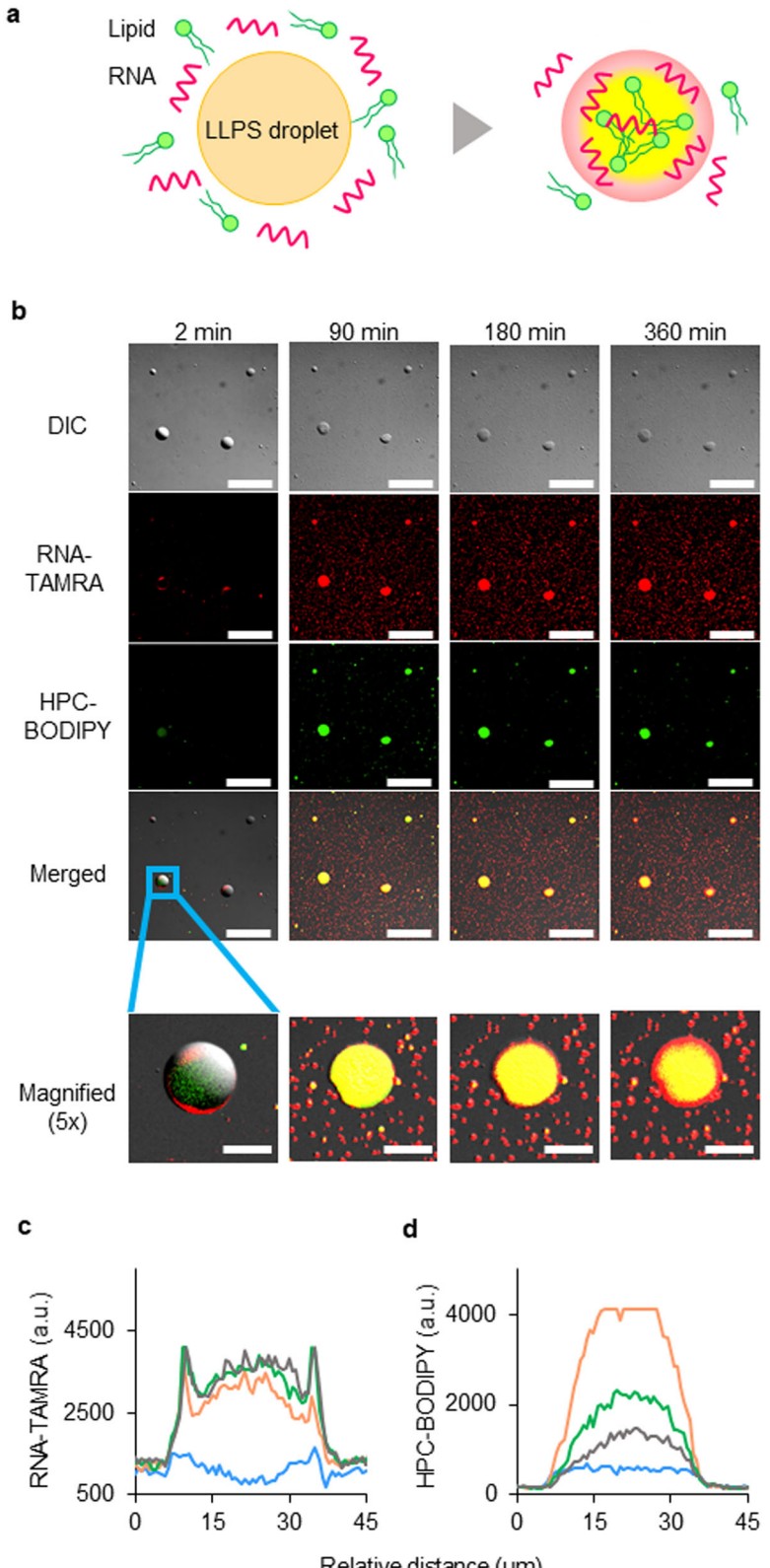

moiety was changed (Supplementary Fig. 25d, e). These results indicated that RNA oligomers and phospholipids were not only incorporated but also concentrated in the BnSH phases, and that oligopeptides could further enhance the RNA enrichment of the droplets because of the high permeability of the membraneless structure of the LLPS-formed droplets. Nucleic acids are known to be incorporated into relatively hydrophobic regions by the formation of complexes with lipids due to electrostatic interactions[45,46]. In the current study, complex formation due to intermolecular interactions between RNA and lipids and/or peptides would have promoted the incorporation of RNA oligomers and phospholipids into the BnSH phase.

The LLPS-formed droplets that we produced under mild conditions were capable of thioester reaction-induced

**Fig. 6 Incorporation, concentration, and separation of biological molecules inside LLPS-formed droplets by internal structural heterogeneity.** Nucleic acids and lipids that were concentrated in the LLPS-formed droplets exhibited higher-order functions of the droplets. **a** Due to the spatial heterogeneity of a LLPS-formed droplet, the RNA and lipids were concentrated and spatially separated, and higher-order properties that had not been exhibited by the droplet itself were apparent. **b** Confocal laser scanning (CLS) fluorescence microscopy images of the LLPS-formed droplet dispersion at 2, 90, 180, and 360 min after addition of the TAMRA-RNA (10 μM) and BODIPY-HPC (10 μM) solutions; both the individual and merged signals are shown (DIC = differential interference contrast microscopy image). The scale bars of the images in the top four rows are 100 μm, whereas those in the bottom row (magnified) images represent 20 μm. The blue rectangle in the photo in the merged image is a magnification of the image below. The result was verified by five trials. **c, d** Temporal evolution of line profiles of fluorescence intensities along a horizontal line through the centre of the LLPS-formed droplet after addition of TAMRA-RNA and BODIPY-HPC: blue line, 2 min; yellow line, 90 min; green line, 180 min; black line, 360 min. **c** Line profiles of TAMRA-RNA fluorescence intensity of the LLPS-formed droplet shown in the bottom row of (**b**). **d** Line profiles of BODIPY-HPC fluorescence intensity of the LLPS-formed droplet shown in the bottom row of (**b**). RNase-free water was used as a substitute for deionised water. Source Data of Fig. 6c, d are provided in the Source Data File.

compartmentalisation, autocatalytic self-reproduction, a steady growth-division cycle associated with extrusion/shear, macromolecular enrichment, and robustness. Peptide formation and accompanying droplet formation are developments that are necessary but not sufficient to ensure that a droplet can proliferate because self-reproduction—the production of components of the droplet by the droplet itself—is also indispensable. In the case of the current LLPS-formed droplets, proliferation was achieved by physicochemical interactions without specific molecular recognition.

A physical autocatalytic reaction, which is induced by an equilibrium shift due to the incorporation of products into formed molecular aggregates, is one kind of autocatalytic reaction[16–19]. More important for the realisation of a self-reproducing system that can generate protocells is a concentration-induced autocatalysis (CiA): an autocatalytic reaction made possible by the molecular assemblies of substrates and catalysts formed by the reaction. Examples include the self-reproduced liposomes reported in previous studies[20,21]. A CiA system is a self-reproducing system because the components of the molecular assembly are generated in the molecular assemblage. In the system described here, CiA could occur because of the gradual accumulation of monomers with a hydrophilic cysteine moiety and a hydrophobic benzyl mercaptan moiety into droplets with an amphiphilic gradient and a membraneless structure. Although autocatalytic reactions that lead to self-replication (i.e., reactions that lead to the construction of an identical copy of the system) of RNA or peptides generally require specific complementary molecular recognition, CiA can be induced by non-specific intermolecular interactions, but only to the extent of phase separation. For this reason, self-reproduction of prebiotic polymers by CiA based on molecular phase behaviour must have played as important a role as LLPS in the emergence of a protocell that was a simple unicellular organism in a primitive environment where intermolecular interactions were inefficient enough to cause autocatalytic reactions based on specific molecular recognition.

The proliferation of the LLPS-formed droplets was consistent with the thioester world scenario and demonstrated that peptide formation via thioester reactions could have facilitated the formation of CD-based protocells on primitive Earth. We, therefore, considered the viability of the molecules used in the current study in a prebiotic environment. Monomer **M** was a thioester of cysteine and BnSH. In a prebiotic environment, thioesters can be synthesised at near-neutral pH through metal oxidation[47]. Moreover, thioesters have been proposed to function as energy currency in primitive metabolism because they can receive energy from the electron transport system and deliver it to ADP for ATP synthesis independent of the membrane structure in the current electron transport system[48]. They may also have contributed to the self-production of primitive cells. The mechanism responsible for the prebiotic syntheses of cysteine and cystine proposed by Sagan et al.[49] based on experiments conducted under more-or-less prebiotic conditions in a reductive

environment are currently viewed with scepticism, but the pathways that led to their prebiotic synthesis are currently unknown[50]. BnSH is a model hydrophobic thiol. For example, an alkylthiol that is capable of forming droplets due to hydrophobic interactions would be a good prebiotic candidate for a hydrophobic thiol[51]. In contrast to the environment of primitive Earth, oxygen was abundant under our experimental conditions. The precursor $M_{pre}$, which was a cross-linkage of two **M** monomers, was therefore reduced with the model reductant DTT to provide **M** in water. A plausible scenario for the prebiotic synthesis of components is explained in Supplementary Note 1. In this system, phase separation eventually occurred, and the molecules were enriched even at low concentrations, although the reaction time was longer. The concentration in this system thus affects only the rate of the chemical reaction. Dry–wet cycles must have facilitated the enrichment process at low concentrations in a prebiotic environment[52]. This system could therefore be regarded as a model system that mimicked the emergence and proliferation of protocells under the following prebiotic scenario.

Hydrophobic thiols and amino acids, including cysteine, were produced in an organic soup at high temperature, high pressure, and under either highly acidic or very basic conditions in a geyser. Organic soups emanating from geysers formed ponds on Earth and yielded thioesters by metal oxidation on the surfaces of minerals[53]. The thioester monomers polymerised spontaneously, and the products of polymerisation underwent phase separation to form droplets. With the intermittent flow of organic soup from the geyser, the droplets would have proliferated via dilution, shearing, and incorporation of nutrients. As mentioned above, primitive cells would have flourished around the geyser, and thioesters would have been the principal metabolic agent.

These proliferating droplets composed of peptides could have served as containers to integrate RNA, lipids, and peptides during the early history of Earth because the droplets not only incorporated nucleic acids and lipids but also acquired the ability to survive by accelerating interactions among these constituents such as expression of the homeostasis of particle size. Droplets such as those used in this study must have concentrated various substrates and formed a hydrophobic reaction field, which may have contributed to the formation of reaction networks as well as the synthesis of lipids, nucleic acids, and peptides. The nucleic acids in such reaction networks may have provided a useful roadmap that led to the emergence of an information system. The concentrations of nucleic acids and lipids formed by the mild amphiphilic gradient in droplets facilitated the performance of physicochemical roles by the nucleic acids. The concentrated RNA protected the droplets from dissolution by lipids because the hydrophilic RNA was localised near the inner interface of the droplet by the amphiphilic gradient.

In addition, the nucleic acids that accumulated inside the assembled droplet could have concentrated other potential

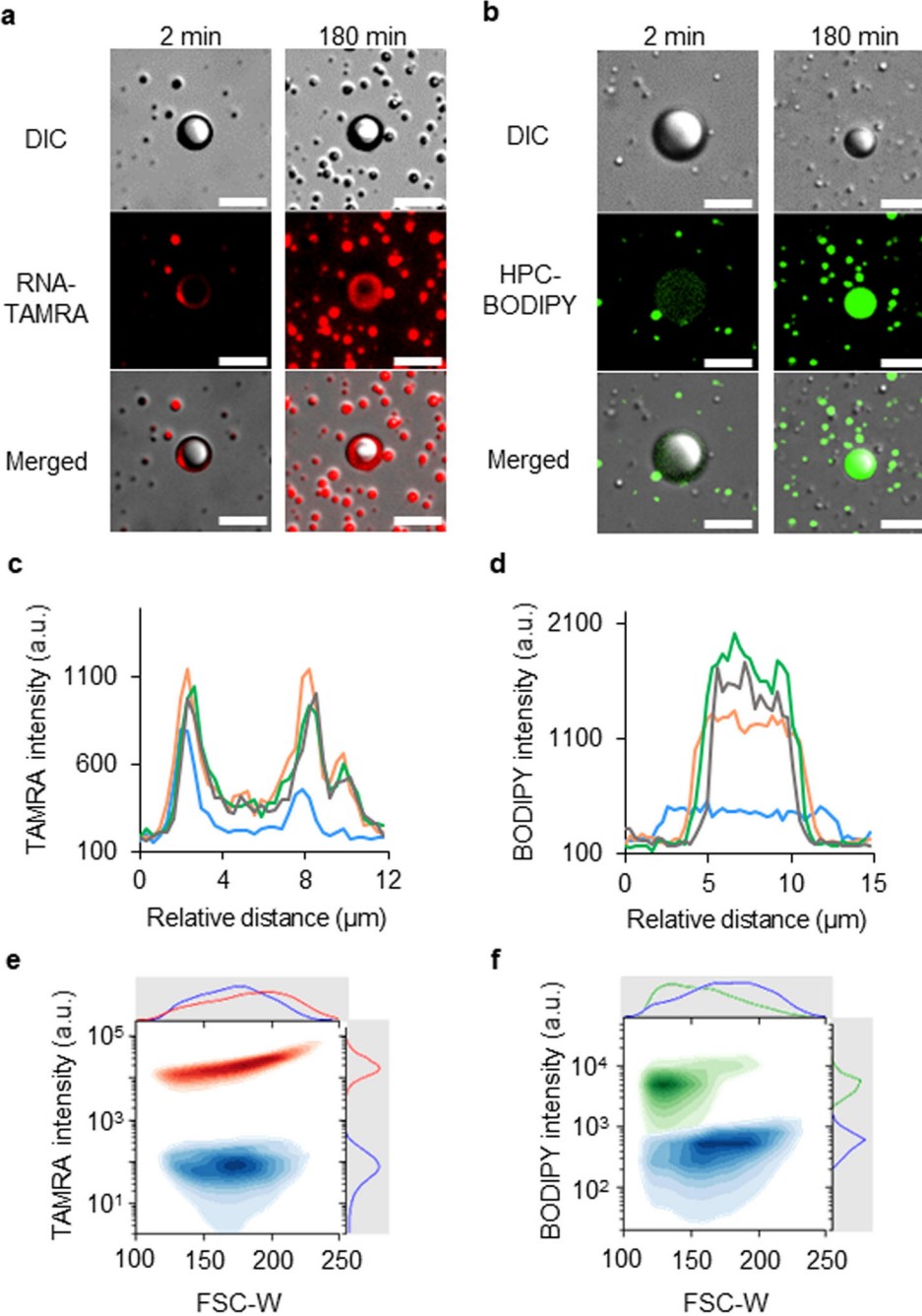

**Fig. 7 Concentrated nucleic acids or lipids in LLPS-formed droplets. a** CLS fluorescence microscopy images of the dispersion 2 and 180 min after addition of 1 μM TAMRA-RNA solution. The result was verified by five trials. **b** CLS fluorescence microscopy images of the dispersion 2 and 180 min after addition of 1 μM BODIPY-HPC solution. The result was verified by five trials. **c**, **d** Temporal evolution of line profiles of fluorescence intensities along a horizontal line through the centre of an LLPS-formed droplet after TAMRA-RNA or BODIPY-HPC addition: blue line, 2 min; yellow line, 90 min; green line, 180 min; black line, 360 min. **c** Line profiles of TAMRA-RNA fluorescence intensity of the LLPS-formed droplet shown in the bottom row of (**a**). **d** Line profiles of BODIPY-HPC fluorescence intensity of the LLPS-formed droplet shown in the bottom row of (**b**). **e** Bivariate kernel density estimation (KDE) analysis applied to the distributions of TAMRA fluorescence intensity and pulse width of forward scatter (FSC-W, corresponding to the particle size) of droplets before and after the addition of TAMRA-DNA. Histograms and scatter plots derived from each droplet before (blue) and after (red) the addition of TAMRA-DNA are shown. The curve on the x-axis shows the histogram of forward scattering intensity, and that on the y-axis shows the histogram of TAMRA fluorescence intensity. **f** Bivariate KDE of the distributions of BODIPY intensity and FSC-W of droplets before and after the addition of BODIPY-HPC. Darker colours represent higher densities. The univariate KDEs for both variables are also shown in the plot. Histograms and scatter plots derived from each droplet before (blue) and after (green) the addition of BODIPY-HPC are shown. The curve on the x-axis shows the histogram of forward scattering intensity, and that on the y-axis shows the histogram of BODIPY fluorescence intensity. The scale bars represent 20 μm in the CLS microscopy images. The numbers of the measured droplets in FCM experiments were 10,000. RNase-free water was used as a substitute for deionised water. Source Data of Fig. 7c−f are provided in the Source Data File.

substrates and catalytic molecules in a process called hyperconcentration. LLPS could demonstrate the heterogeneous localisation of nucleic acids and lipids forming a cell-like structure[54]. Nucleic acids could have functioned as information carriers to control the self-reproduction process. Self-reproduction induced by hyperconcentration has been reported in vesicles[45,46,55]. If the nucleic acids concentrated within the present droplets altered the self-reproduction behaviour of the droplets by collecting molecules around the droplets, it is likely that the droplet functioned as a carrier of information that affected the rates of survival and proliferation. Because the nucleic acids were distributed to newly formed droplets, droplets enriched with the nucleic acids and other molecules would have acquired genetic information via hyperconcentration. LLPS-formed droplets carrying genetic information could have generated primitive living things by combining proliferation with advanced phenomena[56] such as self-propulsion[57,58], droplet–droplet communication[11], and competition[12]. To construct more life-like droplets such as active/dissipative droplets[59–61], the internal products must be released through interactive reactions of these biomolecules through such dynamics.

In summary, because the process of evolution from amino acid thioesters to primitive living things could be realised by the concentration of RNA, lipids, and peptides inside a proliferating droplet and a subsequent expression of a biological-like function, it seems appropriate to call this scenario the "droplet world hypothesis". Various life-like functions can be imparted to a droplet by inserting alternative amino acids or peptides between cysteine and the thioester moiety in the current monomer or by using other alkylthiols as leaving groups. Interestingly, non-ribosomal peptide synthesis using a similar mechanism has been discovered in some bacteria and eukaryotic cells[62,63]. In these in vivo peptide syntheses, amino acid thioesters function as monomers to form peptides. Droplets that are composed of peptides and nucleic acids and that are formed inside a cell can serve as sites of reactions related to gene expression in modern cells[64,65]. These results are consistent with the scenario that the protocell was based on CDs formed by thioester reactions. Furthermore, because the droplet world hypothesis was derived from model experiments, a corollary of the hypothesis is that a protocell may have emerged by CiA-polymerisation of more primitive monomers[38,39] than amino acid thioesters. The system proposed in this study is therefore a very powerful platform not only for verifying the ancient droplet world scenario of the origins of life but also for developing self-sustainable materials that mimic superior forms of life.

## Methods

**Synthesis of 3,3′-disulfanediylbis(1-(benzylthio)-1-oxopropan-2-ammonium) dichloride ((Cys-SBn)₂•2HCl, M_pre)**. N,N′-Di(tert-butoxycarbonyl)-L-cystine [(Boc-Cys-OH)₂] (4.4 g, 10 mmol) was activated by mixing N,N′dicyclohexylcarbodiimide (DCC, 4.5 g, 2.2 eq.), 1-hydroxybenzotriazole (HOBt, 3.4 g, 2.2 eq.) and N,N-diisopropylethylamine (DIEA, 7.7 mL, 6.6 eq.) in dichloromethane. The solution was stirred for 10 min at 0 °C. After addition of benzyl mercaptan (BnSH, 5.2 mL, 6.6 eq.) to the solution, the mixture was stirred for 12 h at 25 °C. The reaction solution was filtered and evaporated under reduced pressure. The obtained crude product was purified by medium-pressure liquid chromatography (hexane/ethyl acetate [4/1, v/v]) and then high-performance liquid chromatography (HPLC) (chloroform) to obtain a clear oil product (Boc-Cys-SBn)₂ in 48% yield. The purified product was identified as (Boc-Cys-SBn)₂ based on its $^1$H NMR spectrum (Supplementary Fig. 2) and electrospray ionisation time-of-flight mass spectrometry (ESI-TOF MS) spectra. The obtained (Boc-Cys-SBn)₂ (653 mg, 1 mmol) was added to 5 mL of 4N HCl/ethyl acetate and stirred at 25 °C for 2 h. After filtration, the residue was washed with ethyl acetate and then dried to obtain the product as a white powder in 92% yield. The product was identified as (Cys-SBn)₂•2HCl (M_pre) via $^1$H NMR (Supplementary Fig. 3), $^{13}$C NMR, and ESI-TOF MS measurements. $^1$H NMR (400 MHz, DMSO-d₆) δ = 8.87(6H, s), 7.50–7.00 (10H, m), 4.70–4.48 (2H, brd), 4.27 (4H, s), 3.27–3.00 (4H, m). $^{13}$C NMR (400 MHz, DMSO-d₆) δ = 194.59, 136.52, 128.78, 128.48, 127.35, 99.37, 57.55, 32.69, 32.49. ESI-TOF-MS(MeOH) m/z = 453.0799 (the [M + H] $^+C_{20}H_{25}N_2O_2S_4$ peak appeared at m/z = 453.0793).

**Synthesis of 2-(benzylthio)-2-oxoethan-1-ammonium chloride [(Gly-SBn)•HCl]**. N′-(tert-butoxycarbonyl)-L-glycine (Boc-Gly-OH, 876 mg, 5 mmol) was activated by mixing N,N′-dicyclohexylcarbodiimide (DCC, 1135 mg, 1.1 eq.), 1-hydroxybenzotriazole (HOBt, 743 mg, 1.1 eq.) and N,N-diisopropylethylamine (DIEA, 1.9 mL, 3.3 eq.) in dichloromethane. The solution was stirred for 10 min at 0 °C. After the addition of benzyl mercaptan (BnSH, 1.3 mL, 3.3 eq.) to the solution, the mixture was stirred for 12 h at 25 °C. The reaction solution was filtered and evaporated under reduced pressure. The obtained crude product was purified by medium-pressure liquid chromatography (hexane/ethyl acetate [19/1, v/v]), and then HPLC (chloroform) to obtain a clear oil product Boc-Gly-SBn. The purified product was identified as Boc-Gly-SBn via the spectra of $^1$H NMR and ESI-TOF-MS. The obtained Boc-Gly-SBn (703 mg, 2.5 mmol) was added to 6 mL of 4N HCl/ethyl acetate and stirred at 25 °C for 2 h. After filtration, the residue was washed with ethyl acetate and then dried to obtain the product as a white powder in 87% yield. The product was identified as [(Gly-SBn)•HCl] via $^1$H NMR measurement and ESI-TOF MS measurement. $^1$H NMR (400 MHz, CD₃OD) δ = 7.40–7.20(5H, m), 4.27 (2H, s), 4.10 (4H, s). ESI-TOF-MS (MeOH) m/z = 182.0654 (the [M + H] $^+C_9H_{12}NOS$ peak appeared at m/z = 182.0634). The NMR spectrum is shown in Supplementary Fig. 5.

**Microscopic observations**. M_pre (5 mg, 10 mmol) and/or the reductant DTT (4 mg, 25 mmol) were added to 1 mL of deionised water and shaken for 15 s. Aqueous solutions (25 μL) were placed on a glass plate, and the plate was then immediately covered using a Frame-Seal™ incubation chamber (9 mm × 9 mm, 25 μL, Bio-Rad Laboratories, Inc.) as a spacer. Bright-field images were obtained with DIC microscopy. Fluorescence microscopic images were obtained with a confocal microscope fitted with a 488 nm excitation laser and a band path (BP) 525/50 nm emission filter, or a 561 nm excitation laser and a BP 609/54 nm emission filter.

**Particle size distribution measurements by dynamic light scattering**. The sizes of droplets were measured at 25 °C with an ELSZ-1000 particle analyser, which is suitable for the measurement of particle sizes from 0.6 nm to 10 μm. A 50 μL sample of aqueous mixtures was placed in a quartz cuvette and equilibrated to 25 °C prior to measurements. The distribution of aggregate sizes was measured every 30 min for 24 h.

**Reaction monitoring via NMR spectroscopy**. M_pre (11 mg, 20 mmol) and DTT (8 mg, 50 mmol) were added to heavy water (2 mL) and vortexed for 15 s at room temperature. The resulting solution was aliquoted into NMR tubes (500 μL) and measured every 10 min for 24 h.

**Reaction-monitoring by fluorescence spectroscopy**. M_pre (11 mg, 20 mmol) and DTT (8 mg, 50 mmol) were added to water (2 mL) in a microtube and then vortexed for 15 s at room temperature. The prepared M_pre solution (20 μL) and fluorescamine solution (90 μL) in super-dehydrated dimethyl sulfoxide (15 mg/2.5 mL) were mixed. Immediately after mixing, 90 μL of phosphate buffer solution was added, and then the microtube was flicked five times gently. The mixed solution was incubated for 30 min, and then 100 μL of the solution was dispensed into a quartz cell and measured via a fluorescence spectrometer. This procedure was conducted at each point in time for monitoring purposes. The rate of consumption of primary amine was calculated from the changes in absorbance at 476 nm and equated to the monomer conversion rate.

**Reporting summary**. Further information on research design is available in the Nature Research Reporting Summary linked to this article.

## Data availability

Source data are provided for Figs. 2c, 3b, c, d, 4b, c, 5b, 6c, d, 7c−f and Supplementary Figs. 10b, 13b, c, 14, 16, 17a, c, 18, 19, 20c, 22b, 24b, d, f, h and j as a Source Data file. The data that support the findings of this study are available from the authors upon reasonable request. Source data are provided with this paper.

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

## Acknowledgements

The authors thank Prof. Taro Toyota of The University of Tokyo for his enlightening comments on this work. This work was financially supported by the Okazaki ORION project of the Exploratory Research Centre on Life and Living Systems (ExCELLS) and by Japan Society for the Promotion of Science Grants-In-Aid for Scientific Research (KAKENHI) (grant numbers JP17H04876, JP15K17850 and 20K20951). Additional financial support was provided by grants from the Yoshida Scholarship Foundation; the Kurita Water and Environment Foundation (15E011, 16E018, 18D007, 19K011); the Oil & Fat Industry Kaikan; Kao Foundation for Arts and Sciences; The Public Foundation of Chubu Science and Technology Centre; The Hori Sciences and Arts Foundation; a Grant-in-aid for the Graduate School of Integrated Science for Life, Hiroshima University; the "Innovation Inspired by Nature" Research Support Programme of Sekisui Chemical Co. Ltd. The Raman microscopy and spectrometry were performed by the Division of Instrumental Analysis, Research Support Centre, Institute for Research

Promotion, Kagoshima University. This work also received technical and official support from the Functional Genomics Facility of the National Institute for Basic Biology Core Research Facilities; the Instrument Centre of the Institute of Molecular Science; and Prof. Satoshi Nakata of Hiroshima University.

## Author contributions

K.K. and M.M. designed the experiments. M.M. performed the experiments, analysed the data, and prepared the draft. K.K. and M.M. wrote the paper.

## Competing interests

The authors declare no competing interests.
