## [Peer Review File · Nature Communications]

Reviewer #1 (Remarks to the Author):

This paper titled “Self-proliferating Liquid-liquid Phase separated Droplet; the missing link between Chemistry and Biology on the Origin of Life” by Matsuo and Kurihara describes coacervate droplet formation by *in-situ* synthesized peptides that can accumulate and distribute RNA and lipid heterogeneously within the droplet combining RNA, lipid and peptide in one droplet. They show self-proliferating growth-division cycles of the droplet. There are some areas of novelty in this work, in particular, in showing that *in-situ* peptide formation can induce droplet formation. However, we found there are a number of places where the experimental data is not sufficient to support all of the assertions made in the paper. Above all, a complete and thorough characterization of the coacervate droplet has not been demonstrated; for example, it is not clear whether the droplets are forming via a charge separation or a hydrophobic interaction. Introduction of several technical terms need additional experimental verification to justify the use of these statements. As it stands, this manuscript would not be acceptable for Nature Communications, however, due to the novelty in some aspects of the work (as described previously), we recommend that this work should only be accepted by Nature communications after major corrections.

Major comments:

1. The style of writing in the abstract and introduction is difficult to understand in places. This is partly due to the grammar where the sentence structure hinders the flow and the meaning of the text. In addition, the authors use terminology such as self-proliferating, self-maintaining, self-reproduction etc. while it is not always clear the reason why these terms are used considering that they do not always correlate with the experimental observations. For example, in this article, droplets were described as self-proliferating. Self-proliferating suggests that the system changes without any external input. To drive division of the droplet, the authors introduce external shearing which is not self-driven proliferation.
2. The authors make an interesting reaction setup and observation regarding the *in-situ* formation of droplets after dithio bond cleavage and peptide bond formation to generate a peptide chain. This is novel work. However, the additional control experiments which show that BnSH is required to drive the droplet formation do not fully describe or explain the mechanism of droplet formation. Throughout the text, the authors describe this as a coacervate droplet and the authors show and give no clear evidence that the droplet is forming by coacervation which typically occurs between two oppositely charged components. A more detailed understanding is required to confirm that droplet formation is driven solely by coacervation as described. This could be achieved by purifying the peptides and characterizing the phase diagrams. Additionally, they could characterize the properties of the droplets, for example, FRAP and/or coalescence properties to determine whether they are indeed coacervate droplets. It would be interesting to see if fluorescently labelled amino acids (or peptides) localize inside the droplets to observe their specific distribution within the droplet. Indeed, the structure of the droplet could help in explaining the RNA and lipid accumulation in the latter part of the manuscript.
3. The observation of the heterogeneous accumulation of the RNA and lipid is interesting however as it stands it is only a phenomenological observation. With some additional experiments i.e. partitioning of different molecules with different hydrophilicity and zeta potential experiments, the authors would be able to gain insights into the structure of the droplet which may also help to provide an understanding of how the droplet forms and what are the critical components for droplet formation is (see point 2). For the RNA and lipid localization experiment, control experiments showing the localization of the dyes alone would be required to determine whether the dye plays a role in the heterogeneous distribution of RNA/lipid. Is there a possibility that the heterogeneous distribution of the RNA and lipid could be driven by a size exclusion effect of RNA rather than a hydrophobic/hydrophilic effect as has been observed with other polymer systems?
4. We recommend clear labelling of the figure legend i.e. identification or marking the curves in the inset of the figure and full experimental descriptions. Labelling of graph axis should be consistent to avoid misinterpretation of the data and axis ticks should be either linear or log scale, please see figure 4b and 4c x-axis and 4e y-axis which needs to be amended. General figure presentations could be improved, for example, page number 7, figure 2c – the data curves are too thick and the black line and error bars cannot be seen at this resolution.

5. Mass spectra (not just the values) should be included and the peptide multimers should be quantified. It would also be interesting to see the change in the peptide length by mass spectrometry during consequent reactant addition-shearing cycle to clearly understand the effect of periodic additions on peptide length and quantity with a correlation to the droplet size.
6. On Page 4 line 2 the authors state that: "We also showed that self-maintenance ability of the LLPS droplet against dissolution by lipids and co-existence between nucleic acids and lipids in the same LLPS droplet, depending on the concentrated nucleic acids localization into the inner boundary of the LLPS droplet with the assistance of generated peptides." This is a big statement and when considering the data, it is not clear how the droplets are self-maintaining themselves. With reference to point 2 and 3, it would be interesting to identify the co-existence of the layers of two separate phases specifically made of peptide-RNA and BnSH-lipid inside the droplet to see if there is any correlation between the additional components and the size of the droplet.
7. Please describe the method of shearing i.e. what is it and how is it applied. It would be useful to have a control experiment i.e. experiment 4e without adding Mpre to observe any changes in droplet size and number over the time. This would allow one to understand the effect of shearing, alone, and to confirm that the growth of droplets changes due to the addition of precursors. Is the data from figure 4e obtained from DLS data such as that presented in Figure 4c? If so, then it is not clear how the error bars are correlated, for example, the distribution of the data from the DLS is approximately +/- 150 um while the error in the 4e looks circa +/-70 nm. Please clarify this. Including the raw data of Figure 4e into the SI would add additional clarity.
8. Page number 9, line number 15: "When a smaller amount of DTT was added to reduce the reaction rate, the Mpre residual rate curve assumed a sigmoidal shape (Fig. 3c), confirming the autocatalytic nature of the peptide synthesis." - In an autocatalytic reaction, concentration vs time curve shows a sigmoidal shape originated from stochastic delay as well as product enhancement that is also one of the reactants. Here the product is going out of the reaction phase (by LLPS) hence shifting the reaction equilibrium towards the right-hand side and does not fulfil the criteria of autocatalysis. Indeed, the authors do not fit the curve to an autocatalytic model to confirm that the reaction is autocatalytic. To confirm that the reaction is autocatalytic and sigmoidal the authors could lower DTT concentrations and undertake a proper curve fitting to the data. Please also clarify why this reaction would be autocatalytic. Does the size of the droplet increase correspond to an autocatalytic reaction rate?

Minor comments:

9. In page number 2, line number 10, the authors mentioned that "A CD is formed from organic molecules via liquid-liquid phase separation (LLPS), mainly resulting from the spontaneous assembly of oppositely charged molecules or from hydrophobic polymers in general and prebiotic polymers or oligomers on the abiogenesis scenario in particular" – proper referencing is required here especially as coacervation is typically not driven by hydrophobic interactions.
10. The authors state on page number 5, line number 21: "These results indicate nanometre-sized molecular aggregates were formed until 1 hour after mixing the Mpre and DTT, and then they grew or fused to large enough to be observed with a microscope" – If the droplet size is not measurable it is unreasonable to speculate on the droplet size and better to make a clear remark that the droplets were observable by microscopy after 1 hour of the experiment.
11. Is there any difference in the chemical shift of the NMR spectra of the amino acid or peptide signals before and after droplet formation? How would this affect the experimental observations?
12. For Figure 3a the images which show turbid solutions does not confirm that droplets are formed. These images could also show the formation of precipitates, therefore, please verify that the visual turbidity correlates to droplet formation by optical microscopy.
13. On page number 11, line number 9 the authors say: "Of course, some oligopeptides will also be generated outside and then incorporated into the existing LLPS droplets in the latter case. Referencing these two possibilities, in order to identify the actual oligomerisation site, equal volumes of Mpre and DTT were added to 1 mL of LLPS droplet dispersion 24 hours after the first addition of Mpre (10 mM) and DTT (25 mM), and the time evolution of the size distribution of

the LLPS droplets was then monitored (Fig. 4c). The particle size of the existing LLPS droplets increased with time, and no additional peak corresponding to newly formed LLPS droplets was detected. This result strongly supports the view that oligomerization occurs inside or at the interface of the LLPS droplets: that is, these findings point to autocatalytic self-reproduction of the LLPS droplets due to the ability of LLPS droplets to play as active sites for oligopeptide generation.” This result is not as convincing as they say especially as microscopy images shows small droplets (less than 5 μm in solution) which are not shown in the DLS data. Could it be that these smaller droplets are also present in the solution but not detected by the DLS?

14. After shearing, it would be expected that the number of droplets will increase. If this were the case then this would be observable by DLS experiments. In Figure 4e, from the 3rd to 6th period, there is an increase in size of the droplet just after addition and then a decrease and stabilisation of droplet size after time. It is interesting to see that these droplets maintain a steady size after external perturbations which is not typical of coacervate droplets. An explanation or reasoning of this behaviour is required. In Figure 4e, labelling shows first addition and then extrusion while the reverse is described in the article. Moreover, the size of the droplets from the DLS does not correlate to the microscopy data (see point 12). The average size in DLS is much larger than observed in the microscope whilst there is no evidence of the small 5-10 μm sized droplets in the DLS that are clearly seen in the microscopy images.
15. Page number 16 line number 23 and S13: “In the presence of RNA, the fluorescence intensity of the aqueous layer containing the oligopeptides was less than one-third of that of the aqueous layer without the oligopeptides” – It is not clear how the fluorescence intensity of tagged RNA decreases in the presence of the peptide in the aqueous layer, please can the authors explain this observation further.
16. Page number 17, figure 5 shows a nice experimental design, demonstration and representation, although a better explanation of the FACS data is required in the figure legend and in the text. In addition, the microscopy images are quite low resolution and it is difficult to see some of the droplets.
17. On page 3 line 3, the authors state” Very few studies have reported recursive self-reproduction of supramolecular assemblies under periodic stimuli because these metastable assemblies tend towards an equilibrium state”. This statement requires some additional references as there are some studies which have shown this already, for example, recent work by the Spruijt Lab.
18. The authors claim that this is the first example of a coacervate which combines peptide, RNA and lipids in one example. This is not the case as there has been previously published work which shows examples of coacervate droplets with these materials in combination, see work from the Mann Lab. This should be amended and the citations included.
19. Page 3, line 13, “In principle, a CD can self-reproduce only if both the CD formation condition and the peptides representing the CD building blocks reproduction condition are satisfied.” This statement requires some additional clarity. What reproduction condition needs to be satisfied?
20. “The C-terminus of M was capped with a benzyl mercaptan (BnSH) leaving group, which works as a seed droplet to promote self-assembly in terms of hydrophobic interactions.” Please explain the role of BnSH in generating the seed droplet.

Specific comments:

21. The authors used non-conventional abbreviations for TAMRA (TR) and BODIPY (BP). TAMRA and BODIPY are already abbreviations and should not be abbreviated further.
22. Figure 5i and 5j: A clear description of the plots and figure legends are required.
23. Scale bars in microscopy images are not suitable for the size of the droplets.
24. Figure S9: An explanation in the text is required to explain why the droplet population in the two images look similar.
25. Page number 17, line number 6: “Figs. S13b and 8d, respectively.” There is no figure number 8d.

Reviewer #2 (Remarks to the Author):

The manuscript by Matsuo and Kurihara describes an interesting system of active liquid-like droplets that are formed spontaneously upon release of a reactive thioester monomer from a disulfide precursor. The authors demonstrate that their droplets contain an oligomer of the reactive cysteine thioester monomer, and that they are formed more rapidly when more reducing agent is present. The droplets can undergo a so-called self-reproduction cycle by periodically extruding them through a small filter and feeding the extrudate with precursor molecules again. Finally, these droplets are shown to concentrate both fluorescently-labelled RNA and lipid molecules, suggesting that they may have played a role to reconcile the seemingly separate theories for the emergence of living systems based on lipids, nucleic acids and droplets.

Research into protocells has gained significant attention over the past few years, and several groups have reported systems in which active droplets start showing convincing life-like properties. An important barrier is that many of these systems are still based on complex synthetic molecules. The system reported here seems to make a significant contribution to the field of protocells, as it provides a pathway for the spontaneous generation of droplets out of relatively simple precursor molecules. However, in my opinion the system and the experiments presented here are poorly characterized, and convincing proof for many of the bold statements and “grand” properties attributed to these droplets is lacking. This makes it hard to value the true relevance or contribution to the origin of cellular life or even LUCA. If the required characterization and evidence could be provided and all other problems would be addressed, I would certainly be favourable of reconsidering this work for publication in Nature Communications, since the potential significance is quite high. However, in the current form, I do not recommend publication.

My main concerns are the following.

- The reaction described in Figure 2a produces significant amounts of droplets, which show fascinating behaviour in the rest of the manuscript, yet the droplets are not characterized. The authors later speculate that the droplets have a spatial heterogeneity (Figure 5), but without evidence. It seems (page 16, line 16) that the authors assume (and even conclude on page 15, line 19-20) the droplet is made of BnSH, stabilized with an oligopeptide shell, but there is no direct evidence for this. However, knowing the composition of these droplets is in my opinion essential for a proper understanding of the experiments presented here, and the authors should substantiate their assumptions by collecting the droplets by centrifugation and doing LC-MS analysis of the separated phases (supernatant and droplet) with the right controls to determine the droplet composition. It would be more convincing even if to simply reproduce the droplets by mixing these known concentrations of BnSH and oligocysteine (the authors could use for instance the trimer, which they find predominantly).
- These type of reactions and assembly processes are usually quite sensitive to pH, yet pH or control of the pH is not mentioned anywhere in the paper. I would guess that the pH is quite low, but it would be useful to verify this and to report how stable the pH is over the course of the growth and self-reproduction experiments. The authors report a control experiment in which BnSH, cystine dihydrochloride and DTT mixed together create solid precipitates (Fig. S4b), which seems strange. I was not able to reproduce these findings. Is it possible that the cystine was not properly dissolved first before mixing? Was the mixing order relevant? In the system shown in Fig. 2, the DTT first reacts with the disulfide precursor, and BnSH is produced only later. The description in the main text (page 6, line 14) is ambiguous: there was no turbidity, but aggregates were found to precipitate.

- The authors regularly use self-important terminology to describe their system, including autocatalytic, proliferation, self-reproduction, growth-division, higher-order function. Most of these terms have a defined meaning, but evidence that they apply to the current system is lacking. In an autocatalytic system, for example, a product enhances the rate of a process/reaction in which it is being formed. This can be tested by seeding the initial mixture with some of the product, or by fitting a series of kinetic experiments to the appropriate autocatalytic rate equation. The green line in Fig. 3c is hardly convincing evidence of an autocatalytic process. It is difficult to see in the low-quality figure without data markers, but it seems that it has been attributed a sigmoidal shape based on a single data point. Additional kinetic traces are missing and the existing traces are not fitted to show that an autocatalytic reaction gives a significantly better fit than, for example, a simpler first-order decay.
- The same can be said of other terms used by the authors, such as self-reproduction. The conclusion based on DLS size distributions is not straightforward for several reasons. First of all, the larger droplets contribute disproportionately much to the scattered intensity, and it is not unlikely that very small droplets are not detected with the automatic fitting procedure. Did the authors do a control in which they mixed equal amounts of a mixture of a few minutes old and one of several hours, to show that the distinct populations of droplets can be detected? Another concern with this method is the fact that additional processes leading to increase in droplet size also play a role in the background. True liquid phase separated droplets also coarsen over time by ripening and coalescence (accelerated by flow/shaking, see review by Berry et al. 2018), and dispersions settle under the influence of gravity, which is also dependent on droplet size. All these processes may lead to a gradual increase in apparent size over time. Control experiments are missing for a passive system (e.g., without addition of precursor and DTT), and with intermediate homogenization.
- It is not helpful that the corresponding figure (Fig.3) is highly confusing. The x-axis seems to have the wrong scale (these droplets are probably not 500 micrometer large – see Fig.2), and the y-axis displays normalized number, but it is not clear what this is normalized with and how normalization has been done (all traces seem to have different peak heights and areas under the curve).
- The growth-division cycle shown in Fig. 4d-e are also not conclusive. Division suggests that the number of droplets increases over time, but there is no evidence to show this. The trace in Fig. 4e shows a consistent decrease in size about 3-4 data points after each new addition of precursor, but no explanation is given. This seems to be in conflict with the continuous increase in size shown in Fig. 4c (over 16 h!) and it seems to contradict the conclusion that there is growth. Control experiments without feeding are missing and also here the y-axis seems incorrect.
- Finally, the title is overly pretentious. This manuscript shows absolutely no link to biology, everything in here is based on principles of (physical) chemistry.

Minor points

- The paragraph on page 20, starting line 16 is purely speculative and does not serve a clear goal. This manuscript provides no evidence for any of these points, and if this paragraph is retained, the speculative nature should be emphasized.

- All references to works by the groups of Boekhoven, Cronin and Huck, on active/dissipative droplets, their life-like behavior and their potential roles in the origin of life are missing. Some of these would certainly be relevant to put the current work in context.
- The quality of many graphs is very poor. It was impossible to see for example the open squares and crosses (described on page 6, lines 11-13). One of these should have half the turbidity of the mixture solution, but there is no line in Fig. 2c at half the maximum turbidity.
- Fig. 5b-f: something is wrong with the scale bars here. The droplet in 5b is larger than 100 micrometer according to the scale bar, but the cross sections in 5c,e are only 35-40 micrometer.

Reviewer #3:

Remarks to the Author:

In this manuscript, the authors demonstrate the assembly of droplets using a peptide synthesis system from thioester derivatized amino acids under the same conditions. These oligopeptide droplets can grow over time, eventually catalyzing their own synthesis and growth autocatalytically. Repeated cycles of growth and division through dilution were demonstrated upon addition of an added physical stimulus, and the droplets appeared to be able to localize nucleic acids and lipids. In fact, lipid uptake decreased the size of the droplets, while nucleic acid uptake helped to prevent this dissolution. While this study is interesting and the experiments are relevant, there are a number of major and minor points that I have found related to the claims, background, and some experiments that preclude it from being published in its current form. Specifically, there is a general lack of information and discussion related to the prebiotic relevance of the system, including almost no discussion on primitive geological environments and processes as well as only minor discussion on just a few origins of life scenarios; the prebiotic relevance of the molecules chosen, their synthesis, the structures formed, and their resulting functions should be discussed in greater detail for it to have the amount of impact it deserves. I have listed these points below, and should the authors adequately address these points, I would recommend this manuscript for publication.

I would also highly recommend that the authors ask one of their colleagues or collaborators, who is a native English speaker, to carefully proofread their manuscript. There are also paid services available for this. As of right now, there are a large number of spelling and grammatical errors that should be fixed in order for the paper to be published. A few representative errors from the first few pages are listed in this review. I would recommend all of the errors be fixed before publication.

General comments:

1. The authors assume that "chemistry" means polymer generation, and "biology" means self-assembly and function. However, this is not necessarily the case, as many researchers would consider all of these (polymer generation, self-assembly, and function) to be "chemistry". The authors should clarify specifically what they mean by "chemistry" and "biology".
2. Because of the issue with definitions in point 1, the title and the abstract are too ambiguous with respect to linking "chemistry" and "biology". Perhaps it would be much clearer if these terms could be replaced with more specific concepts such as "generation" and "self-assembly/function". This way, a potential reader who only reads the title can understand immediately without having to read much further down for a specific definition.
3. There is generally a lack of tight transitions from one idea point to another in terms of writing style. At points, it feels like the manuscript is simply listing things in logical order (and to be fair, the order presented is very logical), but without the proper logical transitions. I think the manuscript would benefit greatly from insertion of proper and tighter transitions between ideas, both at the paragraph level, as well as the sentence/statement level. For example, the two full sentences in page 3, lines 13-17 are logical in order, but there is no logical transition word or statement. Something as simple as, "Although... (synthesis satisfied in certain conditions)...however...(assembly not satisfied in same conditions)..." could help greatly. I would extend this statement to the rest of the manuscript as well.
4. Page 3, line 18. What is the purpose and significance of this study? In this sentence, which is the thesis of the study, only "what" and "how" were listed. "Why" was not stated, and that is an important point that must be included, especially in relation to the previous paragraph, where "why" is framed. If "why" can be included in this sentence, then the reader can immediately understand the entire point of the paper very quickly.
5. In the introduction, it is unclear what the "autocatalytic" feature of the system is. How does feeding monomers into the system eventually catalyze its division? That is unclear, and it should be discussed in slightly more detail. In Figure 1, physical stimulus is mentioned, and autocatalytic growth is discussed in the results, but it is not mentioned in the introduction.
6. Can the entire system be considered to be undergoing autocatalysis? While the growth is likely autocatalytic, I am not sure about how exactly the division/proliferation step could be considered autocatalytic. You may want to include some discussion about the definition of autocatalysis and why you believe this system as a whole satisfies those definitions, not just the growth aspect. In my opinion, the division step is not autocatalytic, because a physical stimulus is necessary. If you can argue that physical stimulus is necessary for the growth as well, or that the autocatalytic

synthesis affects the division, then autocatalysis can be argued. However, as it stands, it is simply a system which grows autocatalytically, but which can exhibit self-replication through possibly non-autocatalytic means. However, if this assumption is incorrect, please explain why.

7. There is no discussion as to the prebiotic relevance or synthesis of nucleic acids or lipids (and which within these categories are more prebiotically plausible, such as fatty acids), even though the authors incorporate these other biomolecules into the system. The authors should discuss this. Additionally, the authors should discuss whether the concentrations of monomers and other biomolecules used in the experiments, and the experimental conditions, are prebiotically plausible, and if not, why this system is still prebiotically relevant.

8. The monomer used is likely not to have been prebiotically relevant, especially considering the BnSH group that was incorporated. In fact, the word used in Page 5, line 3 is "designed", which by definition makes the monomer not prebiotic. The authors also don't discuss the prebiotic relevance of the system. While it is understandable that the authors could have used this system as a modern model system to study a prebiotic process, that is not discussed at all. This discussion must be added, otherwise it is not clear to the readers that this system is relevant to prebiotic chemistry.

9. One of the major features of this system is that reduction of M is required to proceed with the reaction. Is synthesis of M plausible under the reducing atmospheric conditions on early Earth? And even if M could be produced, how can these reducing conditions on early Earth directly lead to synthesis and self-assembly. I think that given the correct discussion and argument, by incorporating the importance of reducing conditions into this system could greatly increase the relevance and significance of this system with respect to prebiotic chemistry.

10. In the dilution-extrusion system, there is no mention of how this simulates a primitive geological environment. The authors should discuss which geological and environmental features and processes on early Earth could provide the dilution, shear, and nutrient import needed for this process. The authors should also discuss how this process can be considered autocatalytic, if the system requires external stimuli.

11. In the section "Nucleic acid/lipid concentration in droplets" there is a discussion of nucleic acids, lipids, and peptides being the most important biomolecules in modern biology. However, the next statement quickly states that these must be incorporated into the current system to be relevant to origins of life studies. First of all, what is the rationale behind this? Just because something appears in modern biology, absolutely does not mean that it was required during the origin of life, as discussed by theories relating to non-biological origin of life (Chandru, K.; Mamajanov, I.; Cleaves, H.J., II; Jia, T.Z. Polyesters as a Model System for Building Primitive Biologies from Non-Biological Prebiotic Chemistry. *Life* 2020, 10, 6.) and the shadow biosphere (Davies PC, Benner SA, Cleland CE, Lineweaver CH, McKay CP, Wolfe-Simon F. Signatures of a shadow biosphere, *Astrobiology* 2009, 9, 241). I believe that the authors should qualify their assumptions that modern biomolecules can be used realistically in prebiotic chemistry experiments.

12. In addition to the previous point, assuming that a valid reasoning is included for the above point, the authors should differentiate between which classes of lipids, peptides, and nucleic acids could be considered prebiotic, prebiotic model systems, or not prebiotic. This is important because only certain peptides have been shown to have a relevant prebiotic synthesis, fatty acids/acyls are much more prebiotically plausible than phospholipids, and shorter polymers are much more relevant than longer polymers. The fluorescent tags on the biomolecules will also contribute to their ability to be incorporated within the droplets, and this possible contribution (or if there is no contribution) should be discussed. Of course, the fluorescent tags are also not prebiotic, so the authors should validate their incorporation into the system. Were experiments performed with just the dyes themselves?

13. Page 17, line 12-14. I don't think the claim that the RNA aggregating to the outside helped to protect droplets from size decrease from lipid addition is shown robustly by the data. Perhaps nucleic acid uptake contributed or was the main mechanism of action, but aggregation to the outside being the main mechanism of action is definitely not proven by the data. I think to robustly claim this, you must demonstrate another system, in addition to the lipids, in which addition of nucleic acids helps the system to avoid decrease in size. A control sample containing lipid and another molecule or biomolecule should also be tested to show that the size protection ability is conferred only by nucleic acids. It may be that incorporation of any molecule that localizes to the droplet boundary can protect it from being made smaller by lipids. Is it dependent on the size of the nucleic acid? RNA and DNA? Duplex single strand? Absence of fluorescent probe? Can

nucleotides have the same effect? Additionally, what is the mechanism by which the lipid decreases the size of the droplet, and how exactly does the RNA on the surface prevent this? This mechanism is key to this claim, and should be shown or discussed.

14. Page 17, line 8. It appears that the TR-RNA, with lipid present, enters the apparent BnSH center. However, in the absence of lipid, TR-RNA only resides on the outside. Is there a specific reason for this or experiments showing why this is the case? There may be some unknown lipid action that has not been probed, and this should be discussed.

15. Page 19, line 4. Again, the "mild prebiotic conditions" were not discussed. What was the pH of the solution and how does that compare to the proposed pH of primitive ocean waters? Was this study done in an oceanic environment, or a freshwater environment. In addition, no discussion of dehydration/rehydration cycles were included, even though there were some hydration processes (for dilution) used in the system. These are the discussions that should be included throughout the paper in order to claim that this system is without a doubt prebiotically relevant.

16. Page 20, line 15. Diffusion of components away from the system is not the main unsolved problem of the RNA world theory. There are many issues, including reactivation of spent monomers, spent monomer inhibition of replication, synthesis of RNA monomers in a complex chemical milieu, replication and catalysis of RNA in messy systems, fast hydrolysis of RNA and monomers, the "reannealing problem", among others, each of which have partial solutions, but not full solutions. While increasing concentration of such nucleic acids is certainly helpful to achieve some of the RNA world scenarios, there are still many more pressing problems that must be solved, which are being tackled by Szostak, Sutherland, Powner, Brenner, Hud, and many others in the RNA origins field. I think a deeper understanding and discussion of this field, and the recent research, should be achieved and included, respectively, rather than a broad generalization of how solving one problem could tie this system to an RNA world origins model.

17. Page 20, line 16 and page 21, line 4. A simple fluorescence recovery after photobleaching (FRAP) experiment should be performed to robustly show that whether the RNA is stably compartmentalized within the droplets without diffusion. This would also show whether the information from the RNA could be delivered to other droplets in a robust manner, without such diffusion into the bulk. If large amounts of diffusion occur, then it is likely that genetic exchange occurs even in the absence of self-replication.

18. The conclusion does not talk at all about the "missing link between biology and chemistry". In fact, after the introduction (where it is mentioned in passing), this point is not discussed at all, although it is in the title. Either the title should be amended, or, this point should be discussed in much greater detail in both the introduction and the discussion.

Minor comments:

1. Page 2, line 9. Is a protocell required just for exploration of the origin of life (this implies that it is not a real structure and is only used in lab simulations of origins of life studies)? Or is it by definition required for life? I think this should be clarified.

2. Page 2, line 10. Some, like Szostak and Deamer (among others), have suggested that a protocell is a membrane-bound minimal vesicle. We don't know what is the "real" protocell, so it would be wise to include some background on non-coacervate droplet-based protocell models before assuming that CDs (especially from LLPS) are the globally accepted protocell model. This is but one of many theories.

3. Page 2, line 10. The sentence starting with "A CD..." should have at least one reference.

4. Page 2, line 15. Although the authors discuss various biomolecular "world" scenarios, it would be relevant to also discuss non-biomolecular or composomic "world" scenarios, as proposed by Chandru, et al. (Chandru, K.; Mamajanov, I.; Cleaves, H.J., II; Jia, T.Z. Polyesters as a Model System for Building Primitive Biologies from Non-Biological Prebiotic Chemistry. *Life* 2020, 10, 6.) and Segré, et al. (Segré, D.; Ben-Eli, D.; Lancet, D. Compositional genomes: Prebiotic information transfer in mutually catalytic noncovalent assemblies. *PNAS* 2010, 97(8), 4112), respectively.

5. Page 2, line 22. I would not argue that Mann was the main pioneer of CDs in origins of life research. He is one of many who have been pioneering this field, including Keating, Bevilacqua, Boekhoven and Tang.

6. Page 3, line 4. How about the study by Yin et al. (Yin, Y., Niu, L., Zhu, X. et al. Non-equilibrium behaviour in coacervate-based protocells under electric-field-induced excitation. *Nat Commun* 7, 10658 (2016)), where they demonstrate budding and growth due to changes in electric field

stimuli?

7. Page 3, line 14. What about the conditions where there is an unlimited reservoir of peptides available (and were synthesized in conditions before the assembly/reproduction conditions)? Is such a condition possible?

8. Page 3, paragraph beginning with line 18. What is the significance of thioesters in prebiotic chemistry? Are they themselves prebiotic? Is their synthesis and conjugation prebiotic? Why were they believed to have been the most relevant system by de Duve? These questions should be answered to frame the significance of this theory in the global framework.

9. Page 5, line 3. How do the monomers synthesize peptides? Do you mean that they can be synthesized into peptides? This is a major difference in nomenclature, as the word "synthesize" assumes that the monomers in some way are affecting their own synthesis. Is this true? This should either be changed, or explained.

10. Page 5, line 7. The authors should explain why this molecule is "activated", and why activated monomers are essential to the process. In addition, is the precursor prebiotically relevant? Was the synthesis prebiotic?

11. Page 5, line 10. The prebiotic relevance of BnSH has to be discussed. And if it is being used as a modern model analog system, why can it be used and the results extended to true prebiotic chemistry? What happens to the system in the absence of BnSH? Please also include a reference regarding how BnSH can promote self assembly.

12. Fig. S1. Please define all of the abbreviations used in the synthesis scheme. This includes the difference between SBn and BnSH, as well as DCC, Boc, and HOBt. These are all non-standard chemicals that must be defined, especially as they are used in the subsequent figures. Additionally, this synthesis scheme (especially the BnSH addition step) should be added to the methods section. Finally, Mpre should be represented identically in both the main text and in the supplement. Right now, the two chemical structures, although technically identical in structure, are not depicted identically (the SI uses Rn→ notation, while the main text does not), which may lead to major confusion amongst readers.

13. Figs. S2-S3. What is the significance of these NMR spectra? This can be as simple as stating something like "...indicating the successful synthesis of Mpre." Additionally, if the journal requires it, please submit an NMR peak list.

14. Page 5, line 12. What is the significance of DTT? Is it prebiotically relevant? Just as a modern model reducing agent? A discussion of the relevance of a reducing atmospheric and ocean conditions on early Earth should be discussed here. Such a discussion would increase the significance and relevance of the paper.

15. Page 5, line 21. How can you be sure nanometer-sized aggregates were formed? Did you validate this through experiments?

16. Page 6, line 3: how long did it take before the droplets completely coalesced into a single structure? Or was this not observed?

17. Page 6, lines 11-13: I don't think that Fig. 2c has open squares or crosses. The lines are instead indicated by different colors. This should be changed either in the text, or the figure.

18. Fig. S5. The synthetic method of this molecule should be listed in the methods.

19. Figure 2c: the black line is not easily visible.

20. Fig. S8b. While the table is presented, please directly show the full spectrum. This is a better way to visualize the data.

21. Page 9, line 3. what does "disperse" mean? Did you mean turn turbid? Additionally, these micrographs should be included as well.

22. Page 8, line 8. What evidence do you have of associative LLPS? Why could it not have been segregative LLPS?

23. Fig. 4: The figure has very low image quality and is very blurry. It should be replaced with a much higher quality figure.

24. Page 14, line 16. I think this statement should be introduced in the introduction, in order to immediately highlight the significance of this study.

25. Page 15. What is the mechanism by which some fluorescent molecules concentrated on the edges, while others concentrated in the center? For example, it was suggested that the edges are more hydrophilic, while the interior is more hydrophobic. Is there any additional evidence for this?

26. Fig. S12. What was the reason that nucleic acid solutions were observed 30 minutes after addition, while lipid solutions were only observed 16 hours after addition?

27. Page 15, line 22. About the size change when both lipid and nucleic acid were added together, this data should be introduced here and also a similar FACS analysis should be shown.

28. Fig. 5. The letter labels for Fig. 5 are out of order. They should be listed in increasing order from left to right, and then to the next row. As it stands, it is very difficult to read this figure properly (for example, b should actually be f).
29. Fig. 5. You cannot call a part of a figure before that part of the figure was called. For example, in the figure caption, 5g was called in 5d. However, you must call 5g first, and then it can be referenced in another part of the figure. Thus, Fig. 5 should be completely rearranged so that this is the case.
30. Page 16, line 16. Why is this the assumption exactly? If there is experimental or theoretical evidence, then this should be presented. Some type of IR or Raman microscopy, DESI-MS (or another type of spatial MS such as SIMS), or even a peptide derivatizing fluorescence dye which can allow spatial fluorescence can show direct evidence of the spatial chemical structure within the droplets.
31. Page 19, line 5. Other theories of origins of life could also be relevant, such as the polyester-based origins of life model (Tony Z. Jia, Kuhan Chandru, Yayoi Hongo, Rehana Afrin, Tomohiro Usui, Kunihiro Myojo, and H. James Cleaves II. Membraneless polyester microdroplets as primordial compartments at the origins of life, PNAS 2019, 116, 15830), the Shadow Biosphere (Be Davies PC, Benner SA, Cleland CE, Lineweaver CH, McKay CP, Wolfe-Simon F. Signatures of a shadow biosphere, Astrobiology 2009, 9, 241), or composome model (Segré, D.; Ben-Eli, D.; Lancet, D. Compositional genomes: Prebiotic information transfer in mutually catalytic noncovalent assemblies. PNAS 2010, 97(8), 4112). How these theories could be incorporated by the system should also be discussed briefly.
32. Page 19, line 6. This sentence should be rewritten. As it stands, it does not convey the meaning that the authors likely want to convey.
33. Page 19, line 9. The thioester world scenario was not introduced in adequate detail. This should be done in the introduction.
34. Page 20, line 22. It should be noted that because RNA is so labile, that some of these potential substrates could quickly hydrolyze the RNA. This possibility should be discussed, and also underscores why it is important to continue discussing the conditions used throughout the manuscript, and how these conditions are conducive to biomolecule function and persistence, but also to true geological conditions on primitive Earth. It is also necessary to mention that many RNA reactions can only be performed
35. What happens to the system in the presence of Magnesium (or another divalent cation)? Most RNA-based reactions such as ribozyme catalysis or non-enzymatic replication require such a cation, and the incorporation of such cations into the system should be discussed.
36. Page 21, line 16. To claim "droplet world" also requires the authors to properly cite those who have proposed and researched it in the past. Please include these citations and give proper credit where credit is due.

It is understandable that as the authors are not native English speakers, that a large number of minor spelling and grammatical errors exist (which don't detract from the science presented, just the readability). To improve the readability of the manuscript, I would recommend that the authors ask one of their colleagues or collaborators, who is a native English speaker, to carefully proofread their manuscript. If this is not possible, there are a number of paid services available. I think this step is imperative to achieve the most easily and widely understood manuscript as possible. Here, I have listed a few examples of things that should be changed from the first few pages, although this is not an exhaustive list. I hope that this gives the authors a sample of the number of errors in the entire manuscript (and numerous errors in the supplementary information) that must be fixed.

1. The semicolon in the title should be a colon.
2. "a" in line 16 of the abstract should be "one".
3. In line 17 of the abstract, "yet" should be inserted between "not" and "been".
4. In line 20 of the abstract, "a" should be inserted before "proliferating".
5. In line 1 of page 2, "droplet" should be plural.
6. The sentence in line 1, page 2, beginning with "The droplet showed..." is grammatically incorrect and is a statement.
7. Line 8 of page 2, "scene" should be "sense".
8. And many others throughout the manuscript.

Reviewer #1 (Remarks to the Author):

This paper titled “Self-proliferating Liquid-liquid Phase separated Droplet; the missing link between Chemistry and Biology on the Origin of Life” by Matsuo and Kurihara describes coacervate droplet formation by in-situ synthesized peptides that can accumulate and distribute RNA and lipid heterogeneously within the droplet combining RNA, lipid and peptide in one droplet. They show selfproliferating growth-division cycles of the droplet. There are some areas of novelty in this work, in particular, in showing that in-situ peptide formation can induce droplet formation. However, we found there are a number of places where the experimental data is not sufficient to support all of the assertions made in the paper. Above all, a complete and thorough characterization of the coacervate droplet has not been demonstrated; for example, it is not clear whether the droplets are forming via a charge separation or a hydrophobic interaction. Introduction of several technical terms need additional experimental verification to justify the use of these statements. As it stands, this manuscript would not be acceptable for Nature Communications, however, due to the novelty in some aspects of the work (as described previously), we recommend that this work should only be accepted by Nature communications after major corrections.

Major comments:

1. The style of writing in the abstract and introduction is difficult to understand in places. This is partly due to the grammar where the sentence structure hinders the flow and the meaning of the text. In addition, the authors use terminology such as self-proliferating, self-maintaining, selfreproduction etc. while it is not always clear the reason why these terms are used considering that they do not always correlate with the experimental observations. For example, in this article, droplets were described as self-proliferating. Self-proliferating suggests that the system changes without any external input. To drive division of the droplet, the authors introduce external shearing which is not self-driven proliferation.

Answer

The growth of droplets in this study does not always require the external shearing for division itself. We introduced shearing processes in the DLS particle measurement to measure the size of the droplets before/after the addition of precursors quantitatively and repeatedly. In fact, we also detected by particle size measurements that the droplets grew and then divided right after droplet precursors were added. We rephrased “self-proliferation” to “proliferation” because of a mistake in writing. The definitions of “self-reproduction (P.2, L.18)”, “self-maintenance (P.3, L.14)” and “self-replication (P.22, L.6)” are clarified in the text.

2. The authors make an interesting reaction setup and observation regarding the in-situ formation of droplets after dithio bond cleavage and peptide bond formation to generate a

peptide chain. This is novel work. However, the additional control experiments which show that BnSH is required to drive the droplet formation do not fully describe or explain the mechanism of droplet formation. Throughout the text, the authors describe this as a coacervate droplet and the authors show and give no clear evidence that the droplet is forming by coacervation which typically occurs between two oppositely charged components. A more detailed understanding is required to confirm that droplet formation is driven solely by coacervation as described. This could be achieved by purifying the peptides and characterizing the phase diagrams. Additionally, they could characterize the properties of the droplets, for example, FRAP and/or coalescence properties to determine whether they are indeed coacervate droplets. It would be interesting to see if fluorescently labelled amino acids (or peptides) localize inside the droplets to observe their specific distribution within the droplet. Indeed, the structure of the droplet could help in explaining the RNA and lipid accumulation in the latter part of the manuscript.

Answer

Thank you for your valuable opinion. As the reviewer's statement, we directly determined the composition of the droplet by performing Raman microscopy measurements. The results were summarized in Fig. S13 of the new SI manuscript. We are grateful to you.

3. The observation of the heterogeneous accumulation of the RNA and lipid is interesting however as it stands it is only a phenomenological observation. With some additional experiments i.e. partitioning of different molecules with different hydrophilicity and zeta potential experiments, the authors would be able to gain insights into the structure of the droplet which may also help to provide an understanding of how the droplet forms and what are the critical components for droplet formation is (see point 2). For the RNA and lipid localization experiment, control experiments showing the localization of the dyes alone would be required to determine whether the dye plays a role in the heterogeneous distribution of RNA/lipid. Is there a possibility that the heterogeneous distribution of the RNA and lipid could be driven by a size exclusion effect of RNA rather than a hydrophobic/hydrophilic effect as has been observed with other polymer systems?

Answer

To investigate the localization of the hydrophilic DNA, we added 20mer DNA to the droplet dispersion and measured the droplets after 1 hour by Raman spectrometry. The results are shown in Figure A1. Focusing on the DNA's phosphate peaks (784cm^{-1} , 830 cm^{-1} , 1090 cm^{-1} [ref. *RSC Adv.*, 2018, 8, 25888]), it was found that there was slightly more DNA in the peripheral part in the droplet. However, due to the large peaks of the droplets and the possibility of DNA damage by laser irradiation, the difference was not as clear as in the fluorescence microscope image (Fig.5g).

Figure A1. Raman spectra of the droplets 1 h after addition of DNA. red: the central part of the droplet, blue: the peripheral part of the droplet. The peaks derived from DNA indicate that DNA present in peripheral part.

4. We recommend clear labelling of the figure legend i.e. identification or marking the curves in the inset of the figure and full experimental descriptions. Labelling of graph axis should be consistent to avoid misinterpretation of the data and axis ticks should be either linear or log scale, please see figure 4b and 4c x-axis and 4e y-axis which needs to be amended. General figure presentations could be improved, for example, page number 7, figure 2c – the data curves are too thick and the black line and error bars cannot be seen at this resolution.

Answer

We revised the Fig. 4b, 4c and 4e (μm to nm). We revised the Fig.4b and 4c in order for readers to understand. The new vertical axis in the Fig. 4b and 4c in the revised manuscript shows the percentage of the number of the observed droplets at the each particle size, where the total number of droplets in the sample is set to 100(%). The results and our claim haven't changed.

As the reviewer pointed out, the figure was revised. In the previous manuscript, we had a mistake in the comments about the Fig. 2c. There were no white squares and crosses in the Fig2c. We revised the full circles, open squares, and crosses in the previous manuscript to the red line (P.5, L20), green line (P.6, L15), and blue line (P.6, L17) in revised manuscript respectively. In addition, we deleted the sentence in the previous manuscript “which is about half of the mixture solution of **Mpre** and DTT” (P6, L15).

5. Mass spectra (not just the values) should be included and the peptide multimers should be quantified. It would also be interesting to see the change in the peptide length by mass spectrometry during consequent reactant addition-shearing cycle to clearly understand the

effect of periodic additions on peptide length and quantity with a correlation to the droplet size.

Answer

We inserted the complete MS spectrum in Fig. S8, and also the magnified spectra the peptides of which were detected in the Table of Fig.S8b. We also performed another MS experiment. The intensity changes over time of non-thioesterified peptide **A**s and thioesterified peptide **B**s in the droplets were measured by ESI-TOF mass spectrometry. The results are shown in Figure A2. The names of the peptides detected by MS measurement are indicated by the peptide type **A** or **B** following the degree of polymerisation n . The vertical axis is the normalized intensity of each peptide (left to right, **1A**, **1B**, ...) detected in the sample at 0 minutes, where the initial intensity was set to 100. Smaller degrees of polymerisation ($n = 1, 2$) are decreasing with time, whereas larger degree of polymerisation ($n > 2$) was found to increase at 24 h after. These results suggest that the peptide monomers were gradually polymerised to form oligopeptides.

Figure A2. Structural formula of detected peptides and graph of time course for peaks of ESI-TOF MS.

6. On Page 4 line 2 the authors state that: “We also showed that self-maintenance ability of the LLPS droplet against dissolution by lipids and co-existence between nucleic acids and lipids in the same LLPS droplet, depending on the concentrated nucleic acids localization into the inner boundary of the LLPS droplet with the assistance of generated peptides.” This is a big statement and when considering the data, it is not clear how the droplets are self-maintaining themselves. With reference to point 2 and 3, it would be interesting to identify the co-existence of the layers of two separate phases specifically made of peptide-RNA and BnSH-lipid inside the droplet to see if there is any correlation between the additional components and the size of the droplet.

Answer

As described in Supplementary Figures S13 and 14, a difference in hydrophobicity appeared between

the central part and the peripheral part of the droplet, and the peripheral part was more hydrophilic. Therefore, it is considered that the nucleic acid tends to be localized in the peripheral part. It can be said that the presence of the nucleic acids and the peptides in the peripheral portion makes it difficult for the central part of the droplet to be affected by water, and thus makes it difficult for the droplet to collapse. This was also supported by the fluorescence microscopy image in Fig. 5g.

7. Please describe the method of shearing i.e what is it and how is it applied. It would be useful to have a control experiment i.e. experiment 4e without adding Mpre to observe any changes in droplet size and number over the time. This would allow one to understand the effect of shearing, alone, and to confirm that the growth of droplets changes due to the addition of precursors. Is the data from figure 4e obtained from DLS data such as that presented in Figure 4c? If so, then it is not clear how the error bars are correlated, for example, the distribution of the data from the DLS is approximately +/- 150 um while the error in the 4e looks circa +/-70 nm. Please clarify this. Including the raw data of Figure 4e into the SI would add additional clarity.

Answer

We specified shearing as an extrusion method using a syringe (p. 13, L.5). Figs. 4a-c refers to a series of experiments. Fig. 4b corresponds to the first step in Fig. 4e, and shows the DLS measurement when the first addition of M_{pre} to the water. Fig. 4c shows the change in particle size when M_{pre} was added to preexisting droplets, and the preexisting droplets were not extruded. The experiment in Fig. 4c verifies the passage of droplet formation by the addition of M_{pre} , which was shown in Fig. 4a. On the other hand, Fig. 4e shows an experiment in which the addition of M_{pre} and extrusion were repeated to confirm the recursiveness of the droplet size when M_{pre} was added to the droplets. The extrusion process was performed because it was necessary to keep the size of droplets the same each time. Therefore, the experiment in Fig.4c and that in Fig. 4e are completely independent. To validate the reviewer's proposed conditions, we added water without Mpre to the extruded droplet dispersion. No significant droplet formation was observed (Figure A3).

Figure A3. a) Comparison of droplet dispersion in vial at 0 and 24 h. b) Comparison of phase contrast microscopy images of droplet dispersion at 0 and 24 h. Scale bars represent 50 μm . To show that the increase in size of the droplets is not a growth due to fusion, we performed the experiment that droplets does not increase in size in the case of droplet only. A droplet dispersion was prepared 24 h after the addition of Mpre (5 mg) and DTT (4 mg) to water (1 mL). The droplet dispersion was filtered through a 100 nm pore size membrane filter, and the solutions were compared between immediately and 24 h after the addition of an equal amount of water.

8. Page number 9, line number 15: “When a smaller amount of DTT was added to reduce the reaction rate, the Mpre residual rate curve assumed a sigmoidal shape (Fig. 3c), confirming the autocatalytic nature of the peptide synthesis.” - In an autocatalytic reaction, concentration vs time curve shows a sigmoidal shape originated from stochastic delay as well as product enhancement that is also one of the reactants. Here the product is going out of the reaction phase (by LLPS) hence shifting the reaction equilibrium towards the right-hand side and does not fulfil the criteria of autocatalysis. Indeed, the authors do not fit the curve to an autocatalytic model to confirm that the reaction is autocatalytic. To confirm that the reaction is autocatalytic and sigmoidal the authors could lower DTT concentrations and undertake a proper curve fitting to the data. Please also clarify why this reaction would be autocatalytic. Does the size of the droplet increase correspond to an autocatalytic reaction rate?

Answer

Since the system is physical autocatalytic reaction, which was proposed by Fletcher, with the droplet as the reaction field, the equilibrium of the reaction may be shifted to the right when the product is removed from the reaction field by LLPS environment and may not meet the criteria for an autocatalytic reaction. Here, we confirmed whether this reaction is autocatalytic or not from the change of the chemical shift of the benzene ring proton in the \mathbf{M}_{pre} with respect to the density of the extruded aggregates. We describe the sentence in Supplementary Figure 11 as follows: “After adding 5 mg of \mathbf{M}_{pre} and 4 mg of DTT to 1 mL of D_2O , the change in reaction progress was examined from 1H NMR. The reaction progress was calculated from the integrated value (I) of the proton of the benzene ring with the peak shifted from 7.2 ppm-7.6 ppm (x) to 6.6 ppm-7.1 ppm (y). Reaction progress: $=100 \times I_y / (I_x + I_y)$ %. Number of samples is 3, error bars represent standard deviation.” Since the reaction was fitted on the sigmoidal curve (Figure A4), we concluded that this reaction was an autocatalytic reaction.

Iterative algorithm	Levenberg-Marquardt Method
Formula	$y = a * (1 + (d-1) * \exp(-k*(x-xc)))^{-1/(1-d)}$
a	69.70121 ± 0.66271
xc	77.36148 ± 1.94612
d	0.46532 ± 0.0266
k	0.00822 ± 2.50981E-4
Reduced Chi-Sqr	0.40012
R-Square (COD)	0.99987
Adj. R-Square	0.99985

Figure A4. Reaction progress from NMR measurement and the formula of sigmoidal curve. Reaction progress was estimated from the chemical shift of the benzene ring proton in the M_{pre} .

Raman microscopy (Figure A5) shows that BnSH is also localized in the peripheral part of the droplet although the central part of the droplet is hydrophobic. This localization concentrates the monomer at the interface and subsequently accelerates the amide formation. This acceleration of the reaction results in rapid formation of the interface of the droplet, and thus the reaction is considered to be autocatalytic.

Figure A5. Raman spectroscopy of droplet dispersion. a, Optical microscopy image of a LLPS droplets. The scale bar represents 20 μm . b, Raman microscopy image of the yellow rectangle in Fig. S15a. c, Raman spectra of LLPS droplets. The red line is the central part of the droplet (red rectangle in Fig.S15b). The blue line is the peripheral parts of the droplet (blue rectangle in Fig.S15b). In order to investigate the ratio of hydrophobic and hydrophilic components in both the central and peripheral parts of the LLPS droplets, the Raman spectra of each part were measured by laser Raman microscopy. The benzene ring-derived peak of benzyl mercaptan ($\sim 1000\text{ cm}^{-1}$) as the hydrophobic component and the water-derived peak ($\sim 3400\text{ cm}^{-1}$) as the hydrophilic component were detected and the benzene ring/water ratio (B/W) was compared at each part. Mpre (2.5 mg) and DTT (2 mg) were mixed, and then 100 μL of MilliQ water was added immediately. The mixture was stirred with vortex for 15 seconds. After 1 hour of stirring, the droplets were observed with a laser Raman microscope. The average value was calculated from the spectral data obtained at the four different points of the central parts of the droplets and the baseline-corrected spectrum were shown in Fig.S15 c (red line). Similarly, the spectrum from the peripheral parts were shown in Fig.S15c (blue line). Based on the obtained spectral data, the B/W value at the central part was calculated to be 0.370 ± 0.016 , whereas that at the peripheral part was calculated to be 0.288 ± 0.038 . This result indicates that the central part is more hydrophobic than the peripheral part in the droplets.

9. In page number 2, line number 10, the authors mentioned that “A CD is formed from organic molecules via liquid-liquid phase separation (LLPS), mainly resulting from the spontaneous assembly of oppositely charged molecules or from hydrophobic polymers in general and prebiotic polymers or oligomers on the abiogenesis scenario in particular” – proper referencing is required here especially as coacervation is typically not driven by hydrophobic interactions.

Answer

We cited a paper of Mann's lab as reference. 5 [Koga, S., Williams, D., Perriman, A. et al. Peptide–nucleotide microdroplets as a step towards a membrane-free protocell model. Nature Chem 3, 720–724 (2011)].

10. The authors state on page number 5, line number 21: “These results indicate nanometer sized molecular aggregates were formed until 1 hour after mixing the M_{pre} and DTT, and then they grew or fused to large enough to be observed with a microscope” – If the droplet size is not measurable it is unreasonable to speculate on the droplet size and better to make a clear remark that the droplets were observable by microscopy after 1 hour of the experiment.

Answer

As the reviewer pointed out, the nanometre-sized molecular aggregates could not be observed under an optical microscope in principle. We discovered this fact by the particle size distribution measurements. Therefore, we moved the following statement "*These results indicate nanometre-sized molecular aggregates were formed until 1 hour after mixing the M_{pre} and DTT, and then they grew or fused to large enough to be observed with a microscope*" to the proper location of the size measurement results (P.11, L.16).

11. Is there any difference in the chemical shift of the NMR spectra of the amino acid or peptide signals before and after droplet formation? How would this affect the experimental observations?

Answer

We compared the ¹H NMR spectra of M_{pre} and that of oligomers. The characteristic peaks in the NMR spectrum of oligomers were derived from amides and thiols of oligomers. The proton peaks derived from aromatic rings and residues observed in the spectrum of M_{pre} disappeared in that of the oligomer.

Figure A6. Comparison with ¹H NMR spectra of M_{pre} and Oligomer.

12. For Figure 3a the images which show turbid solutions does not confirm that droplets are

formed. These images could also show the formation of precipitates, therefore, please verify that the visual turbidity correlates to droplet formation by optical microscopy.

Answer

A microscopy image of a droplet with the relevant composition has been shown in Fig. S9. However, as it was unkindly to readers, we inserted the writings about Fig. S9 in both the manuscript (P.10, L.5) and the caption of Fig. 3a.

13. On page number 11, line number 9 the authors say: “Of course, some oligopeptides will also be generated outside and then incorporated into the existing LLPS droplets in the latter case. Referencing these two possibilities, in order to identify the actual oligomerisation site, equal volumes of Mpre and DTT were added to 1 mL of LLPS droplet dispersion 24 hours after the first addition of Mpre (10 mM) and DTT (25 mM), and the time evolution of the size distribution of the LLPS droplets was then monitored (Fig. 4c). The particle size of the existing LLPS droplets increased with time, and no additional peak corresponding to newly formed LLPS droplets was detected. This result strongly supports the view that oligomerization occurs inside or at the interface of the LLPS droplets: that is, these findings point to autocatalytic self-reproduction of the LLPS droplets due to the ability of LLPS droplets to play as active sites for oligopeptide generation.” This result is not as convincing as they say especially as microscopy images shows small droplets (less than 5 um in solution) which are not shown in the DLS data. Could it be that these smaller droplets are also present in the solution but not detected by the DLS?

Answer

To begin with, a result of DLS particle size distribution measurements and that of optical microscopy observation are not inconsistent. The reason for this is as follows: In principle, whereas DLS measurements are performed on molecular aggregates that are capable of Brownian motion, microscopy observation targets large-sized aggregates that have precipitated without Brownian motion. Molecular aggregates targeted by the DLS particle measurements are often dispersed upward in prepared specimens under microscopy observations. In addition, the floating molecular aggregates smaller than 1 micrometer cannot be observed in principle of optical microscopy. Actually, the microscopy images photographed while changing focus along the z-axis are attached below. Indeed, these microscopy images prove that the view is different between above and below part along z-axis (Figure A7).

Figure A7. Microscopy images of the droplets. From the top left photo to the bottom right photo in this figure, the droplets were observed from the bottom of the prepared specimen to the top at every 30 μm . As shown in the above results, particle sizes larger than 3 μm are rare and are only present at the bottom of the specimen glass. Therefore, most of the droplets are smaller than 3 μm , or so small that they cannot be observed under the microscope. The scale bars represent 50 μm .

14. After shearing, it would be expected that the number of droplets will increase. If this were the case then this would be observable by DLS experiments. In Figure 4e, from the 3rd to 6th period, there is an increase in size of the droplet just after addition and then a decrease and stabilization of droplet size after time. It is interesting to see that these droplets maintain a steady size after external perturbations which is not typical of coacervate droplets. An explanation or reasoning of this behaviour is required. In Figure 4e, labelling shows first addition and then extrusion while the reverse is described in the article. Moreover, the size of the droplets from the DLS does not correlate to the microscopy data (see point 12). The average size in DLS is much larger than observed in the microscope whilst there is no evidence of the small 5-10 μm sized droplets in the DLS that are clearly seen in the microscopy images.

Answer

Compared with the conventional coacervates, the steady size of our droplet was not due to the fusion which is a physical phenomenon, but to the self-production reaction. Furthermore, the results of fluorescence microscopy and Raman microscopy spectra suggest that the droplets have a layer with a gradation of hydrophobic and hydrophilic parts. This layer may inhibit the phase separation due to fusion after the chemical reaction. This possibility is supported by the results that the droplets disappeared when the benzyl mercaptans inside the droplets volatilize, or when the droplets themselves were stirred. In addition, as reviewer pointed out, the reverse order in Figure 4e was indeed confusing. We added first triangle in "1st Period" of Figure 4e to emphasize that **Mpre** already existed in the water. We also revised the sentence in the manuscript, "*We monitored the droplet population over six periods: initial droplet formation period and five cycles of **Mpre** (nutrient, white triangle) addition and extrusion (shear, black triangle) to the existing uniformly-sized droplets (Fig. 4e).*" (P.13, L.3-6). With respect to the correlation between DLS particle measurement and microscopy

observation, we discussed in the previous answer A.13.

15. Page number 16 line number 23 and S13: “In the presence of RNA, the fluorescence intensity of the aqueous layer containing the oligopeptides was less than one-third of that of the aqueous layer without the oligopeptides” – It is not clear how the fluorescence intensity of tagged RNA decreases in the presence of the peptide in the aqueous layer, please can the authors explain this observation further.

Answer

We revised and simplified the writings (P.20, L.24- P.21, L.6) about results to emphasize the claims by moving the details of the experiment to the S16 section. We revised the sentences as follows: *“The fluorescence intensity in the aqueous phase decreased in the both cases of TAMRA-RNA solution and BODIPY-lipid solution(Fig.S16a), but there was a difference in the degree of the fluorescence decrease between the two in the presence of the oligopeptides (Figs.S16b, d). In the case of TAMRA-RNA, the fluorescence intensity in the presence of the oligopeptides was less than one-third that in the absence of the oligopeptides, whereas in the case of BODIPY-lipids, there was little difference in the fluorescence intensity due to the presence of the oligopeptides. This result was similar even when the fluorescence moiety was changed (Fig.S16c, e)”*.

In the S16, we illustrated a protocol and added a description of it in the caption to further enhance the reader's understanding of the experiments.

16. Page number 17, figure 5 shows a nice experimental design, demonstration and representation, although a better explanation of the FACS data is required in the figure legend and in the text. In addition, the microscopy images are quite low resolution and it is difficult to see some of the droplets.

Answer

It was very confusing for readers that the figures and the description were separated. We moved the Fig. 5 to make the relation between the explanations and figures clearer. Also, we increased the resolution of the microscope images.

17. On page 3 line 3, the authors state” Very few studies have reported recursive self-reproduction of supramolecular assemblies under periodic stimuli because these metastable assemblies tend towards an equilibrium state”. This statement requires some additional references as there are some studies which have shown this already, for example, recent work by the Spruijt Lab.

Answer

We quoted the Spruijt Lab's publications properly. Thanks for informing us the interesting works.

Ref.[25] Lu, T., & Spruijt, E. (2020). *Journal of the American Chemical Society*, 142(6), 2905-2914.

Ref.[26] Nakashima, K. K., Baaij, J. F., & Spruijt, E. (2018). *Soft Matter*, 14(3), 361-367.

18. The authors claim that this is the first example of a coacervate which combines peptide, RNA and lipids in one example. This is not the case as there has been previously published work which shows examples of coacervate droplets with these materials in combination, see work from the Mann Lab. This should be amended and the citations included.

Answer

We revised the sentence as follows: "*Although it was reported that a CD⁵ expressing a higher-order function by the encapsulation of the main constituents of living organisms, i.e., RNA, lipid and peptide, no self-reproducing CDs that induces the interactions among the constituents has not yet been constructed.*". (P.2, L.13-16). We added Mann's paper (Koga, S., Williams, D. S., Perriman, A. W., & Mann, S. (2011), *Nature chemistry*, 3(9), 720-724.) as a reference 5 in the revised sentence.

19. Page 3, line 13, "In principle, a CD can self-reproduce only if both the CD formation condition and the peptides representing the CD building blocks reproduction condition are satisfied." This statement requires some additional clarity. What reproduction condition needs to be satisfied?

Answer

We clarified the definition of self-production in our response to the reviewer's comment 1. In addition, we clarified the conditions required in the coacervate's self-production [P.3, L.24].

20. "The C-terminus of M was capped with a benzyl mercaptan (BnSH) leaving group, which works as a seed droplet to promote self-assembly in terms of hydrophobic interactions." Please explain the role of BnSH in generating the seed droplet.

Answer

We deleted the writings after "which" because it was based on speculation. The sentence was revised as follows: *The C-terminus of M was capped with a benzyl mercaptan (BnSH) leaving group.* (P.5, L.16).

21. The authors used non-conventional abbreviations for TAMRA (TR) and BODIPY (BP). TAMRA and BODIPY are already abbreviations and should not be abbreviated further.

Answer

We revised BP to BODIPY [in the text and Figure 5 legend] and TR to TAMRA [in the text and Figure 5 legend] in the text. Thanks for pointing out.

22. Figure 5i and 5j: A clear description of the plots and figure legends are required.

Answer

We revised the legends of Fig 5i and 5j. We also specified the number of the droplets used for kernel density estimation.

23. Scale bars in microscopy images are not suitable for the size of the droplets.

Answer

We observed molecular aggregates larger than 1 μm , which can be observed under an optical microscope. Therefore, the scale bar is suitable for the micrometer order.

24. Figure S9: An explanation in the text is required to explain why the droplet population in the two images look similar.

Answer

We modified the sentence in Fig. S9 legend (SI L.182) as follows: “*DIC microscopy images of solutions. (a) mixtures of oligopeptides (10 mM) and BnSH (10 mM). (b) Mixtures of oligopeptides (10 mM), BnSH (10 mM) and DTT (25 mM). The concentration of these molecules were calculated from the conversion from Mpre. Both solutions formed droplets, indicating that the oligopeptides and BnSH were required for droplet formation. Scale bars represent 20 μm .*”

25. Page number 17, line number 6: “Figs. S13b and 8d, respectively.” There is no figure number 8d.

Answer

Thanks for pointing out the typographical error. We revised and moved the sentence “*Figs.S16b and d, respectively.*” to Supplementary Information.

Reviewer #2 (Remarks to the Author):

The manuscript by Matsuo and Kurihara describes an interesting system of active liquid-like droplets that are formed spontaneously upon release of a reactive thioester monomer from a disulfide precursor. The authors demonstrate that their droplets contain an oligomer of the reactive cysteine thioester monomer, and that they are formed more rapidly when more reducing agent is present. The droplets can undergo a so-called self-reproduction cycle by periodically extruding them through a small filter and feeding the extrudate with precursor molecules again. Finally, these droplets are shown to concentrate both fluorescently-labelled RNA and lipid molecules, suggesting that they may have played a role to reconcile the seemingly separate theories for the emergence of living systems based on lipids, nucleic acids and droplets. Research into protocells has gained significant attention over the past few years, and several groups have reported systems in which active droplets start showing convincing life-like properties. An important barrier is that many of these systems are still based on complex synthetic molecules. The system reported here seems to make a significant contribution to the field of protocells, as it provides a pathway for the spontaneous generation of droplets out of relatively simple precursor molecules. However, in my opinion the system and the experiments presented here are poorly characterized, and convincing proof for many of the bold statements and “grand” properties attributed to these droplets is lacking. This makes it hard to value the true relevance or contribution to the origin of cellular life or even LUCA. If the required characterization and evidence could be provided and all other problems would be addressed, I would certainly be favourable of reconsidering this work for publication in Nature Communications, since the potential significance is quite high. However, in the current form, I do not recommend publication.

1. The reaction described in Figure 2a produces significant amounts of droplets, which show fascinating behaviour in the rest of the manuscript, yet the droplets are not characterized. The authors later speculate that the droplets have a spatial heterogeneity (Figure 5), but without evidence. It seems (page 16, line 16) that the authors assume (and even conclude on page 15, line 19-20) the droplet is made of BnSH, stabilized with an oligopeptide shell, but there is no direct evidence for this. However, knowing the composition of these droplets is in my opinion essential for a proper understanding of the experiments presented here, and the authors should substantiate their assumptions by collecting the droplets by centrifugation and doing LC-MS analysis of the separated phases (supernatant and droplet) with the right controls to determine the droplet composition. It would be more convincing even if to simply reproduce the droplets by mixing these known concentrations of BnSH and oligocysteine (the authors could use for instance the trimer, which they find predominantly).

Answer

The heterogeneity in the droplet was directly visualized by Raman microscopy, whereas it was difficult

to measure the droplet by LC mass spectrometry due to the presence of disulfide. The results of Raman microscopy have been added to Figs.A1 and S15 in Supplemental Information. We showed in a revised Fig.S9 the result that the droplets were reconstructed by mixing a known concentration of BnSH with oligocysteine.

Figure A1. Raman spectroscopy of droplet dispersion. a, Optical microscopy image of a LLPS droplets. The scale bar represents 20 μm . b, Raman microscopy image of the yellow rectangle in Fig. S15a. c, Raman spectra of LLPS droplets. The red line is the central part of the droplet (red rectangle in Fig.S15b). The blue line is the peripheral parts of the droplet (blue rectangle in Fig.S15b). In order to investigate the ratio of hydrophobic and hydrophilic components in both the central and peripheral parts of the LLPS droplets, the Raman spectra of each part were measured by laser Raman microscopy. The benzene ring-derived peak of benzyl mercaptan ($\sim 1000\text{ cm}^{-1}$) as the hydrophobic component and the water-derived peak ($\sim 3400\text{ cm}^{-1}$) as the hydrophilic component were detected and the benzene ring/water ratio (B/W) was compared at each part. Mpre (2.5 mg) and DTT (2 mg) were mixed, and then 100 μL of MilliQ water was added immediately. The mixture was stirred with vortex for 15 seconds. After 1 hour of stirring, the droplets were observed with a laser Raman microscope. The average value was calculated from the spectral data obtained at the four different points of the central parts of the droplets and the baseline-corrected spectrum were shown in Fig.S15 c (red line). Similarly, the spectrum from the peripheral parts were shown in Fig.S15c (blue line). Based on the obtained spectral data, the B/W value at the central part was calculated to be 0.370 ± 0.016 , whereas that at the peripheral part was calculated to be 0.288 ± 0.038 . This result indicates that the central part is more hydrophobic than the peripheral part in the droplets.

2. These type of reactions and assembly processes are usually quite sensitive to pH, yet pH or control of the pH is not mentioned anywhere in the paper. I would guess that the pH is quite low, but it would be useful to verify this and to report how stable the pH is over the course of the growth and self-reproduction experiments. The authors report a control experiment in which BnSH, cysteine dihydrochloride and DTT mixed together create solid precipitates (Fig. S4b), which seems strange. I was not able to reproduce these findings. Is it possible that the cystine was not properly dissolved first before mixing? Was the mixing order relevant? In the system shown in Fig. 2, the DTT first reacts with the disulfide precursor, and BnSH is produced only later. The description in the main text (page 6, line 14) is ambiguous: there was no turbidity, but aggregates were found to precipitate.

Answer

Identification of solid precipitates in mixed solution

We again dissolved cystine dihydrochloride, DTT, and BnSH in water, and then observed them. The optical micrographs are shown in Figure A2 (all scale bars represent 50 μm). When mixing the three molecules, it is important to keep DTT and cystine close together in the vial and BnSH a little apart from them. We added water to the reagent immediately, it becomes cloudy. If all three reagents are dissolved in water at the same time, then the mixture would become cloudy. Since this experiment of Figure A2a is a control experiment for Figure 3a, the samples must be mixed at the same time. Figure A2a shows the sample 5 minutes after mixing and Figure A2b shows the sample 12 hours after mixing. Again, we found undissolved solid precipitates in the prepared specimen. Even 12 hours after mixing, the solids not droplets were still precipitated. Indeed, immediately after mixing, we could observe by the naked eye as white and cloudy in the vial. However, after 5 minutes, precipitation had occurred and the water on the top of the vial was colorless and transparent. This suggests that the droplets were not dispersed, but white microcrystals were floating within 5 minutes. A microscopy image of only cystine dihydrochloride is shown in Figure A2c. From this photograph, we concluded that cystine only did not dissolve, but was precipitated as a microcrystals, and that the cloudy white in the vial was caused by this microcrystals.

Figure A2. Microscopy images of mixture solution of BnSH, cystine and DTT. (a) 5 min after mixing, (b) 12 after mixing, (c) cystine dihydrochloride. All scale bars represent 50 μm .

The effects of pH and ion concentration on the formation of droplets

We confirmed the effects of pH and ion concentration on the formation of droplets. We added **Mpre** (10 mM) and DTT (25 mM) to the each aqueous solution with various pH values and ion concentration. Figure A9 shows the change in turbidity in the vial: (a) 5 min after mixing, (b) 30 min after, (c) 1.5 h after, (d) 3 h after, (e) 12 h after, respectively. Each aqueous solution was prepared containing (i) hydrochloric acid (1 mM, pH=3), (ii) MilliQ water (pH=7), (iii) sodium chloride (1 mM, pH=7), and (iv) sodium hydroxide (1 mM, pH=11). The pH values above are the values of the prepared aqueous solutions. The vials in each photo (Figure A3a-f) are in the order of (i) to (iv) from left to right. First, the experimental results showed that the salt concentration had no particular effect on the results. Secondly, for the first 30 minutes, droplet formation was more active at higher pH values, but after 1.5 hours, droplets are formed at all aqueous pH values, and the effect of the pH change was indistinguishable.

Figure A3. The photographs of the solution of Mpre (10 mM) and DTT (25 mM) with various pH values and ion concentration. (a) 5 min after mixing, (b) 30 min after, (c) 1.5 h after, (d) 3 h after, (e) 12 h after, respectively. Each aqueous solution was prepared containing (i) hydrochloric acid (1 mM, pH=3), (ii) MilliQ water (pH=7), (iii) sodium chloride (1 mM, pH=7), and (iv) sodium hydroxide (1 mM, pH=11).

In addition, microscopy images are shown in Figure A4. This observation results shows that droplets can be formed regardless of the pH value. The scale bars indicate 50 μm .

Figure A4. The microscopy photographs of the solution The mixture solution of **Mpre** (10 mM) and DTT (10 mM) with various pH values and ion concentration.

3. The authors regularly use self-important terminology to describe their system, including autocatalytic, proliferation, self-reproduction, growth-division, higher-order function. Most of these terms have a defined meaning, but evidence that they apply to the current system is lacking. In an autocatalytic system, for example, a product enhances the rate of a process/reaction in which it is being formed. This can be tested by seeding the initial mixture with some of the product, or by fitting a series of kinetic experiments to the appropriate autocatalytic rate equation. The green line in Fig. 3c is hardly convincing evidence of an autocatalytic process. It is difficult to see in the lowquality figure without data markers, but it seems that it has been attributed a sigmoidal shape based on a single data point. Additional kinetic traces are missing and the existing traces are not fitted to show that an autocatalytic reaction gives a significantly better fit than, for example, a simpler first-order decay.

Answer

We clarified the definitions or added explanations to the words and phrases we had used. We believe this manipulation make it easier to read.

Since the system is physical autocatalytic reaction, which was proposed by Fletcher, with the droplet as the reaction field, the equilibrium of the reaction may be shifted to the right when the product is removed from the reaction field by LLPS environment and may not meet the criteria for an autocatalytic reaction. Here, we confirmed whether this reaction is autocatalytic or not from the change of the chemical shift of the benzene ring proton in the M_{pre} with respect to the density of the extruded aggregates. Since the reaction was fitted on the sigmoidal curve (Figure A5), we concluded that this

reaction was an autocatalytic reaction.

Iterative algorithm	Levenberg-Marquardt Method
Formula	$y = a * (1 + (d-1) * \exp(-k*(x-xc)))^{-1/(1-d)}$
a	69.70121 ± 0.66271
xc	77.36148 ± 1.94612
d	0.46532 ± 0.0266
k	0.00822 ± 2.50981E-4
Reduced Chi-Sqr	0.40012
R-Square (COD)	0.99987
Adj. R-Square	0.99985

Figure A5. Reaction progress from NMR measurement and the formula of sigmoidal curve. Reaction progress was estimated from the chemical shift of the benzene ring proton in the M_{pre} .

Raman microscopy (Figure A1) shows that BnSH is also localized in the peripheral part of the droplet although the central part of the droplet is hydrophobic. This localization concentrates the monomer at the interface and subsequently accelerates the amide formation. This acceleration of the reaction results in rapid formation of the interface of the droplet, and thus the reaction is considered to be autocatalytic.

4. The same can be said of other terms used by the authors, such as self-reproduction. The conclusion based on DLS size distributions is not straightforward for several reasons. First of all, the large droplets contribute disproportionately much to the scattered intensity, and it is not unlikely that very small droplets are not detected with the automatic fitting procedure. Did the authors do a control in which they mixed equal amounts of a mixture of a few minutes old and one of several hours, to show that the distinct populations of droplets can be detected? Another concern with this method is the fact that additional processes leading to increase in droplet size also play a role in the background. True liquid phase separated droplets also coarsen over time by ripening and coalescence (accelerated by flow/shaking, see review by Berry et al. 2018), and dispersions settle under the influence of gravity, which is also dependent on droplet size. All these processes may lead to a gradual increase in apparent size over time. Control experiments are missing for a passive system (e.g., without addition of precursor and DTT), and with intermediate homogenization.

Answer

We observed the fusion of droplets under a microscope and specified the results in the text (P.6, L.17). In addition, self-production of droplets was supported by the DLS size experiments that the droplets originated from scratch and no new distribution of droplets at the second addition of **Mpre** was detected. However, since the noted sentence was at a stage prior to the description of those results, we revised the sentence as follows: “*After the first addition of **Mpre**, the particle size distribution is expected to fully shift to the right, due to the growth*” (P.11, L.14). The shape of the actual particle size distribution histogram was shifted towards the larger size with repeated oscillations (Figure 4c). This result also strongly supports the repeated self-production and fusion of droplets. To validate the reviewer's proposed conditions, we added water without **Mpre** to the extruded droplet dispersion. No significant droplet formation was observed (Figure A6). For these reasons, fusion alone cannot explain the autocatalytic process.

Figure A6. a) Comparison of droplet dispersion in vial at 0 and 24 h. b) Comparison of phase contrast microscopy images of droplet dispersion at 0 and 24 h. Scale bars represent 50 μm . To show that the increase in size of the droplets is not a growth due to fusion, we performed the experiment that droplets does not increase in size in the case of droplet only. A droplet dispersion was prepared 24 h after the addition of Mpre (5 mg) and DTT (4 mg) to water (1 mL). The droplet dispersion was filtered through a 100 nm pore size membrane filter, and the solutions were compared between immediately and 24 h after the addition of an equal amount of water.

5. It is not helpful that the corresponding figure (Fig.3) is highly confusing. The x-axis seems to have the wrong scale (these droplets are probably not 500 micrometer large – see Fig.2), and the y-axis displays normalized number, but it is not clear what this is normalized with and how normalization has been done (all traces seem to have different peak heights and areas under the curve).

Answer

To begin with, a result of DLS particle size distribution measurements and that of optical microscopy observation are not inconsistent. The reason for this is as follows: In principle, whereas DLS measurements are performed on molecular aggregates that are capable of Brownian motion, microscopy observation targets large-sized aggregates that have precipitated without Brownian motion. Molecular aggregates targeted by the DLS particle measurements are often dispersed upward in prepared specimens under microscopy observations. In addition, the floating molecular aggregates smaller than 1 micrometer cannot be observed in principle of optical microscopy. Actually, the microscopy images photographed while changing focus along the z-axis are shown in Figure A7. Indeed, these microscopy images prove that the view is different between above and below part along z-axis.

Figure A7. Microscopy images of the droplets. From the top left photo to the bottom right photo in this figure, the droplets were observed from the bottom of the prepared specimen to the top at every 30 μm . As shown in the above results, particle sizes larger than 3 μm are rare and are only present at the bottom of the specimen glass. Therefore, most of the droplets are smaller than 3 μm , or so small that they cannot be observed under the microscope. The scale bars represent 50 μm .

In addition, we revised the Fig.4b and 4c in order for readers to understand. The new vertical axis in the Fig. 4b and 4c in the revised manuscript shows the percentage of the number of the observed droplets at the each particle size, where the total number of droplets in the sample is set to 100(%). The results and our claim haven't changed.

6. The growth-division cycle shown in Fig. 4d-e are also not conclusive. Division suggests that the number of droplets increases over time, but there is no evidence to show this. The trace in Fig. 4e shows a consistent decrease in size about 3-4 data points after each new addition of precursor, but no explanation is given. This seems to be in conflict with the continuous increase in size shown in Fig. 4c (over 16 h!) and it seems to contradict the conclusion that there is growth. Control experiments without feeding are missing and also here the y-axis seems incorrect.

Answer

Figs. 4a-c refers to a series of experiments. Fig. 4b corresponds to the first step in Fig. 4e, and shows the DLS measurement when the first addition of \mathbf{M}_{pre} to the water. Fig. 4c shows the change in particle size when \mathbf{M}_{pre} was added to preexisting droplets, and the preexisting droplets were not extruded. The experiment in Fig. 4c verifies the passage of droplet formation by the addition of \mathbf{M}_{pre} , which was shown in Fig. 4a. On the other hand, Fig. 4e shows an experiment in which the addition of \mathbf{M}_{pre} and extrusion were repeated to confirm the recursiveness of the droplet size when \mathbf{M}_{pre} was added to the droplets. The extrusion process was performed because it was necessary to keep the size of droplets the same each time. Therefore, the experiment in Fig.4c and that in Fig. 4e are completely independent. To validate the reviewer's proposed conditions, we added water without \mathbf{M}_{pre} to the extruded droplet dispersion. No significant droplet formation was observed (Figure A6).

7. Finally, the title is overly pretentious. This manuscript shows absolutely no link to biology, everything in here is based on principles of (physical) chemistry.

Answer

We revised the title as follows: *Proliferating Droplet Emerged by Prebiotic Polymerisation; the Missing Link between Chemistry and Biology on the Origin of Life*. To further clarify our claim for "chemistry" and "biology", we inserted the words "chemistry" and "biology" into the abstract text (P.1, L.15).

8. The paragraph on page 20, starting line 16 is purely speculative and does not serve a clear goal. This manuscript provides no evidence for any of these points, and if this paragraph is retained, the speculative nature should be emphasized.

Answer

We added a sentence " *The nucleic acid enrichment of this study may provide a useful roadmap for solving some of the major problems associated with the RNA world hypothesis, e.g., the concentration and condensation reactions of nucleic acid substrates or oligonucleotides against the diffusion of the molecules to a vast expanse of ocean*⁴³." (P.22, L.18) at the beginning of the paragraph to make it clear that the content of this paragraph is speculative conclusion.

9. All references to works by the groups of Boekhoven, Cronin and Huck, on active/dissipative droplets, their life-like behavior and their potential roles in the origin of life are missing. Some of these would certainly be relevant to put the current work in context.

Answer

We cited the papers of the above groups. The sentence is as follows: "LLPS droplets carrying genetic information could generated the LUCA by combining self-proliferation with advanced phenomena⁴⁶ such as self-propulsion^{47,48}, active/dissipative droplets⁴⁹⁻⁵¹, droplet-droplet communication¹⁴ and competition¹⁵."

10. The quality of many graphs is very poor. It was impossible to see for example the open squares and crosses (described on page 6, lines 11-13). One of these should have half the turbidity of the mixture solution, but there is no line in Fig. 2c at half the maximum turbidity.

Answer

As the reviewer pointed out, the figure was revised. In the previous manuscript, we had a mistake in the comments about the Fig. 2c. There were no white squares and crosses in the Fig2c. We revised the full circles, open squares, and crosses in the manuscript to the red line (P.5, L20), green line (P.6, L15), and blue line (P.6, L17) respectively. In addition, we deleted the sentence in the previous manuscript "*which is about half of the mixture solution of Mpre and DTT*". (P6, L15)

11. Fig. 5b-f: something is wrong with the scale bars here. The droplet in 5b is larger than 100 micrometer according to the scale bar, but the cross sections in 5c,e are only 35-40 micrometer.

Answer

In the photographs from the top row (DIC) to the 4th row (Merged) shown in Fig. 5b, the scale bar represents 100 micrometers. On the other hand, the scale bar in the bottom photograph (Magnified) of Fig. 5b is 20 micrometers, because the bottom row photographs are magnified ones of the 4th row photographs. Since the scale bar in the bottom photographs are 20 micrometers as described in the figure caption, the droplet size is consistent with that in Figs 5c and 5e. However, it was difficult for readers to identify that the photographs at the bottom row were magnified, so we modified the Fig. 5b for clarity.

Reviewer #3 (Remarks to the Author):

In this manuscript, the authors demonstrate the assembly of droplets using a peptide synthesis system from thioester derivatized amino acids under the same conditions. These oligopeptide droplets can grow over time, eventually catalyzing their own synthesis and growth autocatalytically. Repeated cycles of growth and division through dilution were demonstrated upon addition of an added physical stimulus, and the droplets appeared to be able to localize nucleic acids and lipids. In fact, lipid uptake decreased the size of the droplets, while nucleic acid uptake helped to prevent this dissolution. While this study is interesting and the experiments are relevant, there are a number of major and minor points that I have found related to the claims, background, and some experiments that preclude it from being published in its current form. Specifically, there is a general lack of information and discussion related to the prebiotic relevance of the system, including almost no discussion on primitive geological environments and processes as well as only minor discussion on just a few origins of life scenarios; the prebiotic relevance of the molecules chosen, their synthesis, the structures formed, and their resulting functions should be discussed in greater detail for it to have the amount of impact it deserves. I have listed these points below, and should the authors adequately address these points, I would recommend this manuscript for publication.

I would also highly recommend that the authors ask one of their colleagues or collaborators, who is a native English speaker, to carefully proofread their manuscript. There are also paid services available for this. As of right now, there are a large number of spelling and grammatical errors that should be fixed in order for the paper to be published. A few representative errors from the first few pages are listed in this review. I would recommend all of the errors be fixed before publication.

1. The authors assume that “chemistry” means polymer generation, and “biology” means self-assembly and function. However, this is not necessarily the case, as many researchers would consider all of these (polymer generation, self-assembly, and function) to be “chemistry”. The authors should clarify specifically what they mean by “chemistry” and “biology”.

Answer

We revised the title as follows: *Proliferating Droplet Emerged by Prebiotic Polymerisation; the Missing Link between Chemistry and Biology on the Origin of Life*. To further clarify our claim for "chemistry" and "biology", we inserted the words "chemistry" and "biology" into the abstract text (P.1, L.15).

2. Because of the issue with definitions in point 1, the title and the abstract are too ambiguous with respect to linking “chemistry” and “biology”. Perhaps it would be much clearer if these terms could be replaced with more specific concepts such as “generation” and “self-

assembly/function". This way, a potential reader who only reads the title can understand immediately without having to read much further down for a specific definition.

Answer

We answered on Answer 1.

3. There is generally a lack of tight transitions from one idea point to another in terms of writing style. At points, it feels like the manuscript is simply listing things in logical order (and to be fair, the order presented is very logical), but without the proper logical transitions. I think the manuscript would benefit greatly from insertion of proper and tighter transitions between ideas, both at the paragraph level, as well as the sentence/statement level. For example, the two full sentences in page 3, lines 13-17 are logical in order, but there is no logical transition word or statement. Something as simple as, "Although... (synthesis satisfied in certain conditions)...however...(assembly not satisfied in same conditions)..." could help greatly. I would extend this statement to the rest of the manuscript as well.

Answer

Thank you for your pointing out. We asked the native speakers to check the structure of the revised text. We believe the logical structure could be emphasized and improved.

4. Page 3, line 18. What is the purpose and significance of this study? In this sentence, which is the thesis of the study, only "what" and "how" were listed. "Why" was not stated, and that is an important point that must be included, especially in relation to the previous paragraph, where "why" is framed. If "why" can be included in this sentence, then the reader can immediately understand the entire point of the paper very quickly.

Answer

To clarify the purpose of this study, we added the phrase: "*to realize the emergence of a proliferating droplet with prebiotic polymerisation*" (P.3, L.18).

5. In the introduction, it is unclear what the "autocatalytic" feature of the system is. How does feeding monomers into the system eventually catalyze its division? That is unclear, and it should be discussed in slightly more detail. In Figure 1, physical stimulus is mentioned, and autocatalytic growth is discussed in the results, but it is not mentioned in the introduction.

Answer

It is clear that autocatalytic reactions are important for the evolution of primitive life. In order to evolve and survive, the primitive life needs to proliferate, it would be the only way that they could proliferate acceleratively as a single species.

6. Can the entire system be considered to be undergoing autocatalysis? While the growth is likely autocatalytic, I am not sure about how exactly the division/proliferation step could be considered autocatalytic. You may want to include some discussion about the definition of autocatalysis and why you believe this system as a whole satisfies those definitions, not just the growth aspect. In my opinion, the division step is not autocatalytic, because a physical stimulus is necessary. If you can argue that physical stimulus is necessary for the growth as well, or that the autocatalytic synthesis affects the division, then autocatalysis can be argued. However, as it stands, it is simply a system which grows autocatalytically, but which can exhibit self-replication through possibly non-autocatalytic means. However, if this assumption is incorrect, please explain why.

Answer

We clarified the definitions or added explanations to the words and phrases we had used. We believe this manipulation make it easier to read.

Since the system is physical autocatalytic reaction, which was proposed by Fletcher, with the droplet as the reaction field, the equilibrium of the reaction may be shifted to the right when the product is removed from the reaction field by LLPS environment and may not meet the criteria for an autocatalytic reaction. Here, we confirmed whether this reaction is autocatalytic or not from the change of the chemical shift of the benzene ring proton in the M_{pre} with respect to the density of the extruded aggregates. Since the reaction was fitted on the sigmoidal curve (Figure A1), we concluded that this reaction was an autocatalytic reaction.

Iterative algorithm	Levenberg-Marquardt Method
Formula	$y = a * (1 + (d-1) * \exp(-k * (x-xc)))^{-1/(1-d)}$
a	69.70121 ± 0.66271
xc	77.36148 ± 1.94612
d	0.46532 ± 0.0266
k	0.00822 ± 2.50981E-4
Reduced Chi-Sqr	0.40012
R-Square (COD)	0.99987
Adj. R-Square	0.99985

Figure A1. Reaction progress from NMR measurement and the formula of sigmoidal curve. Reaction progress was estimated from the chemical shift of the benzene ring proton in the M_{pre} .

7. There is no discussion as to the prebiotic relevance or synthesis of nucleic acids or lipids (and which within these categories are more prebiotically plausible, such as fatty acids), even though the authors incorporate these other biomolecules into the system. The authors should discuss this. Additionally, the authors should discuss whether the concentrations of monomers and other biomolecules used in the experiments, and the experimental conditions, are prebiotically plausible, and if not, why this system is still prebiotically relevant.

Answer

We answered on Answers 15 and 29.

8. The monomer used is likely not to have been prebiotically relevant, especially considering the BnSH group that was incorporated. In fact, the word used in Page 5, line 3 is “designed”, which by definition makes the monomer not prebiotic. They authors also don’t discuss the prebiotic relevance of the system. While it is understandable that the authors could have used this system as a modern model system to study a prebiotic process, that is not discussed at all. This discussion must be added, otherwise it is not clear to the readers that this system is relevant to prebiotic chemistry.

Answer

We answered on Answers 15 and 29.

9. One of the major features of this system is that reduction of M is required to proceed with the reaction. Is synthesis of M plausible under the reducing atmospheric conditions on early Earth? And even if M could be produced, how can these reducing conditions on early Earth directly

lead to synthesis and self-assembly. I think that given the correct discussion and argument, by incorporating the importance of reducing conditions into this system could greatly increase the relevance and significance of this system with respect to prebiotic chemistry.

Answer

We answered on Answers 15 and 29.

10. In the dilution-extrusion system, there is no mention of how this simulates a primitive geological environment. The authors should discuss which geological and environmental features and processes on early Earth could provide the dilution, shear, and nutrient import needed for this process. The authors should also discuss how this process can be considered autocatalytic, if the system requires external stimuli.

Answer

We answered on Answer 15.

11. In the section “Nucleic acid/lipid concentration in droplets” there is a discussion of nucleic acids, lipids, and peptides being the most important biomolecules in modern biology. However, the next statement quickly states that these must be incorporated into the current system to be relevant to origins of life studies. First of all, what is the rationale behind this? Just because something appears in modern biology, absolutely does not mean that it was required during the origin of life, as discussed by theories relating to non-biological origin of life (Chandru, K.; Mamajanov, I.; Cleaves, H.J., II; Jia, T.Z. Polyesters as a Model System for Building Primitive Biologies from Non-Biological Prebiotic Chemistry. *Life* 2020, 10, 6.) and the shadow biosphere (Davies PC, Benner SA, Cleland CE, Lineweaver CH, McKay CP, Wolfe-Simon F. Signatures of a shadow biosphere, *Astrobiology* 2009, 9, 241). I believe that the authors should qualify their assumptions that modern biomolecules can be used realistically in prebiotic chemistry experiments.

Answer

According to Jia et al [*Life*, 2020; 10(1)], they pointed out the possibility that a system of non-biological molecules could have been the origin of life. This is a very stimulating idea. On the other hand, RNA, peptides, and phospholipids are the common biomolecules on the Earth, and an early life would have already possessed these materials and self-sustained.

12. In addition to the previous point, assuming that a valid reasoning is included for the above point, the authors should differentiate between which classes of lipids, peptides, and nucleic acids could be considered prebiotic, prebiotic model systems, or not prebiotic. This is important because only certain peptides have been shown to have a relevant prebiotic synthesis, fatty

acids/acyls are much more prebiotically plausible than phospholipids, and shorter polymers are much more relevant than longer polymers. The fluorescent tags on the biomolecules will also contribute to their ability to be incorporated within the droplets, and this possible contribution (or if there is no contribution) should be discussed. Of course, the fluorescent tags are also not prebiotic, so the authors should validate their incorporation into the system. Were experiments performed with just the dyes themselves?

Answer

We inserted the following sentence: *To show that this experiment was not affected by the fluorophore itself, it was confirmed that the results were similar for experiments labeled with different fluorophores.*(P.15, L.13). See also Figure S14. In additional experiments, Raman microscopy (Figure A2) shows that BnSH is also localized in the peripheral part of the droplet although the central part of the droplet is hydrophobic. This localization concentrates the monomer at the interface and subsequently accelerates the amide formation. This acceleration of the reaction results in rapid formation of the interface of the droplet, and thus the reaction is considered to be autocatalytic.

Figure A2. Raman spectroscopy of droplet dispersion. a, Optical microscopy image of a LLPS droplets. The scale bar represents 20 μm . b, Raman microscopy image of the yellow rectangle in Fig. S15a. c, Raman spectra of LLPS droplets. The red line is the central part of the droplet (red rectangle in Fig.S15b). The blue line is the peripheral parts of the droplet (blue rectangle in Fig.S15b). In order to investigate the ratio of hydrophobic and hydrophilic components in both the central and peripheral parts of the LLPS droplets, the Raman spectra of each part were measured by laser Raman microscopy. The benzene ring-derived peak of benzyl mercaptan ($\sim 1000\text{ cm}^{-1}$) as the hydrophobic component and the water-derived peak ($\sim 3400\text{ cm}^{-1}$) as the hydrophilic component were detected and the benzene ring/water ratio (B/W) was compared at each part. Mpre (2.5 mg) and DTT (2 mg) were mixed, and then 100 μL of MilliQ water was added immediately. The mixture was stirred with vortex for 15

seconds. After 1 hour of stirring, the droplets were observed with a laser Raman microscope. The average value was calculated from the spectral data obtained at the four different points of the central parts of the droplets and the baseline-corrected spectrum were shown in Fig.S15 c (red line). Similarly, the spectrum from the peripheral parts were shown in Fig.S15c (blue line). Based on the obtained spectral data, the B/W value at the central part was calculated to be 0.370 ± 0.016 , whereas that at the peripheral part was calculated to be 0.288 ± 0.038 . This result indicates that the central part is more hydrophobic than the peripheral part in the droplets.

To investigate the localization of the hydrophilic DNA without any probes, we added 20mer DNA to the droplet dispersion and measured the droplets after 1 hour by Raman spectrometry. The results are shown in Figure A3. Focusing on the DNA's phosphate peaks (784cm^{-1} , 830cm^{-1} , 1090cm^{-1} [ref. *RSC Adv.*, 2018, 8, 25888]), it was found that there was slightly more DNA in the peripheral part in the droplet. However, due to the large peaks of the droplets and the possibility of DNA damage by laser irradiation, the difference was not as clear as in the fluorescence microscope image (Fig.5g). These results suggest that nucleic acids were incorporated into droplets depending on compositional heterogeneity of a droplet, not depending on the presence of probe.

Figure A3. Raman spectra of the droplets 1 h after addition of DNA. red: the central part of the droplet, blue: the peripheral part of the droplet. The peaks derived from DNA indicate that DNA present in peripheral part.

13. Page 17, line 12-14. I don't think the claim that the RNA aggregating to the outside helped to protect droplets from size decrease from lipid addition is shown robustly by the data. Perhaps nucleic acid uptake contributed or was the main mechanism of action, but aggregation to the outside being the main mechanism of action is definitely not proven by the data. I think to robustly claim this, you must demonstrate another system, in addition to the lipids, in which

addition of nucleic acids helps the system to avoid decrease in size. A control sample containing lipid and another molecule or biomolecule should also be tested to show that the size protection ability is conferred only by nucleic acids. It may be that incorporation of any molecule that localizes to the droplet boundary can protect it from being made smaller by lipids. Is it dependent on the size of the nucleic acid? RNA and DNA? Duplex single strand? Absence of fluorescent probe? Can nucleotides have the same effect? Additionally, what is the mechanism by which the lipid decreases the size of the droplet, and how exactly does the RNA on the surface prevent this? This mechanism is key to this claim, and should be shown or discussed.

Answer

Thanks for your interesting point. We'll try to perform an information delivery experiment and discuss it in our next paper. Since nucleic acids, peptides, and lipids have been shown to proliferate autocatalytically, we need to pay attention to them in order to realize the origin of life in the laboratory.

14. Page 17, line 8. It appears that the TR-RNA, with lipid present, enters the apparent BnSH center. However, in the absence of lipid, TR-RNA only resides on the outside. Is there a specific reason for this or experiments showing why this is the case? There may be some unknown lipid action that has not been probed, and this should be discussed.

Answer

It is possible that lipids and nucleic acids interacted with each other to form a lipoplex, and that lipoplex allowed nucleic acids to enter into the center of the droplet. The combination of other lipids and nucleic acids is an issue for future research. As a reference, we performed a following experiment that DNA with fluorescent intercalators was incorporated in the droplets. The fluorescent-labeled DNA gradually entered the droplet and saturated after 6 hours (Figure A4).

Figure A4. (a) Fluorescence microscopy image of the droplet containing DNA with fluorescent probe and (b) the line profile of the microscopy image.

15. Page 19, line 4. Again, the “mild prebiotic conditions” were not discussed. What was the pH of the solution and how does that compare to the proposed pH of primitive ocean waters? Was this study done in an oceanic environment, or a freshwater environment. In addition, no discussion of dehydration/rehydration cycles were included, even though there were some hydration processes (for dilution) used in the system. These are the discussions that should be included throughout the paper in order to claim that this system is without a doubt prebiotically relevant.

Answer

Our system is assumed not to be a high-temperature, high-pressure and alkaline environment like hydrothermal vents, but to be 0-100 °C, 3 < pH < 10 and 0-1 atm. In other words, we assume that our system is like a puddle of water and its surroundings, such as a hot spring or geyser, that allows water to exist as a liquid at the surface of the earth. We also assumed that the system will reproduce by using a steady supply of precursors and water in such an environment. In fact, this environment falls into the Isua region in Greenland. In Isua's basalt (pillow lava), which actually appears to be 3.8 billion years old, traces of sulfides and organics as well as water traces have been found.

16. Page 20, line 15. Diffusion of components away from the system is not the main unsolved problem of the RNA world theory. There are many issues, including reactivation of spent monomers, spent monomer inhibition of replication, synthesis of RNA monomers in a complex chemical milieu, replication and catalysis of RNA in messy systems, fast hydrolysis of RNA and monomers, the “reannealing problem”, among others, each of which have partial solutions, but not full solutions. While increasing concentration of such nucleic acids is certainly helpful to achieve some of the RNA world scenarios, there are still many more pressing problems that must be solved, which are being tackled by Szostak, Sutherland, Powner, Brenner, Hud, and many others in the RNA origins field. I think a deeper understanding and discussion of this field, and the recent research, should be achieved and included, respectively, rather than a broad generalization of how solving one problem could tie this system to an RNA world origins model.

Answer

Certainly, as the reviewer said, there are some major problems with the RNA world hypothesis. Thank you for showing us the problems that need to be solved. We have rephrased “the main problem” to “one big issue”.(P.22, L.11)

17. Page 20, line 16 and page 21, line 4. A simple fluorescence recovery after photobleaching (FRAP) experiment should be performed to robustly show that whether the RNA is stably compartmentalized within the droplets without diffusion. This would also show whether the information from the RNA could be delivered to other droplets in a robust manner, without such

diffusion into the bulk. If large amounts of diffusion occur, then it is likely that genetic exchange occurs even in the absence of self-replication.

Answer

In this series of observations, we did not find a sharp decrease in fluorescence derived from RNA, i.e. leakage. We will investigate the behavior of RNA in the droplet in more detail as future work from the perspective of genetic information delivery. Thanks for the suggestion of a useful method.

18. The conclusion does not talk at all about the “missing link between biology and chemistry”. In fact, after the introduction (where it is mentioned in passing), this point is not discussed at all, although it is in the title. Either the title should be amended, or, this point should be discussed in much greater detail in both the introduction and the discussion.

Answer

We answered on Answer 1.

19. Page 2, line 9. Is a protocell required just for exploration of the origin of life (this implies that it is not a real structure and is only used in lab simulations of origins of life studies)? Or is it by definition required for life? I think this should be clarified.

Answer

We rewrote the part of the sentence: “*is considered an essential step in exploring the origin of life or synthesising life*” (P.2, L.9)

20. Page 2, line 10. Some, like Szostak and Deamer (among others), have suggested that a protocell is a membrane-bound minimal vesicle. We don’t know what is the “real” protocell, so it would be wise to include some background on non-coacervate droplet-based protocell models before assuming that CDs (especially from LLPS) are the globally accepted protocell model. This is but one of many theories.

Answer

Thank you for your valuable opinion.

We described the advantages and features of coacervates while referring vesicle-based protocells in the latter part.

21. Page 2, line 10. The sentence starting with “A CD...” should have at least one reference.

Answer

We properly cited the same review as ref.5 in this passage as well.

Ref.5 Koga, S., Williams, D., Perriman, A. et al. Peptide–nucleotide microdroplets as a step towards a membrane-free protocell model. *Nature Chem* 3, 720–724 (2011).

22. Page 2, line 15. Although the authors discuss various biomolecular “world” scenarios, it would be relevant to also discuss non-biomolecular or composomic “world” scenarios, as proposed by Chandru, et al. (Chandru, K.; Mamajanov, I.; Cleaves, H.J., II; Jia, T.Z. Polyesters as a Model System for Building Primitive Biologies from Non-Biological Prebiotic Chemistry. *Life* 2020, 10, 6.) and Segré, et al. (Segré, D.; Ben-Eli, D.; Lancet, D. Compositional genomes: Prebiotic information transfer in mutually catalytic noncovalent assemblies. *PNAS* 2010, 97(8), 4112), respectively.

Answer

Thank you for your valuable input. We certainly think that the above researches are important for us as well. We will make use of the reviewer's comment in our future study.

23. Page 2, line 22. I would not argue that Mann was the main pioneer of CDs in origins of life research. He is one of many who have been pioneering this field, including Keating, Bevilacqua, Boekhoven and Tang.

Answer

Since “pioneering” was an overstatement, we rephrased to “novel”. (P.3, L.3) We also cited the work of Keating, Bevilacqua, Boekhoven and Tang.

24. Page 3, line 4. How about the study by Yin et al. (Yin, Y., Niu, L., Zhu, X. et al. Non-equilibrium behaviour in coacervate-based protocells under electric-field-induced excitation. *Nat Commun* 7, 10658 (2016)), where they demonstrate budding and growth due to changes in electric field stimuli?

Answer

Thank you for introducing the interesting research. We cited the recommended paper and revised the sentence as follows: "*A previous study on fusion and division of molecular assemblies have reported the importance of non-equilibrium states²³. However, few studies have reported recursive self-reproduction of supramolecular assemblies under periodic stimuli²⁴ or growth and division using physical stimuli, because metastable assemblies tend to move toward equilibrium.*" (P.3, L.3-6)

25. Page 3, line 14. What about the conditions where there is an unlimited reservoir of peptides available (and were synthesized in conditions before the assembly/reproduction conditions)? Is such a condition possible?

Answer

Formation and maintenance of the droplet can be achieved with continuous supply of peptides and

physical stimuli. However, self-production, which is one of the greatest features in living things, cannot be achieved only by these stimuli.

26. Page 3, paragraph beginning with line 18. What is the significance of thioesters in prebiotic chemistry? Are they themselves prebiotic? Is their synthesis and conjugation prebiotic? Why were they believed to have been the most relevant system by de Duve? These questions should be answered to frame the significance of this theory in the global framework.

Answer

The pathway of peptide formation from amino acids on the primitive earth remains a mystery. One solution to this problem has been a peptide production from amino acid thioesters, and Powner et al. have reported a new thioester-mediated peptide synthesis. The fact that thioesters are also used as a common energy currency in modern organisms is evidence that thioesters played a central role in metabolism at a very early stage of life.

27. Page 5, line 3. How do the monomers synthesize peptides? Do you mean that they can be synthesized into peptides? This is a major difference in nomenclature, as the word “synthesize” assumes that the monomers in some way are affecting their own synthesis. Is this true? This should either be changed, or explained.

Answer

As the reviewer pointed out, this sentence was a misleading statement. We revised it as follows: “*We designed a monomer **M** which was capable of producing peptides and facilitating self-molecular assemblies under the mild aqueous condition which allows formation of droplets and self-reproduction of its building blocks.*”(P.5, L.6-8)

28. Page 5, line 7. The authors should explain why this molecule is “activated”, and why activated monomers are essential to the process. In addition, is the precursor prebiotically relevant? Was the synthesis prebiotic?

Answer

We deleted the misleading word "activated", because the monomers were reactive not activated. Although we should prepare monomers in theory, sulfhydryl group of the monomers (**M** in Figure 2a) are easily oxidized in the laboratory. Therefore, we designed and synthesized the oxidized molecule (**Mpre**), and reduced it to produce reactive monomers immediately before observation. Therefore, the structure of the precursor and the presence of the reductants (DTT) are due to technical requirements.

29. Page 5, line 10. The prebiotic relevance of BnSH has to be discussed. And if it is being used as a modern model analog system, why can it be used and the results extended to true prebiotic

chemistry? What happens to the system in the absence of BnSH? Please also include a reference regarding how BnSH can promote self assembly.

Answer

Our BnSH is one model of the leaving groups because it is hydrophobic and is easier to form droplets. Using a peptide containing a cysteinyl prolyl ester (CPE) moiety at the C-terminus is alternative strategy, because CPE-peptide is a simple structure.

30. Fig. S1. Please define all of the abbreviations used in the synthesis scheme. This includes the difference between SBn and BnSH, as well as DCC, Boc, and HOBt. These are all non-standard chemicals that must be defined, especially as they are used in the subsequent figures. Additionally, this synthesis scheme (especially the BnSH addition step) should be added to the methods section. Finally, Mpre should be represented identically in both the main text and in the supplement. Right now, the two chemical structures, although technically identical in structure, are not depicted identically (the SI uses Rn^- notation, while the main text does not), which may lead to major confusion amongst readers.

Answer

In order to clarify the relationship between the spectra and the synthesis method of (Boc-Cys-SBn)₂, the synthetic method is referred to in Fig.S2 and the ¹H-NMR spectra was referred to in the Method section in the article manuscript. Similarly, we referred (Cys-SBn)₂•2HCl (Mpre) to Fig.S3 and Method section.

31. Figs. S2-S3. What is the significance of these NMR spectra? This can be as simple as stating something like “...indicating the successful synthesis of Mpre.” Additionally, if the journal requires it, please submit an NMR peak list.

Answer

In order to clarify the relationship between the spectra and the synthesis method of (Boc-Cys-SBn)₂, the synthetic method is referred to in Fig.S2 and the ¹H-NMR spectra was referred to in the Method section in the article manuscript. Similarly, we referred (Cys-SBn)₂•2HCl (Mpre) to Fig.S3 and Method section.

32. Page 5, line 12. What is the significance of DTT? Is it prebiotically relevant? Just as a modern model reducing agent? A discussion of the relevance of a reducing atmospheric and ocean conditions on early Earth should be discussed here. Such a discussion would increase the significance and relevance of the paper.

Answer

DTT is a model molecule used to create an artificial reductive atmosphere. Of course, the system can

be realized without it, but it is used to accelerate the reduction.

33. Page 5, line 21. How can you be sure nanometer-sized aggregates were formed? Did you validate this through experiments?

Answer

As the reviewer pointed out, the nanometre-sized molecular aggregates could not be observed under an optical microscope in principle. We discovered this fact by the particle size distribution measurements. Therefore, we moved the following statement "*These results indicate nanometre-sized molecular aggregates were formed until 1 hour after mixing the Mpre and DTT, and then they grew or fused to large enough to be observed with a microscope*" to the proper location of the size measurement results (P.12, L2).

34. Page 6, line 3: how long did it take before the droplets completely coalesced into a single structure? Or was this not observed?

Answer

We rewrote the sentence to "*The assemblies fused and maintained round, suggesting that the formed aggregates were droplets (Supplementary Movie 1).*"(P.6, L6), because it was misleading. Incidentally, the droplets were stable even 3 months after it stored and did not spontaneously separate.

35. Page 6, lines 11-13: I don't think that Fig. 2c has open squares or crosses. The lines are instead indicated by different colors. This should be changed either in the text, or the figure.

Answer

As the reviewer pointed out, the figure has been revised. In the manuscript, we had a mistake in the comments about the Fig. 2c. There were no white squares and crosses in the Fig2c. We revised the full circles, open squares, and crosses in the manuscript to the red line (P.5, L16), green line (P.6, L11), and blue line (P.6, L13) respectively. In addition, we deleted the sentence in the previous manuscript (P6, L11) "which is about half of the mixture solution of **Mpre** and DTT".

36. Fig. S5. The synthetic method of this molecule should be listed in the methods.

Answer

We described in the Methods section of the text. However, to make it easier for readers to understand, we inserted a reference to the Methods section in the text, and also added a reference to S5 in the Methods section.

37. Figure 2c: the black line is not easily visible.

Answer

Because the black line was indistinct, We revised the figure as follows. The blue line (Mpre) in the previous manuscript to green in the revised manuscript, and black line (DTT) in the previous manuscript to blue line.

38. Fig. S8b. While the table is presented, please directly show the full spectrum. This is a better way to visualize the data.

Answer

We added the entire mass spectra and the respective peaks shown in Table of Fig.S7 as magnified views to Fig.S8c.

39. Page 9, line 3. what does “disperse” mean? Did you mean turn turbid? Additionally, these micrographs should be included as well.

Answer

We revised the sentence: “*The solutions that contained no oligopeptides or BnSH did not turn turbid,*” (P.9, L.3)

40. Page 8, line 8. What evidence do you have of associative LLPS? Why could it not have been segregative LLPS?

Answer

The evidence that our droplet is an associative LLPS droplet had been provided by Dr. Nicolas Martin's recent review [ref.34].

“The term coacervation was originally introduced by Bungenberg de Jong in the 1920s specifically to describe the associative phase separation between two oppositely charged natural polyelectrolytes.¹⁷ Since then, two limiting cases of associative phase separation, dominated either by desolvation or by electrostatic interactions, have been identified, although associative LLPS can also be driven by other attractive interactions (as discussed below for biomolecular condensates). Whereas simple coacervation refers to systems containing only one colloidal component undergoing phase separation due to decreased solvation (e.g., through addition of a dehydrating chemical agent or changes in temperature or pH), complex coacervation describes a spontaneous phase separation resulting from attractive electrostatic interactions of two oppositely charged molecules or colloids in aqueous solution. Complex coacervates have been produced from a number of colloidal objects, including hundreds of different synthetic or natural polyelectrolytes, proteins,¹⁸ shorter polyions, such as oligopeptides¹⁹ and oligonucleotides,²⁰ and small molecules, such as mononucleotides²¹ and surfactants.²²

It has been proposed that the forces driving the phase separation of two oppositely charged long polyelectrolytes originate from entropy increases associated with the release of counter - ions and rearrangements of water molecules during the formation of macro - ion pairs.²³ These pairs further assemble to produce a liquid - like dense microphase. In some cases, the polyelectrolyte pair eventually precipitates to form a solid - like complex, but the parameters influencing the nature of the complex formed (liquid or solid) for a given polyelectrolyte pair remain unclear.²⁴ Other open questions include the detailed roles of charge distribution and density, long - range electrostatics, hydration effects and contributions of other interactions (hydrogen bonds, cation - π , ...). Recent studies in this area have sought to understand these factors in greater detail by using computational modelling combined with better - defined polyelectrolyte structures,²⁵ in terms of, for example, polymer length dispersity,²⁶ charge distribution,²⁷ chirality²⁸ or base pairing.^{20, 29, 30} These factors, together with the chemical natures of the species involved (long polyelectrolytes or small molecules), influence the material properties of the coacervate phase produced, including the rheological and interfacial properties.³¹ The ease of coacervate formation and the diversity of assembly blocks have provided applications in areas as diverse as food thickening,³² cosmetic³³ or pharmaceutical^{25b} formulations, protein purification,³⁴ wastewater treatment³⁵ or underwater adhesion.³⁶

In the mid - 1920s, Oparin became the first to suggest that liquid - like microdroplets formed by complex coacervation in water could have played a key role in the emergence of the first metabolic cells.³⁷ These coacervate droplets were shown to undergo fission, to favour chemical enrichment through spontaneous solute sequestration and to support enzyme - mediated reactions.³⁸ This idea has re - emerged in recent years with pioneering studies by Mann and co - workers, who demonstrated for the first time the formation of complex coacervate droplets from low - molecular - weight mononucleotides and their use as viable protocell models.²¹ Since then, bio - inspired functions have been demonstrated in a variety of other membrane - free droplets based on complex coacervates, including surfactant - based protocells.^{39, 40}

41. Fig. 4: The figure has very low image quality and is very blurry. It should be replaced with a much higher quality figure.

Answer

Thank you for pointing out. We improved the resolution of Figure 4.

42. Page 14, line 16. I think this statement should be introduced in the introduction, in order to immediately highlight the significance of this study.

Answer

We noted it in the introduction section (P2, L.14)

43. Page 15. What is the mechanism by which some fluorescent molecules concentrated on the edges, while others concentrated in the center? For example, it was suggested that the edges are more hydrophilic, while the interior is more hydrophobic. Is there any additional evidence for this?

Answer

The heterogeneity in the droplet was directly visualized by Raman microscopy, whereas it was difficult to measure the droplet by LC mass spectrometry due to the presence of disulfide. The results of Raman microscopy have been added to Fig.S14 in Supplemental Information. We showed in a revised Fig.S9 the result that the droplets were reconstructed by mixing a known concentration of BnSH with oligo cysteine.

44. Fig. S12. What was the reason that nucleic acid solutions were observed 30 minutes after addition, while lipid solutions were only observed 16 hours after addition?

Answer

Since the rate of lipids uptake into a droplet was slower than that of nucleic acids, it took longer time to observe.

45. Page 15, line 22. About the size change when both lipid and nucleic acid were added together, this data should be introduced here and also a similar FACS analysis should be shown.

Answer

The FACS equipment and analysis software are in a remote location. Unfortunately, due to the COVID-19, we are forbidden to move freely. Therefore we do not have the FACS results. However, we believe that the size of the droplets will remain the same.

46. Fig. 5. The letter labels for Fig. 5 are out of order. They should be listed in increasing order from left to right, and then to the next row. As it stands, it is very difficult to read this figure properly (for example, b should actually be f).

Answer

In fact, before submitting this article, we also tried the same arrangement of figures as suggested by the reviewer. However, we found several articles that changed the figure order following Nature Communication's policy of emphasizing the readability. After some trial and error, we changed the figure order like the precedent.

47. Fig. 5. You cannot call a part of a figure before that part of the figure was called. For example,

in the figure caption, 5g was called in 5d. However, you must call 5g first, and then it can be referenced in another part of the figure. Thus, Fig. 5 should be completely rearranged so that this is the case.

Answer

The same answer as in 46.

48. Page 16, line 16. Why is this the assumption exactly? If there is experimental or theoretical evidence, then this should be presented. Some type of IR or Raman microscopy, DESI-MS (or another type of spatial MS such as SIMS), or even a peptide derivatizing fluorescence dye which can allow spatial fluorescence can show direct evidence of the spatial chemical structure within the droplets.

Answer

We revised the sentence as follows: *From the Raman spectrometry, it was found that the droplets were composed of the more hydrophobic peripheral part with a larger Bn/water value and of the more hydrophilic central part with a smaller Bn/water value (Fig.S14).*(P.19, L.21)

49. Page 19, line 5. Other theories of origins of life could also be relevant, such as the polyester-based origins of life model (Tony Z. Jia, Kuhan Chandru, Yayoi Hongo, Rehana Afrin, Tomohiro Usui, Kunihiro Myojo, and H. James Cleaves II. Membraneless polyester microdroplets as primordial compartments at the origins of life, PNAS 2019, 116, 15830), the Shadow Biosphere (Be Davies PC, Benner SA, Cleland CE, Lineweaver CH, McKay CP, Wolfe-Simon F. Signatures of a shadow biosphere, Astrobiology 2009, 9, 241), or composome model (Segré, D.; Ben-Eli, D.; Lancet, D. Compositional genomes: Prebiotic information transfer in mutually catalytic noncovalent assemblies. PNAS 2010, 97(8), 4112). How these theories could be incorporated by the system should also be discussed briefly.

Answer

Thank you for your valuable input. We certainly think that the above researches are important for us as well. We will make use of the reviewer's comment in our future study.

50. Page 19, line 6. This sentence should be rewritten. As it stands, it does not convey the meaning that the authors likely want to convey.

Answer

We revised the sentence as follows: *The droplets not only incorporated nucleic acids and lipids, but also acquired the ability to survive by accelerating interactions among these materials.*(P.21, L.4)

51. Page 19, line 9. The thioester world scenario was not introduced in adequate detail. This

should be done in the introduction.

Answer

The relationship between the thioester world and this study was described in the introduction section. The amount of text has been kept down to balance the other fields. In the next work, we are considering a study based on the thioester world hypothesis.

52. Page 20, line 22. It should be noted that because RNA is so labile, that some of these potential substrates could quickly hydrolyze the RNA. This possibility should be discussed, and also underscores why it is important to continue discussing the conditions used throughout the manuscript, and how these conditions are conducive to biomolecule function and persistence, but also to true geological conditions on primitive Earth. It is also necessary to mention that many RNA reactions can only be performed

Answer

Although RNA is generally fragile in an aqueous solution condition, its lifetime is known to be extended in the hydrophobic field. In the current system, it is predicted that RNA is incorporated into the hydrophobic field in the same way as DNA, which extends its lifetime. Thus, this could have a function of providing durability to the droplets. This is expected to contribute to the expression of enzymatic activity when RNA is accompanied by metal ions. Further research is expected in the future. Thanks for the interesting suggestions.

53. What happens to the system in the presence of Magnesium (or another divalent cation)? Most RNA-based reactions such as ribozyme catalysis or non-enzymatic replication require such a cation, and the incorporation of such cations into the system should be discussed.

Answer

The same answer as in Answer 46.

54. Page 21, line 16. To claim “droplet world” also requires the authors to properly cite those who have proposed and researched it in the past. Please include these citations and give proper credit where credit is due.

Answer

People google both "Origin of Life" and "Droplet World", and our works and Hanczyc's paper [ref46 in the revised manuscript] come up (as of the end June 2020). However, his proposed the Droplet world, while pioneering in terms of primitive droplets moving and dividing, lacked the concept of self-production processes and intake of prebiotic molecules. We solved this problem experimentally by realizing a liquid-liquid phase separated droplet emergend from simple molecules. Since we were able to show the effectiveness of our droplet as an assembly filed for prebiotic molecules in the origin of

life, we believe that we have succeeded in elevating the Droplet World to the level of a hypothesis. Therefore, we did not cite Martin's paper in our claim.

55. It is understandable that as the authors are not native English speakers, that a large number of minor spelling and grammatical errors exist (which don't detract from the science presented, just the readability). To improve the readability of the manuscript, I would recommend that the authors ask one of their colleagues or collaborators, who is a native English speaker, to carefully proofread their manuscript. If this is not possible, there are a number of paid services available. I think this step is imperative to achieve the most easily and widely understood manuscript as possible. Here, I have listed a few examples of things that should be changed from the first few pages, although this is not an exhaustive list. I hope that this gives the authors a sample of the number of errors in the entire manuscript (and numerous errors in the supplementary information) that must be fixed.

Answer

Any grammatical errors were checked and revised by a native speaker. We will use English editing services as required.

Reviewers' Comments:

Reviewer #1:

Remarks to the Author:

This paper titled "Self-proliferating Liquid-liquid Phase separated Droplet; the missing link between Chemistry and Biology on the Origin of Life" by Matsuo and Kurihara describes coacervate droplet formation by in-situ synthesized peptides that can accumulate and distribute RNA and lipid heterogeneously within the droplet combining RNA, lipid and peptide in one droplet. As described previously there are novelties in this work which make it suitable for nature Communications. However, even after the revision, this manuscript requires revisions before it is suitable for publication. In general, the language of the text is still unclear, the figures and figure legends require some refinement. A whole page figure and figure legend is difficult to be transferred to any journal format. The distribution of data across figures to argue a point makes the manuscript difficult to follow. In addition, please see responses to the author's rebuttal following the same number scheme as the first revision to help to navigate the comments:

1. Even after the revisions, the definitions as well as the experimental data to support the claims that the system, proliferates and self-reproduces are still not in line with general descriptions of these processes. For example, for a proliferating system, one would expect to observe an increase in number of droplets. The point that the droplets reach a steady state in size, alone, as has been described describe proliferation. For a self-reproducing system one would expect that new droplets are made from existing droplets. The data presented to support this argument could be attributed to continual nucleation and formation of new droplets. The additional experiment Figure A1a and A1b, S16 does not support their arguments as there do not appear to be any droplets in the solution. For the self-maintenance argument the FACS data shows a change in distribution with the addition of both RNA and HPC but does this really indicate self maintenance.

2-3. The new Raman microscopy data presented is very interesting.

2. Overlaying the spectra from the two regions would be nice- this will make it easier for reader to see the differences between the spectra in different regions of the droplet.

3. Care should be taken on the choice of region. For example, for Figure S21, the choice of the outer region is different in comparison to d and g. This can significantly bias the data. An average of multiple experiments would be important to verify the findings. Inclusion of a description in the figure legend with regard to the wavelengths chosen for analysis would also help to explain the system.

4. The figures should be further improved. A full page figure will not be so easily reproduced into any journal

5. Thank you for the expanded data for Mass spec. Please also include a better explanation for the labels (1A, 2A etc....)

6. With regard to this point and point 2 and 3. It should be possible to undertake these experiments with RNA as there should not be any issues with RNA stability for these experiments.

7. For figure 4c, was extrusion undertaken on the droplets? What is the error in the number density? Is the difference in the number density significant? With regard to this point..."To validate the reviewer's proposed condition of whether fusion affects droplet growth, we added deionized water without Mpre to the extruded droplet dispersion. No significant droplet formation was observed (Fig. A1).Therefore, the increase in particle size after extrusion is considered to be induced by self-production, not by fusion." Figure A1, shows no droplets at all. This experiment should be undertaken with a dispersion of droplets.

8. Figure A6 / S15 is hard to understand with the 3d plotting

10. For this amendment, do the droplets grow by autocatalytic reaction, or proliferation or reproduction? Here it is described as growing by autocatalysis whilst in other parts of the text, it is described as self-reproduction.

15. No explanation of the observations are given.

18. The paper they have cited was not the correct citation.

23. I disagree, the droplets in the microscopy images are less than 5um and the scale bars are 20, 100 um, making this not appropriate.

Reviewer #2:

Remarks to the Author:

The authors have made substantial changes to their manuscript and improved the overall readability significantly. They have commented on all points raised by the reviewers, but in my opinion, they have not resolved some of the main issues raised. As it stands, the paper still lacks some crucial characterization of the droplets, controls for the monitoring of self-reproduction by DLS, and a realistic discussion of the relevance of these droplets as a missing link between chemistry and biology. For those reasons, I do not think it is acceptable for Nature Communications yet in the current form. However, I do believe the system the authors describe is truly novel and displays very interesting behaviour, and I would be glad to recommend its publication if the authors can provide the necessary characterization and controls.

Composition:

The authors describe their droplets as coacervates, formed by associative phase separation, they write in several places that their droplets are peptide-based, composed of oligopeptides, and that oligopeptides formed droplets, but a characterization of the composition of the droplets is not provided. The Raman microspectroscopy does not show the presence of the oligopeptides and only indirectly shows BnSH. I agree with reviewer 1 that a more detailed characterization of the composition of the droplets is essential to conclude that they are coacervates composed of peptides, formed by associative phase separation. A compositional analysis will also enable others to build on this work. The authors seem to work with significant amounts of droplets; it should be possible to separate them from the solution by centrifugation and analyse them by NMR, UV and/or HPLC with adequate controls and calibrations to identify the constituents.

Self-reproduction:

The self-reproduction monitored by DLS and shown in figure 4 is an important element of the authors' work. As discussed after the first submission, several factors can influence a measured size (distribution) from DLS, and control experiments are needed to be able to draw conclusions about droplet growth, self-reproduction and the location at which the reactions take place from shifting size distributions and traces of the average particle size. It is risky to base a conclusion on the absence of a peak. How can the authors be sure that they will not miss a distinct peak of a newly-born population in DLS in the presence of an older population of much larger droplets? A control in which they detect such a second peak in a mixed sample would solve this question. In addition, I think it is difficult for readers to gauge what the effects of nutrient addition, shearing and passive coarsening on the average DLS size are without a control in which the average size measured by DLS is plotted over several periods for samples without nutrient addition, and samples without applied shear.

Prebiotic relevance and missing link:

After re-reading the manuscript I still think that some terminology and phrases are used without sufficient justification. The title still mentions a missing link between chemistry and biology, but there is no further mention or discussion of how this system exactly provides the missing link. Many scientists would not consider the autocatalytic oligomerization presented here as having a link to biology. The proliferation displayed by living organisms through growth and division is a biological process but very distinct from the phenomena described here.

The authors also conclude that their system is, or contributes in some way to a universal common ancestor (p.25), but that is not shown in the current work, as it would involve demonstrating that living cells could evolve from this droplet. The same for the statement on p.4 that "This study may serve to explain the emergence of the first living organisms on primordial Earth."

Finally, the authors have added a helpful discussion about the prebiotic relevance of their components. I am not an expert in this area, but in light of the discussion on prebiotic relevance, I wonder if the current system should really be called a prebiotic (model) system (as is done in the abstract and introduction). BnSH is not a prebiotic species, but used here as a model thiol (p.23). The authors suggest alkylthiols as good prebiotic alternatives to BnSH, but if the authors are right about the cation-pi interactions being the driving force for LLPS, replacing the BnSH by an alkylthiol would not yield coacervate droplets, self-reproduction or self-maintenance. This should be discussed.

Minor points:

- Figure S17 still has an incorrect y-axis
- p.11, line 5-8 ("However...residual proportion.") The meaning of this sentence is not clear.
- p.14 line 7, remove "the" before fusion
- Figure 4c: why is the x-axis scale different from 4b? If there are small droplets, they would be expected at small particle size.

Reviewer #3:

Remarks to the Author:

I would like to thank the authors for preparing a very thorough and detailed response, and I think that the manuscript has improved greatly. I do have a few remaining minor points that I would suggest the authors consider; after these points are addressed, I believe that the manuscript would be ready for publication.

Minor points:

1. In the original point 7, I had suggested to discuss the prebiotic relevance of the concentration of the chemicals used in the study. However, the authors misunderstood my point and have in fact pointed out the opposite: that the concentrations used are too high to have been prebiotically plausible. An additional statement or short discussion about why it is reasonable to use mM concentrations of chemicals in this study when it is in fact difficult (or almost impossible) to synthesize pure molecules to the mM order in an actual prebiotic environment.

2. In point 7, the prebiotic explanation of the plausible synthesis of the components is very reasonable and comprehensive, and while this level of detail should not be included in the main text, I wonder if it would make sense to include this very nice and detailed discussion somewhere in the SI.

3. All mentions of "origin of life" should ideally be "origins of life", because it is unknown whether there was one or many origins that led to our current life; it is also unknown whether we are the only life.

4. While I believe that the English of the manuscript has improved significantly, there are still a number of minor grammatical errors throughout the document. I would suggest the authors review the document carefully one last time. Here are some examples from only the introduction (not exhaustive):

-Page 2 Line 30, "an" should be "the"

-Page 2 Line 34, both mentions of "are" should be "is"

-Page 2 Line 42, "on" should be "regarding" or similar

-Page 3 Line 66, "property" should be "properties"

-Page 4 Line 73, an Oxford comma should be added before "solid-phase"; please be consistent with using the Oxford comma throughout the document, as this was the convention chosen at the first use.

-Page 4 Line 75, "emerge" is an intransitive verb, but it is used as a transitive verb in this sentence, which is grammatically incorrect.

-Page 4 Line 75, "with" should be "through", as "with" is ambiguous and can also mean "containing", which is not necessarily what the authors mean in this sentence (I believe).

-Page 4 Line 85, "in" should be "at". This is because "in" may imply that the NAs were physically inside or within the inner boundary, whereas what I believe that the manuscript argues that they are, in fact, "at", "on", or "localized" to the boundary.

-Page 5 Line 94, "thioester" should be "thioesters"; more than one thioester molecule was added to the system.

Reviewer #1 (Remarks to the Author):

This paper titled “Self-proliferating Liquid-liquid Phase separated Droplet; the missing link between

Chemistry and Biology on the Origin of Life” by Matsuo and Kurihara describes coacervate droplet formation by in-situ synthesized peptides that can accumulate and distribute RNA and lipid heterogeneously within the droplet combining RNA, lipid and peptide in one droplet. As described previously there are novelties in this work which make it suitable for nature Communications. However, even after the revision, this manuscript requires revisions before it is suitable for publication. In general, the language of the text is still unclear, the figures and figure legends require some refinement. A whole page figure and figure legend is difficult to be transferred to any journal format. The distribution of data across figures to argue a point makes the manuscript difficult to follow. In addition, please see responses to the author’s rebuttal following the same number scheme as the first revision to help to navigate the comments:

1. Even after the revisions, the definitions as well as the experimental data to support the claims that the system, proliferates and self-reproduces are still not in line with general descriptions of these processes. For example, for a proliferating system, one would expect to observe an increase in number of droplets. The point that the droplets reach a steady state in size, alone, as has been described describe proliferation. For a self -reproducing system one would expect that new droplets are made from existing droplets. The data presented to support this argument could be attributed to continual nucleation and formation of new droplets. The additional experiment Figure A1a and A1b, S16 does not support their arguments as there do not appear to be any droplets in the solution. For the self-maintenance argument the FACS data shows a change in distribution with the addition of both RNA and HPC but does this really indicate self-maintenance.

Answer

The various experimental data about self-reproduction in the main text and the Supplementary Information clearly have showed that droplets grew and divided by self-reproduction. Proliferation of droplets in our study, as clearly defined in the main text (P.3, L.60), means growth and division by self-reproduction. In other words, all of the data (Figs.2-4, Figs.S17-20) proving growth and division of the droplets by self-reproduction are equivalent to proving proliferation of them. Therefore, referee 1's opinion, "*The point that the droplets reach a steady state in size, alone, as has been described describe proliferation. For a self-reproducing system one would expect that new droplets are made from existing droplets*", is not valid. The increase in numbers of droplets is strongly suggested by the consistency of particle size in each generation even with addition of precursor M_{pre} . However, we added the following experiment (Fig. A1) to enlighten readers. In the following additional experiment, after the addition of monomer precursor, the dispersion was diluted by the incremental amount after the increase in droplets. It shows that a constant number density was achieved. In the current Fig.A1 below, the observed number of droplets was estimated with the actual volume of the dispersion, thus indicating that the droplets proliferated. We added this result as the Fig.

supplementary
cited this in the

S20 in the
information and
text.

Figure A1 | Relative number of droplets for each generation. Each dispersion obtained due to the same protocol as in Figure 4e were observed under the phase-contrast microscopy microscope. Droplets with the size of over 1 μm in the microscope images ($290 \mu\text{m} \times 217 \mu\text{m}$, $n=5$) were counted. The horizontal axis is the order of the generations, and the vertical axis is the relative number of droplets. Here, the number of droplets each generation is relative to that of the first generation as 1. Error bars are standard deviations.

Additionally, the reviewer claims “The additional experiment Figure A1a and A1b, S16 does not support their arguments as there do not appear to be any droplets in the solution”, which is not correct. Additional experiments (original S16 in the previous SI, new S19 in the revised SI) show that droplet does not grow via droplet fusion. Since growth of droplet size is derived from either self-production of droplet or fusion among droplets, the growth in the current study can be nothing but derived from self-production.

The reason that the droplets are invisible in the former Fig. S16 (Fig.A1 in the former answer sheet) is that the droplets are reduced to a size below that which can be observed with an optical microscope by extruding them through a 100 nm filter to match the size of the particle size distribution measurements. This experiment means that the number of droplets does not increase unless the monomer precursor is added again. In the observation experiments of droplet growth, we observed that the number of droplets increases with the addition of food in a wide field of view (Figure 2d in the article), but when the number of droplets increases to a certain extent, fusion becomes dominant due to the close distance between the droplets. Therefore, in the following additional experiment, after the addition of monomer precursor, the dispersion was diluted by the incremental amount after the increase in droplets. It shows that a constant number density was achieved.

As reviewer mentioned, in the new Figure 6, when lipids were added to the droplet dispersion, the droplet size decreased, indicating dissolution. On the other hand, when both nucleic acids and lipids are added to the dispersion, the droplets show resistance to

dissolution, indicating self-maintenance.

2-3. The new Raman microscopy data presented is very interesting.

2. Overlaying the spectra from the two regions would be nice- this will make it easier for reader to see the differences between the spectra in different regions of the droplet.

Answer

As the reviewer pointed out, we redrew the Figures S22 and S24. Raman spectra of the central part (red line) and the peripheral part (blue line) in both Figures S22 and S24 were overlaid.

3. Care should be taken on the choice of region. For example, for Figure S21, the choice of the outer region is different in comparison to d and g. This can significantly bias the data. An average of multiple experiments would be important to verify the findings. Inclusion of a description in the figure legend with regard to the wavelengths chosen for analysis would also help to explain the system.

Answer

In Figure S22, we added all the regions measured by Raman spectroscopy; four periphery and four central regions per droplet. In the same way, we drew the measurement areas in Figure S24. We also included the wavelengths irradiated in the Raman measurement in the Figure legend to help the readers understand the results.

4. The figures should be further improved. A full page figure will not be so easily reproduced into any journal

Answer

We divided Figure 5 into two parts; the new Figure 5 (microscopy images of

incorporation of both RNA and lipid) and the new Figure 6 (microscopy images, line profiles and flow cytometry experiments of incorporation of RNA or lipid). We also revised the index and sentences in the manuscript accordingly. As the reviewer pointed out, we believe it is easier for the reader to read.

5. Thank you for the expanded data for Mass spec. Please also include a better explanation for the labels (1A, 2A etc....)

Answer

We labeled the character (1A, 2A, ...) beside the peaks showing each peptide in Figure S10c.

6. With regard to this point and point 2 and 3. It should be possible to undertake these experiments with RNA as there should not be any issues with RNA stability for these experiments.

Answer

We performed Raman measurements of mixtures of RNA and droplet dispersions, and also of mixtures of RNA, lipids and droplet dispersions. We show the analysis results for RNA in the Supporting information Figure S24 and Table S1, and combined the results of Raman measurements for DNA. The experimental results for RNA in droplets showed a similar tendency to those for DNA.

7. For figure 4c, was extrusion undertaken on the droplets? What is the error in the number density? Is the difference in the number density significant? With regard to this point..."To validate the reviewer's proposed condition of whether fusion affects droplet growth, we added deionized water without Mpre to the extruded droplet dispersion. No significant droplet formation was observed (Fig. A1).Therefore, the increase in particle size after extrusion is considered to be

induced by self-production, not by fusion.” Figure A1, shows no droplets at all. This experiment should be undertaken with a dispersion of droplets.

Answer

In the experiment shown in Figure 4c, particle size measurements were performed on dispersions extruded through a membrane filter with a 100 nm diameter pore. As we mentioned in Question 1, there are no droplets in the Figure S16 (Figure A1 in the last answer sheet). However, the droplets are below the size that can be observed under the microscope because the size of the droplets were filtered to match the particle size distribution measurement conditions. The S16 experiment shows that the droplets do not grow to a microscopy-observable size spontaneously when they are extruded and fused. This means that the size of the droplets does not change as shown in Figure 4c unless they are fed monomer precursors. As for the number density experiment, the growth of droplets was shown in Figure S20 (additional experiment Figure A1). The number density is significantly different between generations. Although the referee claims that "*Figure A1, shows no droplets at all*", the sentence clearly proved that the dispersed droplets are divided down to a nanometer size, which cannot be observed by the microscope and also can only be detected by DLS size measurement. Thus, the previous Fig.A1 and Fig.4c in the article text satisfied the referee's requirement of "*This experiment should be undertaken with a dispersion of droplets*" in the first place.

8. Figure A6 / S15 is hard to understand with the 3d plotting

Answer

We changed Figure from 3D plots to 2D plots as follows (Figure A3). We sure it's easier to understand.

Figure A3 | Time-course changes in droplet size and reaction rate. Change in particle size and reaction rate over time. The cube root of the reaction rate was calculated from the droplet formation rate (Fig 3b). The reaction rate reached flat at about 3 h, whereas the increase in droplet size continued thereafter, although the growth rate of particle size was significantly lower.

9. For this amendment, do the droplets grow by autocatalytic reaction, or proliferation or reproduction? Here it is described as growing by autocatalysis whilst in other parts of the text, it is described as self-reproduction.

Answer

This system consists of a flow of autocatalytic reactions, self-reproduction and proliferation. The droplets produce peptides from monomers through autocatalysis, and then the droplets grow. The grown droplets take in more precursor molecules and grow larger, i.e., self-reproduction. And then the droplets divide into the smaller ones. As this self-reproduction takes place all over the system, the number of droplets increased, i.e. proliferation.

10. (#15) No explanation of the observations are given.

Answer

We added the following explanation of the observations to the main text (P.23, L.397-400); *“Nucleic acids are known to be incorporated into more hydrophobic regions by forming complexes with lipids due to electrostatic interactions^{49,50}. In this study, complex formation due to intermolecular interactions between RNA and lipids / peptides would have promoted their incorporation into the BnSH phase.”*

11. (#18) The paper they have cited was not the correct citation.

Answer

We recited Mann group's research properly.

12. (#23) I disagree, the droplets in the microscopy images are less than 5um and the scale bars are 20, 100 um, making this not appropriate.

We believed and still believe that the scale bars were appropriate as shown in a following Fig. A4. We have not previously stated in the paper that the droplets shown in Figure 5b are less than 5 μm in diameter. In this experiment, we deliberately selected the droplets with about 20 μm in diameter to observe the incorporation of the biopolymer easier. The scale bars in Figure 5b are 20 μm and 100 μm . However, the scale bars in Figure 5b are still hard to understand. Therefore we modified the figures slightly. We insert a space between the overall figure and the magnified figures.

Figure A4 | Scale bars.

Reviewer #2 (Remarks to the Author):

The authors have made substantial changes to their manuscript and improved the overall readability significantly. They have commented on all points raised by the reviewers, but in my opinion, they have not resolved some of the main issues raised. As it stands, the paper still lacks some crucial characterization of the droplets, controls for the monitoring of self-reproduction by DLS, and a realistic discussion of the relevance of these droplets as a missing link between chemistry and biology. For those reasons, I do not think it is acceptable for Nature Communications yet in the current form. However, I do believe the system the authors describe is truly novel and displays very interesting behaviour, and I would be glad to recommend its publication if the authors can provide the necessary characterization and controls.

1. Composition:

The authors describe their droplets as coacervates, formed by associative phase separation, they write in several places that their droplets are peptide-based, composed of oligopeptides, and that oligopeptides formed droplets, but a characterization of the composition of the droplets is not provided. The Raman microspectroscopy does not show the presence of the oligopeptides and only indirectly shows BnSH. I agree with reviewer 1 that a more detailed characterization of the composition of the droplets is essential to conclude that they are coacervates composed of peptides, formed by associative phase separation. A compositional analysis will also enable others to build on this work. The authors seem to work with significant amounts of droplets; it should be possible to separate them from the solution by centrifugation and analyse them by NMR, UV and/or HPLC with adequate controls and calibrations to identify the constituents.

Answer

We were unable to separate the droplets from the peptides by centrifugation, although

we attempted. Therefore, we conducted Raman measurement, as the reviewer's previous suggestions of NMR, UV and HPLC were not possible. However, we did not satisfy the reviewer's expectations as the fingerprint region of the peptide was buried in other constructs.

Indeed, we agree to determine the composition inside the droplet. Therefore, we prepared several composition ratios of benzyl mercaptan and peptides to determine the correlation between the formation of droplets and their composition ratios. This figure was added in Supplementary Information (Figure S12).

Figure A1 | Composition ratio of peptides and benzyl mercaptan on droplet formation. **a**, Photographs of the droplet formation dependent on the concentration ratio between benzyl mercaptan (BnSH) and peptides. The concentrations of the oligopeptides are expressed as those of amino acids. Each specimen was prepared by adding BnSH and deionized water to the peptides obtained by the same procedure shown as Figure 3a. **b**, Heat map of the average number of the counted droplet Only droplets with a diameter larger than 1 μm were counted in the microscopy images (290 μm \times 217 μm , $n=3$) of each dispersion. The average number of counted droplets was heat-mapped after Gaussian fitting. The average number of droplets is normalized with [Amino acids]=10mM and [BnSH]=25mM as 100.

2. Self-reproduction:

The self-reproduction monitored by DLS and shown in figure 4 is an important element of the authors' work. As discussed after the first submission, several factors can influence a measured size (distribution) from DLS, and control experiments are needed to be able to draw conclusions about droplet growth, self-reproduction and the location at which the reactions take place from shifting size distributions and traces of the average particle size. It is risky to base a conclusion on the absence of a peak. How can the authors be sure that they will not miss a distinct peak of a newly-born population in DLS in the presence of an older population of much larger droplets? A control in which they detect such a second peak in a mixed sample would solve this question. In addition, I think it is difficult for readers to gauge what the effects of nutrient addition, shearing and passive coarsening on the average DLS size are without a control in which the average size measured by DLS is plotted over several periods for samples without nutrient addition, and samples without applied shear.

Answer

To judge growth, self-reproduction, and reaction location of droplets, based on shifts in size distribution and tracking of the average particle size, we performed other control experiments to measure the dynamic light scattering intensity of the solutions before / after extrusion and their mixed solutions. It is shown that the droplet in the dispersion is gently distributed in size and that the mixed dispersion of small-size droplets and large-size droplets is dominated by the size of the high number of droplets. The results of this experiment show that the measurement accuracy of the DLS measurement can detect small and large droplets simultaneously in mixed samples since the size distribution is different from that before mixing. The scattering intensity distribution of the mixed solution shows an intermediate distribution compared with that of the before mixed solution, and at least both distributions was detected. On the other hand, most of the droplets in the particle size distribution were around 100 nm. This means that the number of droplets in the extruded dispersion is extremely higher than that of unextruded solutions with the same amino acid concentration. In other words, these

results strongly suggest that if a monomer precursor of the same amino acid concentration is added to a droplet dispersion and then the chemical reaction proceeds outside the droplet to generate new smaller droplets, a new distribution is produced and the original distribution disappears. The experimental results in Fig. 4b show that neither of these events occurred, which conclude that the chemical reaction proceeded almost inside the droplets. These figures were added in Supplementary Information (Figure S15).

Figure A2 | a, Scattering intensity measurement of the mixture before/after extruded droplet dispersion. **b,** Size measurements of the mixture before/after extruded droplet dispersion. Mixed droplet dispersions (before filtration : after filtration = 1:1 or 10:1 mol%) before and after extrusion through a 100 nm mesh polycarbonate filter were measured by dynamic light scattering particle size distribution analyser.

3. Prebiotic relevance and missing link:

After re-reading the manuscript I still think that some terminology and phrases are used without sufficient justification. The title still mentions a missing link between chemistry and biology, but there is no further mention or discussion of

how this system exactly provides the missing link. Many scientists would not consider the autocatalytic oligomerization presented here as having a link to biology. The proliferation displayed by living organisms through growth and division is a biological process but very distinct from the phenomena described here.

The authors also conclude that their system is, or contributes in some way to a universal common ancestor (p.25), but that is not shown in the current work, as it would involve demonstrating that living cells could evolve from this droplet. The same for the statement on p.4 that "This study may serve to explain the emergence of the first living organisms on primordial Earth."

Finally, the authors have added a helpful discussion about the prebiotic relevance of their components. I am not an expert in this area, but in light of the discussion on prebiotic relevance, I wonder if the current system should really be called a prebiotic (model) system (as is done in the abstract and introduction). BnSH is not a prebiotic species, but used here as a model thiol (p.23). The authors suggest alkylthiols as good prebiotic alternatives to BnSH, but if the authors are right about the cation- π interactions being the driving force for LLPS, replacing the BnSH by an alkylthiol would not yield coacervate droplets, self-reproduction or self-maintenance. This should be discussed.

Answer

The referee claims that "*Many scientists would not consider the autocatalytic oligomerization presented here as having a link to biology because the proliferation displayed by living organisms through growth and division is a biological process but very distinct from the phenomena described here.*" But that claim is not entirely correct. From the point of view of the origin of life, this system can instill an image of chemistry and biology into readers sufficiently. This is because that at the era of the first life, primitive organisms must have proliferated by a much simpler physical process not by the same complex and sophisticated molecular machinery as today's organisms [Szostak, J. W., Bartel, D. P. & Luisi, P. L., "Synthesizing life", *Nature* **409**, 387–390 (2001). Leaver M, Domínguez-Cuevas P, Coxhead JM, Daniel RA, Errington J . "Life without a

wall or division machine in *Bacillus subtilis*". *Nature* **457**, 849–53 (2009).]. This study shows for the first time that molecular assemblies can be proliferated through primitive polymerization and physical action. Therefore, the phrase "*This study may serve to explain the emergence of the first living organisms on primordial Earth.*", which the referee noted as an overstatement, is reasonable enough. For above reasons, the current study can be sufficiently regarded as providing one of the missing links between chemistry and biology "in the origin of life". As a supplement to this discussion, we also re-cited a paper from Mann's lab, which is a laboratory of origin of life from chemical field, as ref5.

A proliferating protocell is, of course, leading to an organism with the universal property of proliferation, i.e., a 'universal ancestor', but not necessarily a common ancestor. Therefore, we revised our use of the words 'universal ancestor' in the previous revised manuscript, according to the referee's point. These terms should be used clearly and deliberately. In the current version, we did not conclude that "their system is, or contributes in some way to a universal common ancestor." On the other hand, the use of 'ancestor' tends to cause confusion for readers. Accordingly, we changed the word 'ancestor' to 'primitive living things'. For the similar reason, we revised "the origin of life" to "the origins of life" in the text and title.

It is true that benzyl mercaptan, which is a leaving group, is difficult to synthesize in the primitive earth environment and regarded as a model of primitive hydrophobic thiol. However it could be possible to form it from primitive molecules in a stepwise manner [Zhou, Minghe, China, CN106397120 A 2017-02-15; Liu, Xiaozhi; et al, China, CN101186591 A 2008-05-28.] by changing the organic solvent, and it is not entirely far from the pre-biological environment. An alkylthiol, for example, would be a good prebiotic candidate for a hydrophobic thiol [Huber C, Wächtershäuser G. Activated acetic acid by carbon fixation on (Fe,Ni)S under primordial conditions. *Science*. 1997;276(5310):245-247. doi:10.1126/science.276.5310.245]. Syntheses of cysteine and cystine have been attempted by Sagan et al. under more-or-less prebiotic conditions in a reductive environment that are skeptical at the present [Khare, B., Sagan, C. Synthesis of Cystine in Simulated Primitive Conditions. *Nature* 232, 577–579 (1971). doi:10.1038/232577a0], but their prebiotic synthesis pathways are currently unknown

[Parker, E. T. et al. Primordial synthesis of amines and amino acids in a 1958 Miller H₂S-rich spark discharge experiment. Proc Natl Acad Sci USA 108, 5526–5531 (2011). doi: 10.1073/pnas.1019191108]. Therefore, our system can be regarded as a model for the origin of life because we used model molecules as leaving groups and achieves growth by self-production in a prebiotic environment. If an aromatic ring is attached to the alkyl thiol, the same cation- π interaction as benzyl mercaptan can be expected. In addition, the other thiols could also form coacervated droplets unless the interaction between thiols and peptides is limited to the cation- π interaction. Our goal is to show that even simple molecules can produce coacervate droplets based on interactions and build a chemical life model. Furthermore, the purpose of this study is to provide a basis for researcher's discovery of molecules that may have existed at the beginning of life from droplet-forming molecules. Therefore, the use of benzyl mercaptan as a model molecule for cation- π interactions in this study broadens researcher's horizon for future molecular discoveries.

Minor points:

4. Figure S17 still has an incorrect y-axis

Answer

Thank you for pointing out. We changed the y-axis in FigureS16 from micrometers to nanometers.

Figure A3 | A raw image of Figure 4e.

5. p.11, line 5-8 ("However...residual proportion.") The meaning of this sentence is not clear.

Answer

Since the sentence was definitely difficult for readers to understand, we deleted it.

6. p.14 line 7, remove "the" before fusion

Answer

As the reviewer pointed out, "the" has been removed.

7. Figure 4c: why is the x-axis scale different from 4b? If there are small droplets, they would be expected at small particle size.

Answer

As the reviewer noted, we revised the particle size range in Figure 4c as shown in below. The size range was extended to show that small size droplets do not appear. When we revised and resubmitted the manuscript last time, we draw the figure by mistake.

Figure A3 | A new image of Figure 4c.

Reviewer #3 (Remarks to the Author):

I would like to thank the authors for preparing a very thorough and detailed response, and I think that the manuscript has improved greatly. I do have a few remaining minor points that I would suggest the authors consider; after these points are addressed, I believe that the manuscript would be ready for publication.

The referee made this paper more enlightening for readers in the research field of origins of life. We are deeply grateful.

Minor points:

1. In the original point 7, I had suggested to discuss the prebiotic relevance of the concentration of the chemicals used in the study. However, the authors misunderstood my point and have in fact pointed out the opposite: that the concentrations used are too high to have been prebiotically plausible. An additional statement or short discussion about why it is reasonable to use mM concentrations of chemicals in this study when it is in fact difficult (or almost impossible) to synthesize pure molecules to the mM order in an actual prebiotic environment.

Answer

The reviewer previously pointed out that the concentrations of the molecules used in our experiments might be higher than those at the prebiotic era. We agree with that point. The reaction in this system proceeds even at low concentrations if it took over time. However, we added the following sentences to the text because the issue about compound concentration is an important viewpoint on the origin of life; *"In this system, phase separation eventually occurred and the molecules were enriched even at low concentrations, although the reaction time was longer. Thus, the concentration of this system merely affects the chemical reaction rate exclusively. For enrichment process at low concentrations in a prebiotic environment, the dry-wet cycles may be useful⁴⁷."* (P.25, L.442-445).

2. In point 7, the prebiotic explanation of the plausible synthesis of the components is very reasonable and comprehensive, and while this level of detail should not be included in the main text, I wonder if it would make sense to include this very nice and detailed discussion somewhere in the SI.

Answer

We thank the referee for evaluating our experimental results and descriptions of prebiotic scenario.

Thanks to the referee's points, this paper becomes a very fruitful. We added the descriptions in the Supplementary Note section (Note S1).

3. All mentions of “origin of life” should ideally be “origins of life”, because it is unknown whether there was one or many origins that led to our current life; it is also unknown whether we are the only life.

Answer

The referee's point is correct. In fact, other referees also pointed this out. We changed "origin of life" to "origins of life" in title and text in the revised manuscript.

4. While I believe that the English of the manuscript has improved significantly, there are still a number of minor grammatical errors throughout the document. I would suggest the authors review the document carefully one last time. Here are some examples from only the introduction (not exhaustive).

Answer

We checked and revised the grammar or typos you pointed out.

Remarks to the author's rebuttal for the paper " Self-proliferating Liquid-Liquid pHase separated Droplet; the missing link between Chemistry and Biology on the Origin of Life" by Matsuo and Kurihara:

Thank you to the author for the rebuttal to the previous review. Please find responses to these below:

1. Even after the revisions, the definitions as well as the experimental data to support the claims that the system, proliferates and self-reproduces are still not in line with general descriptions of these processes. For example, for a proliferating system, one would expect to observe an increase in number of droplets. The point that the droplets reach a steady state in size, alone, as has been described describe proliferation. For a self-reproducing system one would expect that new droplets are made from existing droplets. The data presented to support this argument could be attributed to continual nucleation and formation of new droplets. The additional experiment Figure A1a and A1b, S16 does not support their arguments as there do not appear to be any droplets in the solution. For the self-maintenance argument the FACS data shows a change in distribution with the addition of both RNA and HPC but does this really indicate self-maintenance.

Answer

The various experimental data about self-reproduction in the main text and the Supplementary Information clearly have showed that droplets grew and divided by self-reproduction. Proliferation of droplets in our study, as clearly defined in the main text (P.3, L.60), means growth and division by self-reproduction. In other words, all of the data (Figs.2-4, Figs.S17-20) proving growth and division of the droplets by self-reproduction are equivalent to proving proliferation of them. Therefore, referee 1's opinion, "The point that the droplets reach a steady state in size, alone, as has been described describe proliferation. For a self-reproducing system one would expect that new droplets are made from existing droplets", is not valid. The increase in numbers of droplets is strongly suggested by the consistency of particle size in each generation even with addition of precursor M_{pre} . However, we added the following experiment (Fig. A1) to enlighten readers. In the following additional experiment, after the addition of monomer precursor, the dispersion was diluted by the incremental amount after the increase in droplets. It shows that a constant number density was achieved. In the current Fig.A1 below, the observed number of droplets was estimated with the actual volume of the dispersion, thus indicating that the droplets proliferated. We added this result as the Fig. S20 in the supplementary information and cited this in the text.

Additionally, the reviewer claims "The additional experiment Figure A1a and A1b, S16 does not support their arguments as there do not appear to be any droplets in the solution", which is not correct. Additional experiments (original S16 in the previous SI, new S19 in the revised SI) show that droplet does not grow via droplet fusion. Since growth of droplet size is derived from either self-production of droplet or fusion among droplets, the growth in the current study can be nothing but derived from self-production.

The reason that the droplets are invisible in the former Fig. S16 (Fig.A1 in the former answer sheet) is that the droplets are reduced to a size below that which can be observed with an optical microscope by extruding them through a 100 nm filter to match the size of the particle size distribution measurements. This experiment means that the number of droplets does not increase unless the monomer precursor is added again. In the observation experiments of droplet growth, we observed that the number of droplets increases with the addition of food in a wide field of view (Figure 2d in the article), but when the number of droplets increases to a certain extent, fusion becomes dominant due to the close distance between the droplets. Therefore, in the following additional experiment, after the addition of monomer precursor, the dispersion was diluted by the incremental amount after the increase in droplets. It shows that a constant number density was achieved.

As reviewer mentioned, in the new Figure 6, when lipids were added to the droplet dispersion, the droplet size decreased, indicating dissolution. On the other hand, when both nucleic acids and lipids are added to the dispersion, the droplets show resistance to dissolution, indicating self-maintenance.

I still am confused by some aspects of this rebuttal. For example, does the data in figure 4e, also be attributed to fusion events? Please clarify the y-axis label for the new figure A1. If it is a rate, should there be units and how does this then correlate to the number density. In addition, the comment regarding new figure S19, is not entirely clear. The images here so no turbidity or droplets so how can the droplets be reproducing?

With reference to figure 2d in the article, please make it clearer what they are referring to when showing that a "number density was achieved".

With respect to the new figure 6. Does this show stability rather than self maintenance?

4. The figures should be further improved. A full page figure will not be so easily reproduced into any journal

Answer

We divided Figure 5 into two parts; the new Figure 5 (microscopy images of incorporation of both RNA and lipid) and the new Figure 6 (microscopy images, line profiles and flow cytometry experiments of incorporation of RNA or lipid). We also revised the index and sentences in the manuscript accordingly. As the reviewer pointed out, we believe it is easier for the reader to read.

Please consider modifying the other figures too.

5. Thank you for the expanded data for Mass spec. Please also include a better explanation for the labels (1A, 2A etc....)

Answer

We labeled the character (1A, 2A, ...) beside the peaks showing each peptide in Figure S10c.

Could the authors add a description as to what 1A, 2A etc means.

7. For figure 4c, was extrusion undertaken on the droplets? What is the error in the number density? Is the difference in the number density significant? With

regard to this point...''To validate the reviewer's proposed condition of whether fusion affects droplet growth, we added deionized water without Mpre to the extruded droplet dispersion. No significant droplet formation was observed (Fig. A1).Therefore, the increase in particle size after extrusion is considered to be induced by self-production, not by fusion.'' Figure A1, shows no droplets at all. This experiment should be undertaken with a dispersion of droplets.

Answer

In the experiment shown in Figure 4c, particle size measurements were performed on dispersions extruded through a membrane filter with a 100 nm diameter pore. As we mentioned in Question 1, there are no droplets in the Figure S16 (Figure A1 in the last answer sheet). However, the droplets are below the size that can be observed under the microscope because the size of the droplets were filtered to match the particle size distribution measurement conditions. The S16 experiment shows that the droplets do not grow to a microscopy-observable size spontaneously when they are extruded and fused. This means that the size of the droplets does not change as shown in Figure 4c unless they are fed monomer precursors. As for the number density experiment, the growth of droplets was shown in Figure S20 (additional experiment Figure A1). The number density is significantly different between generations. Although the referee claims that "Figure A1, shows no droplets at all", the sentence clearly proved that the dispersed droplets are divided down to a nanometer size, which cannot be observed by the microscope and also can only be detected by DLS size measurement. Thus, the previous Fig.A1 and Fig.4c in the article text satisfied the referee's requirement of "This experiment should be undertaken with a dispersion of droplets" in the first place.

Please could the authors clarify the issue raised earlier with the figure labelling on new figure S20.

Additional point:

Haldane proposed a droplet surrounded by an oil layer whilst Oparin is credited with the coacervate idea.

Reviewer #2:

Remarks to the Author:

The authors have submitted a newly revised manuscript in which most of the reviewers' comments have been addressed. I would like to thank the authors for their detailed response. In my opinion, these changes have improved the manuscript substantially. I have a few remaining points that must be addressed before the manuscript can be published. If these points are adequately addressed, I believe this manuscript should be accepted for publication.

Composition:

The authors' crude phase diagram in Fig. A1 is very helpful to support their claim that BnSH and peptides are needed for phase separation. I suggest moving the right panel of this figure with a more easily readable x-axis (e.g., powers of 10) to the main text – it would fit well together with the table in Fig.3.

The authors also explain in their response that it was not possible to separate the droplets from solution, or to do NMR, HPLC or UV measurements on the coacervate phase. While I do not understand why the droplets could not be separated (they clearly fuse, and they should either settle or cream, and could be separated by removing either the top or bottom phase), this information/discussion is very important for future readers who want to build upon this work. A more detailed explanation of what has been tried by the authors to separate the coacervate phase, a note that separating the phases was not successful, and a possible reason why the analysis could not work should therefore be included in a Supplementary note, in order to avoid needless repetitions.

Self-reproduction:

The authors' control experiment in Figure A2 is helpful to show they could distinguish extruded from non-extruded droplets. What is missing from this part of the manuscript and supporting information is a control experiment like in Fig. 4e, in which the authors do not apply an extrusion step (i.e., stop just before the first black arrow and monitor the size over time for 100 h), and a control experiment, in which the authors do not add fresh nutrients (i.e., stop just after the first black arrow, before the first open arrow and monitor the average size over time for 100 h). From what the authors answered, it can be assumed that the size should not change significantly without nutrient addition or shear – which would be important to show (as a supporting figure).

Prebiotic relevance and missing link:

I think the authors have missed one point in the previous discussion. I do understand that benzyl mercaptan was used as a model of primitive hydrophobic thiols, but the prebiotically more plausible alternatives that the authors discuss (alkyl thiols) would, according to the authors' proposed mechanism of associative phase separation (cation- π) not be suitable to form coacervate droplets. While this in no way detracts from the results, it should be discussed in the Discussion section that alkyl thiols would not be expected to form CDs.

Figure axes:

Several figures have unclear axes labels.

- Figure 3b and c: the label indicates that a rate is plotted, but this should be Residual amount of Mpre and Amount of droplet material, respectively. If the rate plateaus at 70, droplets would keep being formed while there is no Mpre left.

- Figure 5b and c: the labels here refer to the fluorescent probes (TAMRA, BODIPY), while the incorporation is determined by the biomolecules to which they are attached. The authors should modify the labels to RNA-TAMRA and Lipid-BODIPY (ideally, specify the lipid, so PLPC-BODIPY rather than Lipid-BODIPY).

- Figure 6a, b, c and d: same as Figure 5: RNA-TAMRA and PLPC-BODIPY instead of TAMRA and BODIPY.
- Figure S13c: same as Figure 3, the y-axis most likely indicates has amount or droplet %, not rate plotted. The inset has no axes labels, and is also not described in the caption.
- Figure S18: the unit of the reaction rate $^{(1/3)}$ on the y-axis is missing
- Figure S20: the unit of growth rate on the y-axis is missing. The numbers could be chosen more naturally (log scale of 10)

Minor points in text:

- Line 30: Oparin wrote his first book on the origin of life in 1923, but this book contains no mention of coacervates yet. Only in the later versions of the book were coacervates introduced (1936). This sentence should therefore be changed to "in the 1930s".
- Line 34: 2x "are" should be "is" (singular function)
- Line 64: I disagree with the authors that the proliferation of molecular assemblies is a "biological" property. Classical work by Rebek, Von Kiedrowski and Ghadiri on replicators is considered a (very significant) advance in systems chemistry.
- Line 124: dispended (what does this mean?)
- Line 185: The rate of formation of droplets – do the authors also mean the amount or % of droplets here?
- Line 207: "predicted" – probably the authors mean "possible scenarios of"
- Line 236: "then divided" – this should be replaced by "and were then divided by extrusion" (or: by applied shear). The droplets did not divide spontaneously.
- Line 410: "a steady growth-division cycle" – the need for extrusion/shear should be mentioned here.
- The authors have used mixed UK/US spelling (e.g., behaviour and behavior). Please check the spelling carefully.

Reviewer #1 (Remarks to the Author):

The referee made this paper more enlightening for readers. We are deeply grateful.

1. I still am confused by some aspects of this rebuttal. For example, does the data in figure 4e, also be attributed to fusion events? Please clarify the y-axis label for the new figure A1. If it is a rate, should there be units and how does this then correlate to the number density. In addition, the comment regarding new figure S19, is not entirely clear. The images here so no turbidity or droplets so how can the droplets be reproducing? With reference to figure 2d in the article, please make it clearer what they are referring to when showing that a “number density was achieved”. With respect to the new figure 6. Does this show stability rather than self-maintenance?

Answer 1

1) Figure 4e (new Fig.5b)

Fig. 4e (new Fig.5b) plots the change in droplet size after physical stimulation (extrusion), with the first stage due to reaction and the second stage due to fusion for each generation, as shown in new Fig. S19 and Fig. S20 (only for 1st generation).

2) Figure A1 (new Fig.S18) and Figure 2d

Scheme A1.

The vertical axis shows a relative number. Therefore, the unit of y value is dimensionless. Because an equal amount of M_{pre} solution was added to each generation one, the relative volume of the n -th generation's solution to the first generation one is n . Therefore, the relation between the value of

vertical axis and number density is: Relative density = Relative droplet number / Relative volume (n) (Scheme A1).

The vertical axis has been revised as shown in Figure A1 (New Figure S18).

Figure A1

3) Figure S19 (New Fig. S20 a, b)

Even with low turbidity, there are droplets of small size (ca. 100 nm), which can self-reproduce. These droplets are sufficiently smaller than the wavelength of visible light that the turbidity derived from the scattered light is extremely small. The presence of these small droplets is guaranteed by the DLS measurement results (Fig. S20 c). In addition, particles smaller than the half-width of visible light cannot be observed by phase contrast microscopy due to the measurement principle.

To maintain readability for the reader, the following sentence was added to the article manuscript (P.15 L.279-282). *Droplets with small diameters (< ca. 100 nm) could disperse and self-reproduce in the dispersion. However the intensity of turbidity, which is quantified on the basis of scattered light, was extremely small because these droplets were much smaller than the wavelength of visible light.*

4) Figure 6

Self-maintenance in the article sentences around Fig.6 was changed to stability. Self-maintenance, which was used in the introduction, was changed to robustness.

2. Please consider modifying the other figures too.

Answer 2

We divided Fig. 4 into two sections: size distributions (a, b, c) and self-reproduction (d, e). Therefore, we revised Figs. 4d and 4e as Figs. 5a and 5b. The indexes of the subsequent figures were modified to reflect these changes.

We revised error bars in the Fig. 2c. Because the Fig.2 is a two-column format figure, not take up an entire page, we believe it easier for readers to comprehend.

On the advice of other reviewers, we merged Fig. S12 into Fig. 3 as Figs. 3b and 3c.

Figure A2

3.

Could the authors add a description as to what 1A, 2A etc means.

Answer 3

We added description of peptides 1A, 2A, etc. to the figure legend in Fig. S10c as follows; *All m/z values obtained from the ESI-TOF mass spectrum were shown. In the spectrum, nA, nB (n=1, 2....) means detected peptides. Here A means non-thioesterified ester, B means thioesterified ester and n means degrees of polymerisation.*

4. Please could the authors clarify the issue raised earlier with the figure labelling on new figure S20.

Answer 4

As we answered in Answer 1, we changed it.

5. Additional point:

Haldane proposed a droplet surrounded by an oil layer whilst Oparin is credited with the coacervate idea.

Answer 5

We revised the text in the introductory section (P. 2 L.34-36) as follows; *In the 1920s, Oparin³ and Haldane⁴ independently proposed an origin-of-life scenario, and in the 1930s they claimed that a protocell, i.e., a primitive cell, was a proliferating droplet such as a coacervate droplet (CD).*

Reviewer #2 (Remarks to the Author):

The referee made this paper exquisite for readers. We deeply appreciate the referee.

The authors have submitted a newly revised manuscript in which most of the reviewers' comments have been addressed. I would like to thank the authors for their detailed response. In my opinion, these changes have improved the manuscript substantially. I have a few remaining points that must be addressed before the manuscript can be published. If these points are adequately addressed, I believe this manuscript should be accepted for publication.

1. Composition:

The authors' crude phase diagram in Fig. A1 is very helpful to support their claim that BnSH and peptides are needed for phase separation. I suggest moving the right panel of this figure with a more easily readable x-axis (e.g., powers of 10) to the main text – it would fit well together with the table in Fig.3.

The authors also explain in their response that it was not possible to separate the droplets from solution, or to do NMR, HPLC or UV measurements on the coacervate phase. While I do not understand why the droplets could not be separated (they clearly fuse, and they should either settle or cream, and could be separated by removing either the top or bottom phase), this information/discussion is very important for future readers who want to build upon this work. A more detailed explanation of what has been tried by the authors to separate the coacervate phase, a note that separating the phases was not successful, and a possible reason why the analysis could not work should therefore be included in a Supplementary note, in order to avoid needless repetitions.

Answer 1

We moved and integrated Fig. S12b as Fig. 3b as shown below.

Figure A1. Integrated figure.

We added the following sentence in Supplementary Note 2: *The total volume of the droplets in the sample was too small (ca. 1 μL / sample) to be centrifuged and analysed by HPLC.*

2. Self-reproduction:

The authors' control experiment in Figure A2 is helpful to show they could distinguish extruded from non-extruded droplets. What is missing from this part of the manuscript and supporting information is a control experiment like in Fig. 4e, in which the authors do not apply an extrusion step (i.e., stop just before the first black arrow and monitor the size over time for 100 h), and a control experiment, in which the authors do not add fresh nutrients (i.e., stop just after the first black arrow, before the first open arrow and monitor the average size over time for 100 h). From what the authors answered, it can be assumed that the size should not change significantly without nutrient addition or shear – which would be important to show (as a supporting figure).

Answer 2

When the droplet solution was not subjected to filtration, a droplet size increased up to 24 h after the **Mpre** addition. The size had not changed after 24 h (at least for 120 h after the addition). On the other hand, when the solution was filtrated 24 h after **Mpre** addition, the particle size decreased to around 100 nm immediately after filtration, and thereafter the particle size remained almost constant at least until 96 h after addition. However, it was not possible to measure the droplet size of the filtrated dispersion 120 h after **Mpre** addition. This is because the loss of droplets due to the volatilization of BnSH, which is a droplet component, becomes not negligible. These results strongly suggested that the droplet size after the second generation was increased by self-reproduction of droplets through **Mpre** addition. Based on these results, we added Figure S20c and revised the text (P15, L288-291) as follows; *In addition, the fact that no significant increase in particle size was observed when only water*

was added to the extruded droplet dispersion with DIC microscope and DLS particle analyser (Fig. S20) indicated that the initial increase in particle size was not due to the fusion of droplets after extrusion but instead was the result of the reaction.

Figure A2. Change in droplet size. ● : Droplet dispersion with added **Mpre**. ○ : Filtered droplet dispersion 24 h after **Mpre** addition.

3. Prebiotic relevance and missing link:

I think the authors have missed one point in the previous discussion. I do understand that benzyl mercaptan was used as a model of primitive hydrophobic thiols, but the prebiotically more plausible alternatives that the authors discuss (alkyl thiols) would, according to the authors' proposed mechanism of associative phase separation (cation- π) not be suitable to form coacervate droplets. While this in no way detracts from the results, it should be discussed in the Discussion section that alkyl thiols would not be expected to form CDs.

Answer 3

As reviewer pointed out, we modified the sentence (P.10, L.181-189) as follows; *In addition to hydrophobic interactions, π - π interactions and cation- π interactions are plausible mechanisms for the droplet formation. The peak shift of the NMR spectrum shows that the π - π interactions effected (Fig. S13a). The fact that the terminus of the peptide has an ammonium cation and that BnSH has a benzene ring suggests the possibility that cation- π interactions is involved in droplet formation. Indeed, it was reported that LLPS droplets in vivo were formed due to cation- π interactions between lysine residues with an ammonium cation and other amino acids residues with an aromatic ring in the protein side chain³⁶. The thiol's distinctive contribution to droplet formation, such as thiol- π interactions or disulfide bonds, should also be considered³⁷.*

Such a contribution to droplet formation can also be expected from alkylthiols. We added this idea to Discussion section (P.25, L.469-471) as follows; *BnSH is a model hydrophobic thiol. For example, an alkylthiol that is capable of forming droplets due to hydrophobic interactions would be a good prebiotic candidate for a hydrophobic thiol*⁵¹.

4. Figure axes:

Several figures have unclear axes labels.

- **Figure 3b and c:** the label indicates that a rate is plotted, but this should be Residual amount of Mpre and Amount of droplet material, respectively. If the rate plateaus at 70, droplets would keep being formed while there is no Mpre left.

Answer 4

We revised the vertical axis as shown in Figure A1 and the text in the legend.

5. Figure 5b and c: the labels here refer to the fluorescent probes (TAMRA, BODIPY), while the incorporation is determined by the biomolecules to which they are attached. The authors should modify the labels to RNA-TAMRA and Lipid-BODIPY (ideally, specify the lipid, so PLPC-BODIPY rather than Lipid-BODIPY).

- **Figure 6a, b, c and d:** same as Figure 5: RNA-TAMRA and PLPC-BODIPY instead of TAMRA and BODIPY.

Answer 5

We revised the labels. We revised lipid to HPC.

6. Figure S13c: same as Figure 3, the y-axis most likely indicates has amount or droplet %, not rate plotted. The inset has no axes labels, and is also not described in the caption.

Answer 6

We revised the vertical axis as below. We also added an explanation of insert figure to legend as follows; *Peaks X and X' were assigned to the benzyl moieties before and after BnSH was removed, respectively. The conversion to droplet material (%) was estimated from the integrated peak areas and equated to $100 \times X' / (X + X')$, where X and X' are the integrated peak areas of the benzyl moieties before and after BnSH was removed, respectively. The ranges of the chemical shifts of the X and X' peaks were 7.0–7.5 and 6.5–7.0, respectively. c A plot of the conversion to droplet material (%) from 0 h to 16 h after addition. The inset shows a magnified view of Fig. 13c up to 60 min.*

7. Figure S18: the unit of the reaction rate $^{1/3}$ on the y-axis is missing

Answer 7

We revised the vertical axis.

8. Figure S20: the unit of growth rate on the y-axis is missing. The numbers could be chosen more naturally (log scale of 10)

Answer 8

We revised the vertical axis.

9. Minor points in text:

Line 30: Oparin wrote his first book on the origin of life in 1923, but this book contains no mention of coacervates yet. Only in the later versions of the book were coacervates introduced (1936). This sentence should therefore be changed to “in the 1930s”.

Answer 9

We revised the sentence (P2, L34-36) as follows; *In the 1920s, Oparin³ and Haldane⁴ independently proposed an origin-of-life scenario, and in the 1930s they claimed that a protocell, i.e., a primitive cell, was a proliferating droplet such as a coacervate droplet (CD).*

10. Line 34: 2x “are” should be “is” (singular function)

Answer 10

We revised the sentence.

11. Line 64: I disagree with the authors that the proliferation of molecular assemblies is a “biological” property. Classical work by Rebek, Von Kiedrowski and Ghadiri on replicators is considered a (very significant) advance in systems chemistry.

Answer 11

The study of Rebek or Kiedrowski on molecular replication is important in system chemistry. There is a clear difference between replication, in which a complete copy is made by a template reaction, and self-reproduction, in which an incomplete copy is made without a template. Since continuous self-reproduction does not necessarily require artificial perturbation, self-reproduction should be clearly distinguished from replication (cf. P. L. Luisi “Emergence of life” Chp.7). We added the important papers commented by Reviewer. We replaced the term “replicator” to "molecular replicator" because they are used molecular template replication reaction.

12. Line 124: dispended (what does this mean?)

Answer 12

We revised the “dispersed”. The text was checked by English editing service again.

13. Line 185: The rate of formation of droplets – do the authors also mean the amount or % of droplets here?

Answer 13

We revised “the conversion to droplet material.”

14. Line 207: “predicted” – probably the authors mean “possible scenarios of”

Answer 14

We revised the sentence.

15. Line 236: “then divided” – this should be replaced by “and were then divided by extrusion” (or: by applied shear). The droplets did not divide spontaneously.

Answer 15

We revised the sentence.

16. Line 410: “a steady growth-division cycle” – the need for extrusion/shear should be mentioned here.

Answer 16

We revised the sentence.

17. The authors have used mixed UK/US spelling (e.g., behaviour and behavior). Please check the spelling carefully.

Answer 17

The text was checked by English editing service again.

Reviewers' Comments:

Reviewer #1:

Remarks to the Author:

All of the comments have been addressed and would be suitable for publication with Nature Communications

Reviewer #2:

Remarks to the Author:

The authors have addressed my comments and made appropriate changes to their manuscript. I have two remaining points, one of which was introduced by one of the changes the authors made. Could the authors please clarify this in their manuscript?

1. Figure 3b: could the authors please use more easily readable concentrations on the x-axis (e.g., powers of 10)?

2. When discussing the new control experiment in Figure S20, the authors write on line 293-295 that "...the initial increase in particle size was not due to the fusion of droplets after extrusion but instead was the result of the reaction."

However, right after that they also write that: "...the droplets generated by the reaction grew primarily by fusion (Fig. S19)." These two statements seem to be conflicting, in particular because the size of the droplets in the control in Fig. S20 was measured at much longer times than 3 h. Should these droplets not have increased in size by fusion? Or is this rate too small to be measured? Could the authors please clarify this for the readers in their manuscript?

Reviewer #2 (Remarks to the Author):

The referee made this paper exquisite for readers. We deeply appreciate the referee.

1. Figure 3b: could the authors please use more easily readable concentrations on the x-axis (e.g., powers of 10)?

Answer 1

We corrected the x-axis in the Figure 3b according to the reviewer's comment.

2. When discussing the new control experiment in Figure S20, the authors write on line 293-295 that “...the initial increase in particle size was not due to the fusion of droplets after extrusion but instead was the result of the reaction.” However, right after that they also write that: “...the droplets generated by the reaction grew primarily by fusion (Fig. S19).” These two statements seem to be conflicting, in particular because the size of the droplets in the control in Fig. S20 was measured at much longer times than 3 h. Should these droplets not have increased in size by fusion? Or is this rate too small to be measured? Could the authors please clarify this for the readers in their manuscript?

Answer 2

We revised the sentences as follows (P.15, L.291-295): “However, the fact that the correlation coefficient between the two experiments (Figs. S14, 5b) more than 3 h after mixing was only 0.068 suggested that the generated droplets grew primarily by fusion after the conversion to droplet material reached plateau. These results imply that more than 3 h after mixing, i.e., at least when the number density of droplets is small, the contribution of fusion to the increase in particle size is negligible.”